# Optimistic Exploration with Learned Features Provably Solves Markov Decision Processes with Neural Dynamics

**Sirui Zheng**[1]    **Lingxiao Wang**[1]    **Shuang Qiu**[2]    **Zuyue Fu**[1]    **Zhuoran Yang**[3]
**Csaba Szepesvári**[4,5]    **Zhaoran Wang**[1]

## Abstract

Incorporated with the recent advances in deep learning, deep reinforcement learning (DRL) has achieved tremendous success in empirical study. However, analyzing DRL is still challenging due to the complexity of the neural network class. In this paper, we address such a challenge by analyzing the Markov decision process (MDP) with neural dynamics, which covers several existing models as special cases, including the kernelized nonlinear regulator (KNR) model and the linear MDP. We propose a novel algorithm that designs exploration incentives via learnable representations of the dynamics model by embedding the neural dynamics into a kernel space induced by the system noise. We further establish an upper bound on the sample complexity of the algorithm, which demonstrates the sample efficiency of the algorithm. We highlight that, unlike previous analyses of RL algorithms with function approximation, our bound on the sample complexity does not depend on the Eluder dimension of the neural network class, which is known to be exponentially large (Dong et al., 2021).

**Keywords:** Reinforcement Learning, Neural Network, Representation Learning.

## 1 Introduction

Reinforcement learning (RL) aims to accomplish sequential decision-making in an uncertain environment via iteratively interacting with the environment (see Sutton et al. (1998)). Equipped with modern function approximators such as deep neural networks, deep RL algorithms achieve tremendous empirical successes (Mnih et al., 2015; Silver et al., 2017; Hafner et al., 2019).

Despite its empirical successes, the theoretical understanding of deep RL is relatively underdeveloped. There are several recent works (Abbasi-Yadkori et al., 2019; Wang et al., 2019; Fan et al., 2020) that analyze RL algorithms with neural network parameterization, including policy iteration (PI) (Lagoudakis & Parr, 2003), policy gradient (PG) (Williams, 1992) and deep Q-learning (Mnih et al., 2013). However, those works depend on restrictive assumptions that either the agent has access to a simulator or the MDPs have bounded concentrability coefficients, which in fact imply that the state space is already well-explored. Another line of research (Jiang et al., 2017; Jin et al., 2020; Cai et al., 2019; Du et al., 2021) further removes such assumptions by conducting provably efficient exploration in RL. Such a direction of research typically hinges on a low-rank MDP assumption. Thus, those works either assume that the MDP is linear in the known feature or propose computational-inefficient algorithms, limiting the ability to explore the environment with neural network parameterization. To explore the environment with neural network parameterization, a recent line of work (Wang et al., 2020; Jin et al., 2021a) analyzes the use of general function approximators in RL, covering neural network parameterization as a special case. Such analyses typically depend on the Eluder dimension (Russo & Van Roy, 2013), which unfortunately can be exponentially large even for a simple neural network class (Dong et al., 2021) and thus makes the results statistically inefficient for neural network parameterization. Therefore, we raise the following question:

[1]Northwestern University    [2]University of Chicago    [3]Yale University    [4]University of Alberta
[5]DeepMind    Corresponding authors: `siruizheng2025@u.northwestern.edu`

*Can we design RL algorithms that can conduct provably efficient exploration in structured environments with neural network parameterization?*

Specifically, Our goal is to **develop computational-efficient algorithms whose sample efficiency does not depend on the Eluder dimension of neural networks** for structured environments with neural network parameterization. Our key insight is that, when the transition dynamics is captured by an energy-based model, we leverage the spectral decomposition of the kernel such that the challenge of distribution shift is characterized by the effective dimension of the kernel. To illustrate this insight, we propose a new model called *MDPs with neural dynamics*, which allows neural network parameterization and captures various MDP models proposed in previous works, including the KNR model (Kakade et al., 2020) and the linear MDP model (Jin et al., 2020). We then propose an algorithm, namely, *Exploration with Learnable Neural Features* (ELNF), and show that ELNF is sample efficient. ELNF iteratively fits the transition dynamics and reward functions with neural networks. Upon fitting the models, ELNF conducts exploration based on upper confidence bounds (UCB) (Abbasi-Yadkori et al. (2011)), which are obtained from the feature maps that correspond to the fitted model. We remark that the bonus in ELNF can be efficiently computed.

**Contributions.** Our contribution is threefold. First, we identify a class of models that incorporates NN feature representation, which captures nonlinearity in the transition dynamics beyond the KNR and linear MDP model. We also show that our proposed setting can generalize to models in previous works (Kakade et al. (2020), Ren et al. (2021)). Second, we propose a new algorithm, namely ELNF, which tackles our proposed *MDPs with neural dynamics*. **Our algorithm is computationally efficient when we have an optimization oracle for the model estimation**. Third, we analyze the sample complexity of ELNF and show that ELNF is sample efficient. A key feature of ELNF is that **the sample complexity of** ELNF **depends only on the covering number of neural network classes and does not depend on the corresponding Eluder dimension.** We highlight that **our work is the first to cover arbitrary NN classes with bounded log-covering numbers.** In contrast, **previous research typically depends on the Eluder dimension (Russo & Van Roy, 2013) of the hypothesis class, which is exponentially large for simple neural network classes (Dong et al., 2021).**

## 1.1 Related Work

Our work is closely related to the line of research on provably efficient exploration in the function approximation setting (Jiang et al., 2017; Jin et al., 2020; Cai et al., 2019; Du et al., 2021; Uehara et al., 2021; Zhang et al., 2022a). Such a line of research typically hinges on MDPs with a low-rank structure. For instance, the study of linear MDPs (Jin et al., 2020; Cai et al., 2019) requires that the transition dynamics are linear in the known feature map. In contrast, the feature maps are unknown in our setting and need to be estimated. The study of low-rank MDPs (Jiang et al., 2017; Du et al., 2021; Uehara et al., 2021; Ren et al., 2022) is more closely aligned to our work in the sense that the feature map is unknown and needs to be estimated. Jiang et al. (2017) and Du et al. (2021) require optimistic planning over the confidence set of transition dynamics, which is computationally inefficient. Uehara et al. (2021) and Ren et al. (2022) propose an algorithm for low-rank MDP that is both computationally efficient and sample efficient. Nevertheless, they only consider finite hypothesis classes, and require sampling from the stationary distribution of the MDP.

Our work is also related to the study of provably efficient exploration with general function approximation (Wang et al., 2020; Jin et al., 2021a). Nevertheless, previous results typically depend on the Eluder dimension (Russo & Van Roy, 2013) of the hypothesis class, which is exponentially large for simple neural network classes (Dong et al., 2021). Yang et al. (2020) achieves sample-efficient exploration based on the overparameterized neural networks (Simsek et al., 2021) as the function approximator. However, their analysis hinges on the neural tangent kernel (NTK) and can not handle NNs beyond NTK regime. In contrast, our analysis can adapt to generic neural network classes.

Our work is also related to the analysis of model-based RL (Osband & Van Roy, 2014; Ayoub et al., 2020; Kakade et al., 2020) and representation learning (Ren et al., 2021; Nachum & Yang, 2021; Zhang et al., 2022b). The definition of our *MDPs with neural dynamics* generalizes that in Kakade et al. (2020) and Ren et al. (2021). In contrast to the KNR model in Kakade et al. (2020), we can handle the infinite neural network hypothesis class and do not require the nonlinear feature map to be known. Ren et al. (2021) require sampling from the posterior distribution of the hypothesis class, which is computational-inefficient when the hypothesis class is large. In addition, the sample

complexity bound of Ren et al. (2021) depends on the Eluder dimension of the feature map class, which is exponentially large for simple neural network classes (Dong et al., 2021). In contrast, our sample complexity bound depends only on the neural network classes only through its capacity.

Our work is motivated by the complexity analysis of neural network classes. Dong et al. (2021) show that the Eluder dimension of one-layer neural network classes is exponentially large, suggesting that the previous analysis of RL algorithms based on the Eluder dimension (Russo & Van Roy, 2013) may not be applicable to neural networks.

## 1.2 NOTATION

For a vector $v \in \mathbb{R}^d$, we define $\|v\|_2 = (\sum_{i=1}^d v_i^2)^{1/2}$, where $v_i$ is the $i$-th element of $v$. For a real-valued function $f : \mathcal{X} \to \mathbb{R}$, we define $\|f\|_\infty = \max_{x \in \mathcal{X}} |f(x)|$. For a vector-valued function $f : \mathcal{X} \to \mathbb{R}^d$, we define $\|f\|_{\infty,2} = \max_{x \in \mathcal{X}} \|f(x)\|_2$. For a sequence of real-valued functions $r = \{r_h\}_{h=1}^H \subset \mathcal{X} \to \mathbb{R}$, we define $\|r\|_\infty = \sup_{h \in [H], x \in \mathcal{X}} |r_h(x)|$. We denote by $\mathcal{N}(\mathcal{F}, \epsilon, \|\cdot\|)$ the $\epsilon$-covering number of the function class $\mathcal{F}$ with respect to the norm $\|\cdot\|$, define $H_\infty(\mathcal{F}, \epsilon) = \log \mathcal{N}(\mathcal{F}, \epsilon, \|\cdot\|_\infty)$ for a real-valued function class $\mathcal{F}$, and define $H_2(\mathcal{F}, \epsilon) = \log \mathcal{N}(\mathcal{F}, \epsilon, \|\cdot\|_{\infty,2})$ for a vector-valued function class $\mathcal{F}$. We further define $[n] = \{1, \ldots, n\}$ when $n$ is an integer. For a set $\mathcal{C}$, we denote by $\Delta(\mathcal{C})$ the set of the distributions over $\mathcal{C}$, and $\mathcal{U}(\mathcal{C})$ the uniform distribution over $\mathcal{C}$. For $g : \mathcal{X} \to \mathbb{R}$ and $\mathcal{X}_n = \{x_1, \ldots, x_n\} \subset \mathcal{X}$, we define $g[\mathcal{X}_n] = (g(x_1), \ldots, g(x_n))^\top$.

## 2 PRELIMINARY

We consider an episodic MDP $\mathcal{V}^* = (\mathcal{S}, \mathcal{A}, H, \mathcal{P}^*, r^*)$ with a state space $\mathcal{S} \in \mathbb{R}^d$, an action space $\mathcal{A}$, a horizon $H$, transition kernels $\mathcal{P}^* = \{\mathcal{P}_h^*\}_{h=1}^H$, and reward functions $r^* = \{r_h^*\}_{h=1}^H$. We assume that the reward functions are bounded and deterministic, that is, $\|r_h^*\|_\infty \in [0, 1]$ for all $h \in [H]$. We also assume that the action space is finite, that is, $|\mathcal{A}| < \infty$. The agent iteratively interacts with the environment as follows. At the beginning of each episode, the agent determines a policy $\pi = \{\pi_h\}_{h=1}^H$, where $\pi_h : \mathcal{S} \to \Delta(\mathcal{A})$ for any $h \in [H]$. Without loss of generality, we assume that the initial state is fixed to $s_{\text{init}} \in \mathcal{S}$ across all episodes. At the $h$-th step, the agent receives a state $s_h$ and takes an action $a_h$ following $a_h \sim \pi_h(\cdot \mid s_h)$. Subsequently, the agent receives a reward $r_h^*(s_h, a_h)$ and the next state following $s_{h+1} \sim \mathcal{P}_{h+1}^*(\cdot \mid s_h, a_h)$. The episode ends after the agent receives the last state $s_{H+1}$. For a given policy $\pi = \{\pi_h\}_{h=1}^H$, where $\pi_h : \mathcal{S} \to \Delta(\mathcal{A})$ for any $h \in [H]$, we define the value function $V_h^\pi$ and the $Q$-function $Q_h^\pi$ for any $h \in [H]$ as

$$
\begin{aligned}
V_h^\pi(s; r^*, \mathcal{P}^*) &= \mathbb{E}_{\pi, \mathcal{P}^*} \Big[ \sum_{i=h}^H r_i^*(s_i, a_i) \,\Big|\, s_h = s \Big], \\
Q_h^\pi(s, a; r^*, \mathcal{P}^*) &= \mathbb{E}_{\pi, \mathcal{P}^*} \Big[ \sum_{i=h}^H r_i^*(s_i, a_i) \,\Big|\, s_h = s, a_h = a \Big].
\end{aligned}
\tag{2.1}
$$

Here the expectation $\mathbb{E}_{\pi, \mathcal{P}^*}[\cdot]$ in (2.1) is taken with respect to $s_{i+1} \sim \mathcal{P}_i^*(\cdot \mid s_i, a_i)$ and $a_i \sim \pi_i(\cdot \mid s_i)$ for $i \in \{h, h+1, \ldots, H\}$. For convenience, we define $V_{H+1}^\pi(s; r, \mathcal{P}) = 0$ for any state $s \in \mathcal{S}$, reward function $r$, transition kernel $\mathcal{P}$ and policy $\pi$. For simplicity, we define the expected total reward $J(\pi; r^*, \mathcal{P}^*)$ as $J(\pi; r^*, \mathcal{P}^*) = V_1^\pi(s_{\text{init}}; r^*, \mathcal{P}^*)$. The goal of RL is to find a policy $\pi^*$ that maximizes the expected total reward. Specifically, for the episodic MDP $\mathcal{V}^* = (\mathcal{S}, \mathcal{A}, H, \mathcal{P}^*, r^*)$, We define $\pi^* \in \arg\max_\pi J(\pi; r^*, \mathcal{P}^*)$ as an optimal policy. Correspondingly, we define the optimal $Q$-function $Q_h^*$ and the optimal value function $V_h^*$ as $Q_h^*(s, a; r^*, \mathcal{P}^*) = Q_h^{\pi^*}(s, a; r^*, \mathcal{P}^*)$ and $V_h^*(s; r^*, \mathcal{P}^*) = V_h^{\pi^*}(s; r^*, \mathcal{P}^*)$ for any $(s, a) \in \mathcal{S} \times \mathcal{A}$.

## 3 MARKOV DECISION PROCESSES WITH NEURAL DYNAMICS

In this paper, our goal is to develop a provably efficient algorithm for RL problems adapted with large feature space, such as neural networks (NNs). To this end, we introduce the MDPs with neural dynamics, whose reward and transition dynamics are parameterized by NNs.

**Motivation.** Our definition is motivated by the kernelized nonlinear regulator (KNR). In a KNR model, the transition kernel takes the following form,

$$s_{h+1} = W_h^* \phi_h^*(s_h, a_h) + \epsilon, \qquad \epsilon \sim \mathcal{N}(0, I_d), \tag{3.1}$$

where $\phi_h^*$ is a known nonlinear feature map. Former research proposes sample-efficient algorithms for such a model. Although such a KNR setting empowers sample efficient RL (Kakade et al. (2020)), it is relatively restrictive in the following two aspects. First, the feature map $\phi_h^*$ and the expected reward $r_h^*$ are known a priori. Second, the model only imposes nonlinearity on $(s_h, a_h)$ via the known feature map, while the conditional expectation of the next state given $s_h, a_h$ is a linear function of $\phi_h^*(s_h, a_h)$. In other words, when $\phi_h^*$ is known, the transition dynamics can be recovered via linear system identification methods such as ridge regression.

To generalize the KNR model, we interpret (3.1) as an energy-based model. More specifically, we can write the transition of the MDP in (3.1) as

$$\mathcal{P}_h^*(s_{h+1} \mid s_h, a_h) \propto \exp\big(-E(s_{h+1}, s_h, a_h)\big), \tag{3.2}$$

where the energy function $E(s_{h+1}, s_h, a_h)$ is defined as

$$E(s_{h+1}, s_h, a_h) = \|s_{h+1} - W_h^* \phi_h^*(s_h, a_h)\|_2^2 / 2. \tag{3.3}$$

Here (3.2) omits a normalization factor, which is a function of $(s_h, a_h)$. We generalize this model and impose nonlinearity on the next state $s_{h+1}$ by substituting a nonlinear feature map $\psi_{h+1}^*(s_{h+1})$ for $s_{h+1}$ in (3.3). Such a generalization allows us to incorporate the nonlinearity of the next state in the model. In addition, we assume that the nonlinear feature maps $\phi_h^*$ and $\psi_{h+1}^*$ are unknown and need to be estimated from pre-specified feature classes $\Phi$ and $\Psi$, which for example, can be two classes of NNs. We further assume that the expected reward $r^*$ is unknown and needs to be estimated from the reward function class $\mathcal{R}$. We formalize our generalization in the following definition.

**Definition 3.1** (MDPs with Neural Dynamics). *An episodic MDP $(\mathcal{S}, \mathcal{A}, H, \mathcal{P}^*, r^*)$ is an MDP with neural dynamics if its reward function $r^* = \{r_h^*\}_{h=1}^H$ belongs to a reward function class $\mathcal{R}$, which is a known function class that consists of NNs, and the transition kernel of the MDP $\mathcal{P}^* = \{\mathcal{P}_h^*\}_{h=1}^H$ takes the following form,*

$$\mathcal{P}_h^*(s_{h+1} \mid s_h, a_h) \propto \exp\Big(-\big\|\psi_{h+1}^*(s_{h+1}) - \phi_h^*(s_h, a_h)\big\|_2^2 / 2\Big). \tag{3.4}$$

*Here $\phi_h^* \in \Phi : \mathbb{R}^d \times \mathcal{A} \to \mathbb{R}^m$ and $\psi_{h+1}^* \in \Psi : \mathbb{R}^d \to \mathbb{R}^m$ are two unknown feature maps, and $\Phi$, $\Psi$ are two known feature map classes that consist of NNs. We denote by $\mathcal{M}$ the set of all the transition kernels that take the form of (3.4), and let $\mathcal{X} \in \mathbb{R}^m$ denote the union of the image spaces of the feature maps, namely,*

$$\mathcal{X} = \{\varphi(s, a) \mid (\varphi, s, a) \in \Phi \times \mathcal{S} \times \mathcal{A}\} \cup \{\varphi(s) \mid (\varphi, s) \in \Psi \times \mathcal{S}\} \subseteq \mathbb{R}^m.$$

**Generality of Definition 3.1.** We remark that Definition 3.1 is a significant generalization of stochastic nonlinear systems beyond KNR. For instance, when $\psi_{h+1}^*(s_{h+1}) = s_{h+1}$, the transition kernel in Definition 3.1 takes the following form,

$$\mathcal{P}_h^*(s_{h+1} \mid s_h, a_h) \propto \exp\Big(-\big\|s_{h+1} - \phi_h^*(s_h, a_h)\big\|_2^2 / 2\Big),$$

which is the transition kernel in Ren et al. (2021). Therefore, we recover the model in Ren et al. (2021) when $\psi_{h+1}^*$ is known to be the identity map and the reward function is known. Moreover, the transition kernel defined in (3.4) also includes a class of nonlinear dynamics satisfying

$$s_{h+1} = (\psi_{h+1}^*)^{-1}\big(\phi_h^*(s_h, a_h) + \epsilon_h\big),$$

where $\mathcal{S} \subseteq \mathbb{R}^m$, $\psi_{h+1}^* : \mathbb{R}^m \to \mathbb{R}^m$, the determinant of the Jacobian matrix of $\psi_{h+1}^*$ is a constant, and $\epsilon_h$ is a Gaussian noise. Our model significantly generalizes such a model by allowing a possibly noninvertible feature map $\psi_{h+1}^*$.

**Relationship with Kernelized Linear MDP.** Recall that $K(x, y) = \exp(-\|x - y\|_2^2 / 2)$ is also known as the Gaussian RBF kernel, which induces a reproducing kernel Hilbert space (RKHS) defined on $\mathbb{R}^m$ (Rahimi et al., 2007). (See Appendix §D for a brief introduction of RKHS.) Intuitively, $K(x, y)$ measures the proximity between $x$ and $y$ in the kernel space. From this perspective, the transition

kernel in (3.4) specifies the next state $s_{h+1}$ by measuring the proximity of the representations $\phi_h^*(s_h, a_h)$ and $\psi_{h+1}^*(s_{h+1})$. Besides, since $K(x, y)$ can be written as $\langle k(x), k(y) \rangle_{\mathcal{H}}$, where the feature map of the RKHS $k$ is defined as $k(x) = K(x, \cdot)$, and $\langle \cdot, \cdot \rangle_{\mathcal{H}}$ is the inner product of the RKHS respectively. Thus, (3.4) can be equivalently written as

$$\mathcal{P}_h^*(s_{h+1} \mid s_h, a_h) = \left\langle Z_h^*(s_h, a_h) \cdot k\big(\phi_h^*(s_h, a_h)\big), k\big(\psi_{h+1}^*(s_{h+1})\big) \right\rangle_{\mathcal{H}}, \tag{3.5}$$

where $Z_h^*(s_h, a_h)$ is the normalization factor in (3.4). Thus, when $Z_h^*$ is known, our model can be regarded as an RKHS extension of the linear MDP model (Jin et al., 2020). This is the case when $\psi^*$ is the identity maps but unknown to the learner, and $Z_h^*(s_h, a_h)$ becomes a constant (Ren et al., 2021). **See Appendix §D for more details of the relationship between the model in** (3.4) **and RKHS.**

**Role of NNs in Our Model.** We would like to remark that the model specified in Definition 3.1 is not restricted to NNs. In fact, the definition only requires proper function classes of the reward function, representations of $(s_h, a_h)$ and $s_{h+1}$, namely, $\mathcal{R}$, $\Phi$, and $\Psi$. Thus, our model can also be defined for other function approximators such as polynomial spline (Unser et al., 1993), classification and regression tree (Syrgkanis & Zampetakis, 2020). Meanwhile, as we will see in the sequel, both our algorithm and the theoretical results do not hinge on NNs and can employ general function classes with bounded capacity. Here we call our model neural dynamics in order to **highlight that our work is the first one that covers arbitrary NN classes with bounded log-covering numbers**.

## 4 ALGORITHM

In this section, we introduce an algorithm for solving MDPs with neural dynamics in the online setting. We first introduce the motivation of the algorithm, and then introduce the procedure in detail.

**Motivation.** To strike a balance between exploration and exploitation, our algorithm follows the principle of *Optimism in the Face of Uncertainty* (Lattimore & Szepesvári, 2020). When we know the true feature maps $\{\phi_h^*\}_{h=1}^H$, we can apply kernel LSVI (Yang et al., 2020) to construct the exploration bonus since the energy-base transition admits a kernel structure. (See §3 for the details.) However, in MDPs with neural dynamics, we do not know $\{\phi_h^*\}_{h=1}^H$. A straightforward solution for handling the unknown feature maps is to learn the feature maps from the data we collect and construct the bonus based on the learned features. However, the bonus constructed by the learned features might be invalid since the learned features have errors. We handle the error in the learned feature by purposefully taking uniform actions when exploring the environment. Such a sampling scheme gives us more diverse data for model estimation. Based on this motivation, we design an iterative algorithm that outputs a policy after $N$ iterations. In particular, in each iteration $n \in [N]$, our algorithm performs the following four steps: (i) sampling new data from the environment, (ii) estimating the model via maximum likelihood estimation, (iii) constructing exploration incentives using the features of the learned model, and (iv) updating the online policy for exploration via planning on the learned model.

**Sampling Scheme.** As we mentioned in §3, the transition of MDPs with neural dynamics can be written as an energy-based model and admits the Gaussian RBF kernel. To exploit the kernel structure in the transition, we explore the environment using the exploration bonus induced by the Gaussian RBF kernel and the feature maps learned from the data, which is motivated by Yang et al. (2020). However, since the bonus is not induced by the true underlying feature, it might fail to indicate the most uncertain state-action pairs for exploration. To mitigate such an issue, we combine the uniform policy, which samples action from the uniform distribution over the action space, with the optimistic policy during the sampling procedure. Intuitively, such a sampling scheme provides a wider coverage over the state-action space and better explores the environment.

To simplify the presentation in the main text, we present the sketch of the sampling scheme as follows. The rigorous presentation of the sampling scheme for the boundary case is deferred to Appendix §B. In the $n$-th iteration of our algorithm, given the previously collected dataset $D_{h,i}^{n-1}$ for $i \in \{0, 1, 2\}$ and $h \in [H]$, we interact with the MDP following the policy $\pi^n = \{\pi_h^n\}_{h=1}^H$ and obtain the new dataset $D_{h,i}^n$ for $i \in \{0, 1, 2\}$ and $h \in [H]$. Specifically, for any $h \in \{-1, \dots, H\}$, we start from the initial state $s_1$ and choose the action $a_{\bar{h}} \sim \pi_{\bar{h}}^n(\cdot \mid s_{\bar{h}})$ in the $\bar{h}$-th step when $\bar{h} \in \{1, \cdots, h\}$, and choose the action $a_{\bar{h}} \sim \mathcal{U}(\mathcal{A})$ when $\bar{h} \in \{h+1, h+2\}$. Here $\mathcal{U}(\mathcal{A})$ is the uniform distribution over the action space $\mathcal{A}$. By following such a procedure, we obtain the following trajectory,

$$s_1, a_1, r_1, \dots, s_{h+2}, a_{h+2}, r_{h+2}, s_{h+3}, \tag{4.1}$$

where $s_1 = s_{\text{init}}$. Then, we label the obtained trajectory as follows,

$$s_{h+i,i}^n = s_{h+i}, \qquad a_{h+i,i}^n = a_{h+i}, \qquad r_{h+i,i}^n = r_{h+i}, \qquad \bar{s}_{h+i,i}^n = s_{h+i+1}, \qquad (4.2)$$

for any $i \in \{0,1,2\}$. We then update the dataset $\mathcal{D}_{h,i}^n$ by $\mathcal{D}_{h,i}^n = \mathcal{D}_{h,i}^{n-1} \cup \{(s_{h,i}^n, a_{h,i}^n, r_{h,i}^n, \bar{s}_{h+i,i}^n)\}$ for any $i \in \{0,1,2\}$ and $h \in [H]$. We use $i$ in (4.2) to indicate how many steps of the uniform policy we need to execute to obtain such a dataset. Intuitively, the dataset with a bigger $i$ has a better coverage over the state-action space $\mathcal{S} \times \mathcal{A}$. We summarize the sampling scheme in Algorithm 1. See Figure 1 in Appendix §B for an illustration of the sampling scheme, and see Algorithm 3 in Appendix §B for the formal presentation of Algorithm 1.

---

**Algorithm 1** Sampling Scheme (Informal)

---

1: **Input:** Policy $\pi^n = \{\pi_h^n\}_{h=1}^H$, datasets $\mathcal{D}_{h,i}^{n-1}$ for $i \in \{0,1,2\}$ and $h \in [H]$.
2: **for** $h = -1, \ldots, H$ **do**
3:      Interact with the environment to obtain the trajectory in (4.1) by first executing $\pi^n$ from $s_1$ to $s_h$, and then execute $\mathcal{U}(\mathcal{A})$ for two more steps. Label the obtained trajectory as (4.2).
4: **end for**
5: Set $\mathcal{D}_{h,i}^n \leftarrow \{(s_{h,i}^\tau, a_{h,i}^\tau, r_{h,i}^\tau, \bar{s}_{h+i,i}^\tau)\}_{\tau=1}^n$ for $h \in [H]$ and $i \in \{0,1,2\}$, where the tuple $(s_{h,i}^\tau, a_{h,i}^\tau, r_{h,i}^\tau, \bar{s}_{h+i,i}^\tau)$ is the data we label in (4.2).          ▷ Updating the datasets
6: **Return:** Datasets $\{\mathcal{D}_{h,i}^n\}_{h=1,i=0}^{h=H,i=2}$.

---

**Model Estimation.** To estimate the model, we solve the following optimization problems,

$$\hat{r}_h^n = \underset{r \in \mathcal{R}}{\operatorname{argmin}} \sum_{i=1}^2 \sum_{(s_h,a_h,r_h,s_{h+1}) \in \mathcal{D}_{h,i}^n} \big[ r_h - r(s_h, a_h) \big]^2, \qquad (4.3)$$

$$\hat{\mathcal{P}}_h^n = \underset{\mathcal{P} \in \mathcal{M}}{\operatorname{argmin}} - \sum_{i=1}^2 \sum_{(s_h,a_h,r_h,s_{h+1}) \in \mathcal{D}_{h,i}^n} \log \mathcal{P}(s_{h+1} \mid s_h, a_h). \qquad (4.4)$$

Here $\mathcal{R}$ and $\mathcal{M}$ are the reward function class and the transition kernel class defined in Definition 3.1. We denote by $\phi_h^n$ and $\psi_{h+1}^n$ the feature maps that correspond to the transition kernel $\hat{\mathcal{P}}_h^n$ estimated in (4.4). To simplify our analysis, we assume that there exists an oracle that returns the global minimum of the optimization problems (4.3) and (4.4). Similar assumption also arises in the previous study of RL (Fan et al., 2020; Kakade et al., 2020; Uehara et al., 2021; Jin et al., 2021a). When the normalization factor in (3.5) is a constant, (4.4) can be easily implemented since it is equivalent with

$$\hat{\mathcal{P}}_h^n = \underset{\phi \in \Phi, \psi \in \Psi}{\operatorname{argmin}} \sum_{i=1}^2 \sum_{(s_h,a_h,r_h,s_{h+1}) \in \mathcal{D}_{h,i}^n} \| \psi(s_{h+1}) - \phi(s_h, a_h) \|_2^2.$$

**Remark 4.1** (Transition Estimation). *We would like to remark that the method for estimating the transition is not restricted to maximum likelihood estimation (MLE). Methods including variational autoencoder (Kingma & Welling, 2013), score matching (Hyvärinen & Dayan, 2005) can also be used for transition estimation. Our sample complexity bound holds for any transition estimator whose total variance error has an upper bound.*

**Exploration Bonus.** The transition kernel in Definition 3.1 is closely related to the radial basis function (RBF) kernel. In the sequel, we define the bonuses for exploration and update the policy based on such bonuses. Specifically, for a fixed feature map $\phi : \mathcal{S} \to \mathbb{R}$, we define the Gram matrix $K_h^n[\phi]$ and the function $k_h^n[\phi] : \mathcal{S} \times \mathcal{A} \to \mathbb{R}^n$ as follows,

$$K_h^n[\phi] = \Big[ K\big(\phi(s_{h,1}^\tau, a_{h,1}^\tau), \phi(s_{h,1}^{\tau'}, a_{h,1}^{\tau'})\big) \Big]_{\tau,\tau'=1}^n \in \mathbb{R}^{n \times n},$$

$$k_h^n[\phi](s,a) = \Big[ K\big(\phi(s,a), \phi(s_{h,1}^1, a_{h,1}^1)\big), \ldots, K\big(\phi(s,a), \phi(s_{h,1}^n, a_{h,1}^n)\big) \Big]^\top \in \mathbb{R}^n, \ \forall (s,a) \in \mathcal{S} \times \mathcal{A},$$

where $\{(s_{h,1}^\tau, a_{h,1}^\tau, r_{h,1}^\tau, \bar{s}_{h,1}^\tau)\}_{\tau=1}^n \in \mathcal{D}_{h,1}^n$. We then define the bonus $u_h^n$ as follows,

$$u_h^n(s,a) = \min\{2H + 2, \beta \tilde{u}_h^n(s,a)/\lambda\},$$
$$\text{where} \quad \tilde{u}_h^n(s,a) = 1 - k_h^n[\phi_h^n](s,a)^\top \big(\lambda I + K_h^n[\phi_h^n]\big)^{-1} k_h^n[\phi_h^n](s,a). \qquad (4.5)$$

Here $\beta > 0$ and $\lambda > 0$ are the tuning parameters. We remark that the form of the bonus in (4.5) aligns with the bonus in other previous works that use kernel functions for function approximation (Srinivas et al., 2009; Yang et al., 2020).

**Remark 4.2** (Dependency of Rewards on Features.). *Here we do not require that the reward depends on the feature in the transition kernel, which is different from the literature in linear MDPs (Cai et al., 2019; Jin et al., 2020). Common sense seems to dictate that we can not characterize the uncertainty using the feature without such a dependency. However, the estimation error of the reward estimators $\widehat{r}_h^n$ in the empirical measure can be bounded from the above by the property of the least square estimator. Therefore, the uncertainty of $r_h^*$ in the distribution induced by any new policy, such as $\pi^*$, can be bounded from the above by the distribution shift, which is characterized by $u_{h-1}^n$. Such an observation allows us to characterize the uncertainty of the reward estimator even when the reward does not depend on the feature in the transition kernel.*

**Policy Update.** We update the policy $\pi^{n+1}$ by setting it as the optimal policy of the learned model, which can be efficiently computed by dynamic programming. Due to the space limit, we defer the details of the planning algorithm to Appendix §B. We remark that we can also apply other model-based algorithms, including *Dyna* (Sutton, 1991) and *Gradient-Aware Model-based Policy Search* (D'Oro et al., 2020), to compute the optimal policy of the learned model, and the suboptimality of the output of Algorithm 2 is bounded when the error of the planning oracle is bounded.

**Remark 4.3** (Computational Efficiency). *Our algorithm is oracle efficient in the sense that our algorithm is computationally efficient given an optimization oracle for model estimation, which also appears in the previous study (Fan et al., 2020; Kakade et al., 2020; Uehara et al., 2021). More specifically, the bonus and the policy in each iteration can be efficiently computed by (4.5) and Algorithm 4 in the appendix. The existing literature on general function approximation requires either global optimism over the confidence set (Kakade et al., 2020; Jin et al., 2021a) or posterior sampling over the hypothesis set (Ren et al., 2021), which can not be computed efficiently.*

---

**Algorithm 2** Exploration with Learnable Neural Features (ELNF)

1: **Input:** Failure probability $\delta > 0$, tuning parameters $\beta, \lambda > 0$.
2: Initialize: Set $\pi^1 = \{\pi_h^1\}_{h=-1}^H$, where $\pi_h^1(\cdot \mid s) = \mathcal{U}(\mathcal{A})$ for any $(s, h) \in \mathcal{S} \times [H]$, and set $\mathcal{D}_{h,i}^0 = \emptyset$ for all $(h, i) \in [H] \times \{0, 1, 2\}$.
3: **for** $n = 1, \ldots, N$ **do**
4:     Set $\{\mathcal{D}_{h,i}^n\}_{h=1,i=0}^{h=H,i=2}$ by applying Algorithm 1 (Sampling Scheme) with the policy $\pi^n$ and the datasets $\{\mathcal{D}_{h,i}^{n-1}\}_{h=1,i=0}^{h=H,i=2}$ as the input.     ▷ Sampling
5:     Set $\{\widehat{r}_h^n\}_{h=1}^H$ and $\{\widehat{\mathcal{P}}_h^n\}_{h=1}^H$ as in (4.3) and (4.4), respectively.     ▷ Model estimation
6:     Set $\{\phi_h^n\}_{h=1}^H$ and $\{\psi_{h+1}^n\}_{h=1}^H$ as the feature maps corresponding to $\{\widehat{\mathcal{P}}_h^n\}_{h=1}^H$.
7:     Set $\{u_h^n\}_{h=1}^H$ as in (4.5).     ▷ Feature estimation and bonus construction
8:     Set $\pi^{n+1}$ by applying Algorithm 4 (Planning Algorithm) in the appendix with the learned model $\{\widehat{r}_h^n\}_{h=1}^H$, $\{\widehat{\mathcal{P}}_h^n\}_{h=1}^H$, and the bonuses $\{\widehat{u}_h^n\}_{h=1}^H$ as the input.     ▷ Planning
9: **end for**
10: **Return:** $\widehat{\pi} = \mathcal{U}\big(\{\pi^n\}_{n=2}^{N+1}\big)$.

---

## 5   Theory

In this section, we present the analysis of ELNF. We first present the boundedness assumption on the model.

**Assumption 5.1** (Boundedness of Model). *We assume that the state space $\mathcal{S}$ is a bounded set of $\mathbb{R}^d$, and the Lebesgue measure of $\mathcal{S}$ is an absolute constant. We also assume that $\max\{\|\phi(s, a)\|_2, \|\psi(s)\|_2\} \leq R$ for all $(s, a, \phi, \psi) \in \mathcal{S} \times \mathcal{A} \times \Phi \times \Psi$. We further assume that $0 \leq r(s, a) \leq 1$ for any $(s, a, r) \in \mathcal{S} \times \mathcal{A} \times \mathcal{R}$.*

Since $\mathcal{S}$ is bounded, Assumption 5.1 is a reasonable regularity condition on the model. Such a regularity assumption is standard and is also assumed in the previous works (Cai et al., 2019; Jin et al., 2020; 2021b). Next, we introduce the following assumption, which characterizes the complexity of the NN classes.

**Assumption 5.2** (Decay Rate of Covering Number). *There exists constants $C_{\mathrm{net}} > 0$ and $\gamma \geq 0$ that only depend on $(\mathcal{R}, \Phi, \Psi)$, such that*

$$H_c(\epsilon) \triangleq \max\{H_\infty(\mathcal{R}, \epsilon), H_2(\Phi, \epsilon), H_2(\Psi, \epsilon)\} \leq C_{\mathrm{net}} \cdot \left(1 + \log(1/\epsilon)\right)/\epsilon^\gamma.$$

In Assumption 5.2, $\gamma$ characterizes the complexity of the NN class by quantifying the growth rate of the covering number when the covering radius $\epsilon$ decays. We remark that previous research bounds the covering number of NN classes from the above at the same scale as Assumption 5.2. For example, Schmidt-Hieber (2020) and Chen et al. (2019) show that NN classes with specific structures satisfy Assumption 5.2 with $\gamma = 0$. See Lemmas C.3 and C.6 in Appendix §C for the details.

**Theorem 5.3** (Sample Complexity of ELNF). *We assume that Assumption 5.2 holds with $\gamma < 1/2$, and we can obtain the exact solution to the optimization problems* (4.3) *and* (4.4). *We set the tuning parameters $\lambda$ and $\beta$ as*

$$\lambda = C' N^{\gamma/(1+\gamma)} m \log(48HRN/\delta),$$
$$\beta = C'' H |\mathcal{A}|^{1/2} m^{1/2} N^{3\gamma/(4+4\gamma)} \sqrt{\log(48HRN/\delta)}$$

*in* ELNF, *where $m$ is the dimension of the image of the feature maps, $C'$, $C''$ are constants that only depend on the regularity parameters in Assumption 5.1. Under Assumption 5.1 and 5.2, for the policy $\widehat{\pi}$ returned by* ELNF, *it holds with probability at least $1 - \delta$ that*

$$J(\pi^*; r^*, \mathcal{P}^*) - J(\widehat{\pi}; r^*, \mathcal{P}^*) \leq CH^5 \cdot |\mathcal{A}|^2 \cdot \xi \cdot N^{(2\gamma-1)/(2+2\gamma)} (\log N)^{m+1}.$$

*Here $C$ is a constant that only depends on the dimension $m$, the bound of the feature maps $R$, and $C_{\mathrm{net}}$ in Assumption 5.2, and $\xi = (\log(48HRN/\delta))^{5/2}$ is a logarithmic factor.*

*Proof.* See Appendix §E for a detailed proof. $\square$

In Theorem 5.3, $\lambda$ is the regularization parameter that trades off between bias and variance, and $\beta$ is the uncertainty coefficient, which scales with $\gamma$ and $N$. We remark that our analysis is not restricted to NN classes, and can be extended to other bounded function classes with bounded covering numbers. We further remark that $m$ in Theorem 5.3 is the dimension of the image of the feature maps, which can be much smaller than the dimension of the state.

Moreover, in Appendix §E, we show that the suboptimality bound in Theorem 5.3 reduces to $\widetilde{\mathcal{O}}(d_{\mathrm{eff}} N^{(2\gamma-1)/(2+2\gamma)})$ in terms of $N$, where $d_{\mathrm{eff}}$ is the effective dimension in Definition E.4 in the appendix. Such a bound connects the sample efficiency of ELNF to the effective dimension of the RKHS and the covering number of NN classes. We further remark that when $\gamma = 0$ in Theorem 5.3, the suboptimality bound is sublinear in $N$, which aligns with the previous theoretical research. A detailed comparison with the related work can be found in Appendix §A.

## 6   PROOF SKETCH OF THEOREM 5.3

In this section, we sketch the proof of Theorem 5.3 and highlight the technique that allows us to remove the dependency on the Eluder dimension of NN classes. Since the policy $\widehat{\pi}$ returned by Algorithm 2 (ELNF) is the mixture of $\pi^1, \ldots, \pi^N$, we can decompose the suboptimality as

$$J(\pi^*; r^*, \mathcal{P}^*) - J(\widehat{\pi}; r^*, \mathcal{P}^*) = \underbrace{\frac{1}{N} \sum_{n=1}^{N} \left[ J(\pi^*; r^*, \mathcal{P}^*) - J(\pi^*; r^n + u^n, \mathcal{P}^n) \right]}_{\text{Term (a)}}$$

$$+ \underbrace{\frac{1}{N} \sum_{n=1}^{N} \left[ J(\pi^*; r^n + u^n, \mathcal{P}^n) - J(\pi^{n+1}; r^n + u^n, \mathcal{P}^n) \right]}_{\text{Term (b)}}$$

$$+ \underbrace{\frac{1}{N} \sum_{n=1}^{N} \left[ J(\pi^{n+1}; r^n + u^n, \mathcal{P}^n) - J(\pi^{n+1}; r^*, \mathcal{P}^*) \right]}_{\text{Term (c)}}.$$

Here Term (a) is the out-of-sample estimation error of the estimated value $J$, Term (b) is the error of the planning algorithm, and Term (c) is the in-sample estimation error of the estimated value $J$. Since the bonus $u^n$ captures the uncertainty of the estimated model with high probability, and the planning algorithm finds the optimal policy of the learned model, Term (a) and Term (b) are small with high probability, which is shown in the following lemma.

**Lemma 6.1** (Informal). *We define $\zeta = \sqrt{C_2 \log(20HRN/\delta)N^{\gamma/(1+\gamma)}}$, where $C_2$ is a constant that depends on the regularity parameters and the parameters of the NN class. Following the same condition of Theorem 5.3, we have $\mathrm{Term(a)} \leq 8H|\mathcal{A}|\zeta\sqrt{N}$ holds with probability at least $1 - \delta$. We also have $\mathrm{Term(b)} \leq 0$ when we use Algorithm 4 for planning.*

*Proof.* This can be directly proved by Lemma E.5 and Lemma E.6. See Appendix §E for the details. □

The remaining analysis is to connect Term (c) with the complexity measure of the model class. Former research (Wang et al., 2020; Jin et al., 2021a) that allows neural network parameterization quantifies the uncertainty by the level set, and the Eluder dimension naturally appears when we telescope the in-sample error. To remove the dependency on the Eluder dimension, we want to quantify the uncertainty by the bonus defined by the true feature. However, since the true feature is unknown, we can only quantify the uncertainty using the learned feature. Inspired by Uehara et al. (2021), we obtain the following lemma, which connects the bonus defined by the learned feature with the bonus defined by the true unseen feature.

**Lemma 6.2** (Bonus Equivalence for the True Model, Informal). *Following the same condition with Theorem 5.3, we have*

$$J(\pi; u^n, \mathcal{P}^*) \leq 2\,|\mathcal{A}|\,\beta d_{\mathrm{eff}}^{1/2}/\sqrt{n} + J(\pi; u^{*,n}, \mathcal{P}^*)$$

*holds for any policy $\pi$ and $n \geq 2$ with high probability, where the bonus defined by the true feature $u^{*,n} = \{u_h^{*,n}\}_{h=1}^H$ is defined in Lemma I.1.*

*Proof.* This is Lemma I.2. See Appendix §I.5 for the details. □

We remark that we cannot obtain the above lemma by directly applying the technique in Uehara et al. (2021) since they only consider the finite-dimensional inner product. Using the lemma above, we have $\mathrm{Term(c)} \leq 32H^2\,|\mathcal{A}|\,\zeta\sqrt{d_{\mathrm{eff}}N} + 2\sum_{n=1}^N J(\pi^{n+1}; u^{*,n}, \mathcal{P}^*)$ holds with high probability. Finally, we show that the sum of the bonuses defined by the true feature can be bounded by the effective dimension of the RKHS induced by the noise instead of the Eluder dimension of NN classes. We conclude the proof of Theorem 5.3 by combining the upper bounds of Term (a), Term (b), and Term (c).

**Removing Dependency on Eluder Dimension.** The existing literature on RL using general function approximators relies on the Eluder dimension when bounding the regret or the suboptimality (Wang et al., 2020; Jin et al., 2021a; Ren et al., 2021), which is exponentially large for a simple neural network class (Dong et al., 2021). However, we can remove such dependency in MDPs with neural dynamics. Our key insight is that we can regard MDPs with neural dynamics as kernel MDPs (Yang et al., 2020) whose feature is the composite map of the neural network and the feature map of the RKHS since the energy-base transition admits a kernel structure, which is shown in §3. Therefore, we can characterize the effect of the distribution shifts by the bonus defined by the true feature, whose sum is bounded by the effective dimension of the RKHS instead of the Eluder dimension of the NN class, without knowing the true feature.

**Role of Uniform Policy in Proof.** The uniform policy in the sampling scheme enables us to bound the influence of the distribution shift and show that the bonuses defined by the learned feature are valid uncertainty quantification. In order to show that the estimated value defined by the learned bonus is almost optimistic (Lemma E.5) for any policy, we need to bound the influence of the distribution shift. The data that we obtain from the uniform sampling has better coverage over $\mathcal{S} \times \mathcal{A}$, and the bonuses we construct with these data quantify the uncertainty with the presence of the distribution shift. Therefore, we take uniform actions when exploring the environment to obtain a valid uncertainty quantification without knowing the true feature, and $|\mathcal{A}|^2$ in the suboptimality bound is the price paid for the uniform sampling.

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

LIST OF NOTATION

In the sequel, we present a list of notations in the paper.

| Notation | Explanation |
|---|---|
| $\mathcal{S}, \mathcal{A}, \mathcal{X}$ | The state, the action, and the feature spaces, respectively. |
| $H, N$ | The length of an episode, and the total number of iterations of Algorithm 2 (ELNF), respectively. |
| $n, h$ | The index that iterates from $1$ to $N$, and the index that iterates from $1$ to $H$, respectively. |
| $\tau, \bar{h}$ | The index that iterates from $1$ to $n$,, and the index that iterates from $1$ to $h$, respectively. |
| $\Phi, \Psi$ | The feature map classes of the current state-action pair and the next state, which parameterized by NNs. |
| $\mathcal{R}$ | The reward function class parameterized by NNs. |
| $\{\mathcal{P}_h^*\}_{h=1}^H, \{r_h^*\}_{h=1}^H$ | The transition kernel and the reward of the MDP. |
| $d, m$ | The dimension of the state space and the image space of the feature maps, respectively. |
| $\{\mathcal{D}_{h,i}^n\}_{h=1,i=0}^{h=H,i=2}, \pi^n$ | The dataset and the policy in the $n$-th iteration of Algorithm 2 (ELNF), respectively. |
| $\{\mathcal{P}_h^n\}_{h=1}^H, \quad \{\widehat{r}_h^n\}_{h=1}^H, \{u_h^n\}_{h=1}^H$ | The estimated transition, the estimated reward, and the bonus in the $n$-th iteration of Algorithm 2 (ELNF), respectively. |
| $\{\phi_h^n\}_{h=1}^H, \{\psi_h^n\}_{h=2}^{H+1}$ | The learned feature in the $n$-th iteration of Algorithm 2 (ELNF), respectively. |
| $K, \mathcal{H}, k$ | $K(x,y) = \exp(-\|x-y\|_2^2)$, $\mathcal{H}$ is the RKHS induced by $K$, and $k$ is the feature map of $\mathcal{H}$. |
| $\Delta(\mathcal{C}), \mathcal{U}(\mathcal{C})$ | The set of the distributions over $\mathcal{C}$, and the uniform distribution over $\mathcal{C}$. |
| $R$ | The upper bound of the norm of the features. |
| $C_{\text{net}}, \gamma$ | The parameters in Assumption 5.2. |
| $\lambda, \beta$ | The tuning parameters of Algorithm 2 (ELNF). |
| $d_{\text{eff}}$ | The effective dimension we defined in Definition E.4. |
| $c[\phi, \psi](s, a)$ | The normalization function we defined in (E.1). |
| $\mathcal{N}(\mathcal{F}, \epsilon, \|\cdot\|)$ | The $\epsilon$-covering number of the function class $\mathcal{F}$ with respect to the norm $\|\cdot\|$. |
| $H_\infty(\mathcal{F}, \epsilon), H_2(\mathcal{F}, \epsilon)$ | $H_\infty(\mathcal{F}, \epsilon) = \log \mathcal{N}(\mathcal{F}, \epsilon, \|\cdot\|_\infty)$, and $H_2(\mathcal{F}, \epsilon) = \log \mathcal{N}(\mathcal{F}, \epsilon, \|\cdot\|_{\infty,2})$. |
| $c_{\max}, c_{\min}, r_{\max}$ | $c_{\max}$ is the upper bound of the feature maps, $c_{\min}$ is the lower bound of the feature maps, and $r_{\max} = c_{\max}/c_{\min}$. |
| $\zeta$ | The parameter that related to the upper bound of the estimation error in Lemma E.3. |

STRUCTURE OF APPENDIX

We provide a detailed comparison between our work and the existing literature in Appendix §A, and provide the supplementary for the planning algorithm, the NNs, and the RKHS in Appendix §B, Appendix §C, and Appendix §D. We then provide a detailed proof of Theorem 5.3 in Appendix §E. We provide the proof of the key lemmas used in Appendix §E, including Lemma E.2, Lemma E.3, and Lemma E.8, in Appendix §F, Appendix §G, and Appendix §H. We provide the proof of other lemmas used in Appendix §E in Appendix §I, and provide the proof of other auxiliary lemmas in Appendix §J.

# Appendix

## Table of Contents

## A  DETAILED COMPARISON WITH EXISTING LITERATURE

In this section, we provide detailed comparison with existing literature.

**Comparison with Kakade et al. (2020).** Kakade et al. (2020) studies the setting of KNRs and proposes the algorithm Lower Confidence-based Continuous Control (LC$^3$). They show that LC$^3$ is sample efficient by bounding the expectation of the regret from the above by $\widetilde{\mathcal{O}}(H^{3/2}N^{1/2}d_{\text{eff}})$, which aligns with our results in terms of $N^{1/2}$ and $d_{\text{eff}}$. However, they assume that the nonlinear feature map is known, while we need to learn it from prespecified NN classes. Thus, our model is significantly more challenging. Moreover, their algorithm requires optimism over the confidence set, which is not computational efficient.

**Comparison with Research on Linear MDPs and Low-rank MDPs.** We show in §D.1 that our model can be generalized to include linear MDP with unknown feature maps and nonlinear reward as a special case. Jin et al. (2020) show that their algorithm achieves $\widetilde{\mathcal{O}}(\sqrt{m^3H^3N})$ regret. Our bound on suboptimality aligns with their results in terms of the dimension $m$ and the number of iterations $N$ without assuming known feature maps and linear reward functions. Uehara et al. (2021) and Ren et al. (2022) propose algorithms for linear MDPs with unknown feature maps and show that the sample complexity of their algorithm is $\widetilde{\mathcal{O}}(m^4/\epsilon^2)$ in infinite-horizon linear MDP with unknown feature maps, which also aligns with us in terms of $\epsilon$. However, they assume that the transition is a finite dimensional inner product, and the unknown feature maps belong to a finite set, which greatly reduces the complexity of the problem.

**Comparison with Dong et al. (2021).** The sample complexity is $\widetilde{\mathcal{O}}(\epsilon^{-2})$ in terms of $N$ when the logarithmic factors are omitted and $\epsilon = 0$. Meanwhile, Theorem 5.1 in Dong et al. (2021) shows that the minimax sample complexity of solving a nonlinear bandit problem with one-layer NNs and ReLU activation is $\Omega(\epsilon^{-(d-2)})$. To obtain such a lower bound, Dong et al. (2021) assume that the action space is the unit sphere $\mathbb{S}^{d-1}$ in $\mathbb{R}^d$, which is an infinite set, while the action space in our setting is finite. In the case where $H = 1$, our model reduces to a finite-arm bandit problem whose reward is parameterized by an NN. Although the Eluder dimension of the NN class is large, the agent only needs to explore the arms of the bandits in our model, while the agent needs to explore the unit sphere

in their model. Therefore, their model does not belong to our model, our result does not contradict the lower bound in Dong et al. (2021), and the sample complexity in our model is dominated by the number of arms instead of the Eluder dimension of the NN class when $H = 1$.

**Comparison with Ren et al. (2021).** Ren et al. (2021) studies a nonlinear model with Gaussian noise. They show that the expectation of the regret of their algorithm is

$$\widetilde{\mathcal{O}}\Big( \sqrt{H^2 N \cdot \log \mathcal{N}(\Phi, N^{-1/2}, \|\cdot\|_2) \cdot \dim_E(\Phi, N^{-1/2})} \Big),$$

where $\dim_E(\Phi, \cdot)$ is the Eluder dimension of $\Phi$. We show in §3 that the model in Ren et al. (2021) is a special case of our model. Our bound on the suboptimality aligns with their result in terms of the number of iterations $N$ when $\gamma = 0$. However, they do not fully exploit the kernel structure in the transition in their analysis, and their result depends on the Eluder dimension of $\Psi$. Lemma C.7 in Appendix §C.2 provides an example of an NN class whose $\epsilon$-Eluder dimension is at least $\Omega(\epsilon^{-(d-1)})$ and the $\epsilon$-log covering number is at most $\mathcal{O}(\log(1/\epsilon))$. Lemma C.7 shows that removing the dependency of the sample complexity on the Eluder dimension significantly improves the sample complexity. In addition, their algorithm requires sampling from the posterior distribution of the hypothesis class, which is difficult to implement in practice. In contrast, our algorithm only requires planning with respect to the learned model, which can be computed efficiently.

**Comparison with Yang et al. (2020).** Yang et al. (2020) use overparameterized NNs for function approximation in the algorithm Neural Optimistic Least-Squares Value Iteration (`NOVI`) and shows that `NOVI` is sample efficient. However, their analysis relies on the connection between overparameterized NNs and neural tangent kernel and can not handle NNs beyond NTK regime.

## B SUPPLEMENTARY FOR ALGORITHM

As we mentioned in §4, our algorithm performs the following four steps in each iteration: (i) sampling new data from the environment, (ii) estimating the model via maximum likelihood estimation, (iii) constructing exploration incentives using the features of the learned model, and (iv) updating the online policy for exploration via planning on the learned model. We simplify the presentation of the sampling algorithm and omit the details of the planning algorithm in the main text due to the space limit. In this section, we first describe the sampling scheme rigorously in detail. We then provide the detail of the planning algorithm.

**Sampling Scheme.** As we mentioned in §3, the transition of MDPs with neural dynamics can be written as an energy-based model and admits the Gaussian RBF kernel. To exploit the kernel structure in the transition, we explore the environment using the exploration bonus induced by the Gaussian RBF kernel and the feature maps learned from the data, which is motivated by Yang et al. (2020). However, since the bonus is not induced by the true underlying feature, it might fail to indicate the most uncertain state-action pairs for exploration. To mitigate such an issue, we combine the uniform policy, which samples action from the uniform distribution over the action space, with the optimistic policy during the sampling procedure. Intuitively, such a sampling scheme provides a wider coverage over the state-action space and better explores the environment.

To simplify the presentation of the algorithm in our work, we introduce an extended MDP, where we assign meanings to steps $h = -1$, $0$, $H + 1$, and $H + 2$. In particular, the interaction of an agent with the extended MDP starts with a dummy initial state $s_{-1}$. During the interaction, all the dummy state and action sequences $\{s_{-1}, a_{-1}, s_0, a_0\}$ lead to the same initial state $s_{\text{init}}$. Moreover, the agent is allowed to interact with the environment for two steps after observing the final state $s_{H+1}$ of an episode. Nevertheless, the agent only collects the reward $r_h(s_h, a_h)$ at steps $h \in [H]$, which leads to the same learning objective as the original MDP. In addition, we denote by $[H]^+ = [-1, 0, \ldots, H + 2]$ the set of steps in the extended MDP. We remark that the dummy state and action sequences $\{s_{-1}, a_{-1}, s_0, a_0\}$ do not exist, and we introduce them just to simplify the rigorous presentation of the boundary case of our algorithm. In the sequel, we do not distinguish between an MDP and an extended MDP for the simplicity of presentation.

Now we describe the sampling procedure in detail. In the $n$-th iteration of our algorithm, given the previously collected dataset $D_{h,i}^{n-1}$ for $i \in \{0, 1, 2\}$ and $h \in [H]$, we interact with the MDP following the policy $\pi^n = \{\pi_h^n\}_{h=-1}^H$ and obtain the new dataset $D_{h,i}^n$ for $i \in \{0, 1, 2\}$ and $h \in [H]$.

Specifically, for any $h \in \{-1, \ldots, H\}$, we start from the initial state $s_{-1}$ and choose the action $a_{\bar{h}} \sim \pi_{\bar{h}}^n(\cdot \mid s_{\bar{h}})$ in the $\bar{h}$-th step when $\bar{h} \in \{-1, \cdots, h\}$, and choose the action $a_{\bar{h}} \sim \mathcal{U}(\mathcal{A})$ when $\bar{h} \in \{h+1, h+2\}$. Here $\mathcal{U}(\mathcal{A})$ is the uniform distribution over the action space $\mathcal{A}$. By following such a procedure, we obtain the following trajectory,

$$s_{-1}, a_{-1}, s_0, a_0, s_1, a_1, r_1, \ldots, s_{h+2}, a_{h+2}, r_{h+2}, s_{h+3}, \tag{B.1}$$

where $s_1 = s_{\text{init}}$. Then, we label the obtained trajectory as follows,

$$s_{h+i,i}^n = s_{h+i}, \qquad a_{h+i,i}^n = a_{h+i}, \qquad r_{h+i,i}^n = r_{h+i}, \qquad \bar{s}_{h+i,i}^n = s_{h+i+1},$$

for any $i \in \{0, 1, 2\}$. We then update the dataset as follows,

$$\mathcal{D}_{h,i}^n = \mathcal{D}_{h,i}^{n-1} \cup \left\{ (s_{h,i}^n, a_{h,i}^n, r_{h,i}^n, \bar{s}_{h+i,i}^n) \right\} = \left\{ (s_{h,i}^\tau, a_{h,i}^\tau, r_{h,i}^\tau, \bar{s}_{h+i,i}^\tau) \right\}_{\tau=1}^n \tag{B.2}$$

for any $i \in \{0, 1, 2\}$ and $h \in [H]$. The index $i$ in (B.2) indicates how many steps of the uniform policy we need to execute to obtain such a dataset. Intuitively, the dataset with a bigger index $i$ has a better coverage over the state-action space $\mathcal{S} \times \mathcal{A}$. See Figure 1 for an illustration of the sampling scheme, which is summarized in Algorithm 3.

---

**Algorithm 3** Sampling Scheme (Formal)

1: **Input:** Policy $\pi^n = \{\pi_h^n\}_{h=-1}^H$, datasets $\mathcal{D}_{h,i}^{n-1}$ for $i \in \{0, 1, 2\}$ and $h \in [H]$.
2: **for** $h = -1, \ldots, H$ **do**
3:     Interact with the environment to obtain the trajectory in (B.1) by first executing $\pi^n$ from $s_{-1}$ to $s_h$, and then executing $\mathcal{U}(\mathcal{A})$ for two more steps.         ▷ Sampling
4:     Set $(s_{h+i,i}^n, a_{h+i,i}^n, r_{h+i,i}^n, \bar{s}_{h+i,i}^n) \leftarrow (s_{h+i}, a_{h+i}, r_{h+i}, s_{h+i+1})$ for $i \in \{0, 1, 2\}$, where $(s_{h+i}, a_{h+i}, r_{h+i}, s_{h+i+1})$ is defined in (B.1).
5:     Set $\bar{\mathcal{D}}_{h+i,i}^n \leftarrow \left\{ (s_{h+i,i}^n, a_{h+i,i}^n, r_{h+i,i}^n, \bar{s}_{h+i,i}^n) \right\}$ for $i \in \{0, 1, 2\}$.         ▷ Labeling
6: **end for**
7: **for** $h = 1, \ldots, H$ **do**         ▷ Updating the datasets
8:     Set $\mathcal{D}_{h,i}^n \leftarrow \mathcal{D}_{h,i}^{n-1} \cup \bar{\mathcal{D}}_{h,i}^n$ for $i \in \{0, 1, 2\}$.
9: **end for**
10: **Return:** Datasets $\{\mathcal{D}_{h,i}^n\}_{h=1,i=0}^{h=H,i=2}$.

---

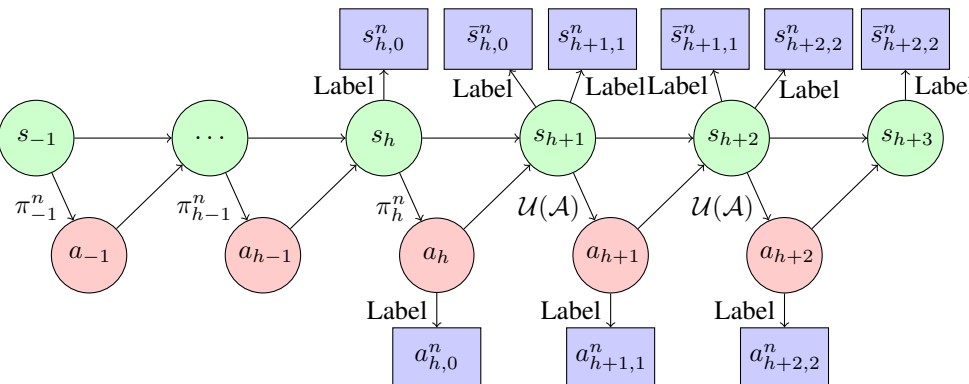

Figure 1: Sampling procedure in the $h$-th trajectory of the $n$-iteration. We first execute the optimistic policy for $h$ steps, and then execute the uniform policy for two steps. Finally, we label the collected data as the figure shows.

**Planning Algorithm.** The details of the planning algorithm is now summarized in Algorithm 4.

---

**Algorithm 4** Planning Algorithm

1: **Input:** Estimated reward $\{\widehat{r}_h^n\}_{h=1}^H$, estimated transition $\{\mathcal{P}_h^n\}_{h=1}^H$, bonus $\{\widehat{u}_h^n\}_{h=1}^H$.
2: Set $Q_H^n(s,a) \leftarrow \widehat{r}_H^n(s,a) + u_H^n(s,a)$.
3: Set $V_H^n(s) \leftarrow \max_{a \in \mathcal{A}} Q_H^n(s,a)$ and $\pi_H^{n+1}(\cdot \mid s) \leftarrow \operatorname{argmax}_{\pi \in \Delta(\mathcal{A})} \sum_{a \in \mathcal{A}} Q_H^n(s,a)\pi(a)$.
4: **for** $h = H-1, \ldots, 1$ **do**
5:    Set $Q_h^n(s,a) \leftarrow \widehat{r}_h^n(s,a) + u_h^n(s,a) + \int_{\mathcal{S}} \mathcal{P}_h^n(s' \mid s,a) V_{h+1}^n(s') \mathrm{d}s'$.
6:    Set $V_h^n(s) \leftarrow \max_{a \in \mathcal{A}} Q_h^n(s,a)$ and $\pi_h^{n+1}(\cdot \mid s) \leftarrow \operatorname{argmax}_{\pi \in \Delta(\mathcal{A})} \sum_{a \in \mathcal{A}} Q_h^n(s,a)\pi(a)$.
7: **end for**
8: Set $\pi_{-1}^{n+1}$ and $\pi_0^{n+1}$ as the uniform policy $\mathcal{U}(\mathcal{A})$.
9: **Return:** $\pi^{n+1} \leftarrow \{\pi_h^{n+1}\}_{h=-1}^H$.

---

We remark that we can also apply other model-based algorithm, including *Dyna* (Sutton, 1991) and *Gradient-Aware Model-based Policy Search* (D'Oro et al., 2020), to compute the optimal policy of the learned model.

## C SUPPLEMENTARY FOR NEURAL NETWORKS

In this section, we provide more details on neural networks. In the first subsection, we introduce the definition of the covering number, which measure the complexity of a function class, and provide two examples of neural networks in detail and show that such neural networks satisfy our assumptions. **We remark that our analysis can be extended to other neural network classes when Assumption 5.2 is satisfied with $\gamma \le 1/2$, and is not restricted to the examples below.** In the second subsection, we provide a lemma that compares the log-covering number of an NN class with the Eluder dimension. Lemma C.7 in Appendix §C.2 shows that removing the dependency of the sample complexity on the Eluder dimension greatly improves the sample complexity.

### C.1 EXAMPLES

We begin with the definition of the covering number.

**Definition C.1** (Covering Number). *Let $(\Phi, \| \cdot \|)$ be a normed space, and $\Phi_0 \subset \Phi$. The set $\{\phi_1, \phi_2, \ldots, \phi_s\}$ is an $\epsilon$-covering of $\Phi_0$ if $\sup_{\phi \in \Phi_0} \inf_{i \in [s]} \|\phi - \phi_i\| \le \epsilon$. We define the covering number $\mathcal{N}(\Phi_0, \epsilon, \|\|)$ as the minimum size of such a covering.*

In what follows, we provide two examples of neural networks in detail and show that such neural networks satisfy our assumptions.

**Example 1.** The first example is the $s$-sparse neural network class. Schmidt-Hieber (2020) show that the $s$-sparse neural network class satisfies Assumptions 5.1 and 5.2. We first introduce the definition.

**Definition C.2** (The $s$-Sparse Neural Network Class). *We define $\sigma(x) = \max\{x, 0\}$. For a vector $v \in \mathbb{R}^r$, we define the shifted activation function $\sigma_v : \mathbb{R}^r \to \mathbb{R}^r$ as*

$$\sigma_v\big((y_1, \ldots, y_r)^\top\big) = \big(\sigma(y_1 - v_1), \ldots, \sigma(y_r - v_r)\big)^\top.$$

*A neural network with network architecture $(L, p)$ is a function that takes the form,*

$$f : \mathbb{R}^{p_0} \to \mathbb{R}^{p_{L+1}}, \qquad f(x) = W_L \sigma_L \ldots \sigma_1 W_0 x. \tag{C.1}$$

*when $x \in \mathbb{R}^{p_0}$. Here $L$ is the depth of the neural network, $W_i$ is a $p_{i+1} \times p_i$ weight matrix, $v_i \in \mathbb{R}^{p_i}$ is a shift vector, and $p = (p_0, \ldots, p_{L+1})^\top$. The neural network class $\mathcal{F}_{nn}(L, p)$ is defined as*

$$\mathcal{F}_{nn}(L, p) = \big\{ f \mid f \text{ takes the form of (C.1) with } \max_{j=0,\ldots,L} \|W_j\|_\infty \vee \|v_j\|_\infty \le 1 \big\}.$$

*The $s$-sparse neural network class is defined as*

$$\mathcal{F}_{nn}(L, p, s, M) = \big\{ f \mid f \in \mathcal{F}_{nn}(L, p), \sum_{j=0}^L \|W_j\|_0 + \|v_j\|_0 \le s, \sup_{x \in \mathcal{X}} \|f(x)\|_\infty \le M \big\}.$$

We directly obtain the boundedness of the neural network class $\mathcal{F}_{\text{nn}}(L, p, s, M)$ by the definition. Moreover, Schmidt-Hieber (2020) bound the covering number of $\mathcal{F}_{\text{nn}}(L, p, s, M)$ from the above by the following lemma.

**Lemma C.3** (Lemma 5 in Schmidt-Hieber (2020)). *We define $V = \prod_{l=0}^{L+1}(p_l + 1)$. Then, for any $\epsilon > 0$, we have*

$$\log \mathcal{N}\Big(\mathcal{F}_{\text{nn}}(L, p, s, M), \epsilon, \|\cdot\|_\infty\Big) \leq (s+1)\log\big(2(L+1)V^2/\epsilon\big).$$

Lemma C.3 verifies that the $s$-sparse neural network class satisfies Assumption 5.2 with $\gamma = 0$. Therefore, the $s$-sparse neural network class defined in Definition C.2 satisfies Assumptions 5.1 and 5.2.

**Example 2.** The second example is the recurrent neural network class. Chen et al. (2019) show that the recurrent neural network class satisfies Assumptions 5.1 and 5.2. We first introduce the definition.

**Definition C.4** (Recurrent Neural Network Class). *A recurrent neural network $f(X_t, t; U, W, h_0)$ is a mapping that, when it is parameterized by $U \in \mathbb{R}^{d_h \times d_h}$, $V \in \mathbb{R}^{d_y \times d_h}$, $W \in \mathbb{R}^{d_h \times d_x}$, $h_0 \in \mathbb{R}^{d_h}$ and get $X_t = (x_1, \ldots, x_t) \in \mathbb{R}^{d_x \times t}$, $t$ as input, it returns $Y_t = (y_1, \ldots, y_t) \in \mathbb{R}^{d_y \times t}$*

$$h_t = \sigma_h(U h_{t-1} + W x_t), \qquad y_t = \sigma_y(V h_t),$$

*where $\sigma_h$ and $\sigma_y$ are two nonlinear activation functions. We define the function class $\mathcal{F}_{\text{RNN}}(t, B)$ as the set of functions that take the form of $f(\cdot, t; U, W, h_0)$, where $f$ is a recurrent neural network that only takes bounded input, and the spectral norm of $U, V$, and $W$ are all bounded by $B$.*

In Chen et al. (2019), they analyze the recurrent neural network class under the following assumption.

**Assumption C.5.** *We assume that the activation functions $\sigma_h$ and $\sigma_y$ are Lipschitz with parameters $\rho_h$ and $\rho_y$ respectively, and $\sigma_h(0) = \sigma_y(0) = 0$. Additionally, we assume that $\sigma_h$ is entrywise bounded by $M$.*

Assumption C.5 can be satisfied by a lot of activation functions. For example, Assumption C.5 is satisfied when we choose $\sigma_h(\cdot) = \tanh(\cdot)$ and $\sigma_y(\cdot) = \max(0, \cdot)$. We directly obtain the boundedness of the function class $\mathcal{F}_{\text{RNN}}(t, B)$ under Assumption C.5. Moreover, Chen et al. (2019) bound the covering number of $\mathcal{F}_{\text{RNN}}(t, B)$ from the above by the following lemma.

**Lemma C.6** (Lemma 3 in Chen et al. (2019)). *Under Assumption C.5, we have*

$$\log \mathcal{N}\big(\mathcal{F}_{\text{RNN}}(t, B), \epsilon, \|\cdot\|_\infty\big) \leq 3d^2 \log\Big(1 + 6ct\sqrt{d}\big((\rho_h B - 1)^t - 1\big)/\big(\epsilon(\rho_h B - 1)\big)\Big),$$

*where $d = \sqrt{d_h(d_x + d_y + d_h)}$ and $c = \rho_y \rho_h B^3 \max\{1, \rho_h B\}$.*

Lemma C.6 verifies that the recurrent neural network class satisfies Assumption 5.2 with $\gamma = 0$. Therefore, the recurrent neural network class defined in Definition C.4 satisfies Assumptions 5.1 and 5.2.

## C.2 COMPARISON OF ELUDER DIMENSION AND LOG-COVERING NUMBER

In this section, we provide a lemma that use an example to illustrate that, the log-covering number of a NN class can grow more moderately as the covering radius decays than the Eluder dimension.

**Lemma C.7.** *Let $\mathcal{R}$ be a one-layer NN class defined as*

$$\mathcal{R} = \big\{r_{\theta, b} : \mathcal{A} \to \mathbb{R} \mid r_{\theta, b}(a) = \sigma(\theta^\top a + b), \ \theta \in \mathbb{R}^d, \ \|\theta\|_2 \leq 1, \ 0 \leq b \leq 1\big\},$$

*where $\sigma$ is the activation function in Definition C.2. Suppose we have $\mathcal{A} \subset \mathbb{R}^d$ and $\|a\|_2 \leq 1$ when $a \in \mathcal{A}$, then the $\epsilon$-Eluder dimension of $\mathcal{R}$ is at least $\Omega(\epsilon^{-(d-1)})$, and the $\epsilon$-log covering number is bounded from the above by $\mathcal{O}(\log(1/\epsilon))$.*

*Proof.* See Appendix §J.4 for a detailed proof. $\square$

Lemma C.7 suggests that, the log-covering number of an NN class can be much smaller than the Eluder dimension. Our algorithm is sample efficient since the sample complexity of ELNF depends on the log-covering number of the NN class instead of the Eluder dimension. We remark that the NN class in Lemma C.7 also satisfies Assumption 5.2 with $\gamma = 0$.

# D SUPPLEMENTARY FOR REPRODUCING KERNEL HILBERT SPACES

As shown in Nachum & Yang (2021) and Ren et al. (2021), the transition kernel in our model is closely related to the RKHS corresponding to the Gaussian kernel. In this section, we provide more details on the RKHS. In the first subsection, we provide the definition of RKHSs and some basic properties of RKHSs. In the second subsection, we lay out several properties of the population operator and the empirical operator defined in (E.4) and (E.2).

## D.1 BASIC CONCEPT

We will be considering the use of RKHS for designing the exploration bonus. We first introduce the definition of RKHS and positive-definite kernels.

**Definition** (Reproducing Hilbert Kernel Space). *Let $\mathcal{H}$ be a vector space which consists of functions that take element in $\mathcal{X}$ as input and take their values in $\mathbb{R}$. We assume that $\mathcal{H}$ is equiped with an inner-product $\langle \cdot, \cdot \rangle_{\mathcal{H}} : \mathcal{H} \times \mathcal{H} \to \mathbb{R}$. The function $K : \mathcal{X} \times \mathcal{X} \to \mathbb{R}$ is a reproducing kernel of $\mathcal{H}$ if it satisfies the following properties, namely, (1) we have $K(x, \cdot) \in \mathcal{H}$ for all $x \in \mathcal{X}$, and (2) we have $\langle g, K(x, \cdot) \rangle_{\mathcal{H}} = g(x)$ for all $x \in \mathcal{X}$ and $g \in \mathcal{H}$. We call $\mathcal{H}$ a reproducing Hilbert kernel space (RKHS) if it is a Hilbert space with a reproducing kernel $K$.*

**Definition** (Positive-Definite Kernel). *A kernel function $K : \mathcal{X} \times \mathcal{X} \to \mathbb{R}$ is positive definite if $\sum_{i=1}^{n} \sum_{j=1}^{n} a_i a_j K(x_i, x_j) \geq 0$ for all $n \geq 1$, $(a_1, \ldots, a_n)^\top \in \mathbb{R}^n$ and $(x_1, \ldots, x_n)^\top \in \mathcal{X}^n$.*

The following lemma reveals the connection between MDPs with neural dynamics and RKHS.

**Lemma.** *There exists an RKHS $\mathcal{H}$, such that $K(x_1, x_2) = \exp(-\|x_1 - x_2\|_2^2 / 2)$ is the kernel of $\mathcal{H}$. The kernel is called the Gaussian kernel.*

*Proof.* Rahimi et al. (2007) show that the Gaussian kernel is a positive-definite kernel on $\mathcal{X} \subset \mathbb{R}^m$. Moore-Aronszajn theorem (Aronszajn (1950)) shows that for every positive-definite kernel, there exists an RKHS $\mathcal{H}$ associated with the kernel. Thus, we conclude the proof. $\square$

We remark that by the similarity between different RKHSs, our analysis can be adapted to the cases when the right-hand-side of (3.4) is another kernel function.

**Definition D.1** (Generalization of Definition 3.1). *An episodic Markov decision process $(\mathcal{S}, \mathcal{A}, H, \mathcal{P}^*, r^*)$ is an MDP with neural dynamics if its reward functions $r^* = \{r_h^*\}_{h=1}^H \subset \mathcal{R}$, where $\mathcal{R}$ is a known reward function class that consists of neural networks, and the transition kernel of the MDP $\mathcal{P}^* = \{\mathcal{P}_h^*\}_{h=1}^H$ takes the following form,*

$$\mathcal{P}_h^*(s_{h+1} \mid s_h, a_h) = K\big(\phi_h^*(s_h, a_h), \psi_{h+1}^*(s_{h+1})\big).$$

*Here $K$ is a positive-definite kernel, $\phi_h^* \in \Phi : \mathbb{R}^d \times \mathcal{A} \to \mathbb{R}^m$ and $\psi_{h+1}^* \in \Psi : \mathbb{R}^d \to \mathbb{R}^m$ are two unknown neural networks, and $\Phi$ and $\Psi$ are two known feature map classes that consist of neural networks. We denote by $\mathcal{M}$ the set of all the transition kernels that take the form of (3.4), and denote by $\mathcal{X} \in \mathbb{R}^m$ the space of the embedded feature.*

We remark that when $K(x_1, x_2) = x_1^\top x_2$ in Definition D.1, we recover linear MDP with unknown feature and nonlinear reward function.

## D.2 OPERATOR PROPERTIES

In this subsection, we provide the properties of operators defined in (E.2) and (E.4). The following lemma shows that $\Gamma_e[\phi, \mathcal{D}, \lambda]$ and $\Gamma_p[\phi, \rho, \lambda, \tau]$ are both positive-definite operators on $\mathcal{H}$, which guarantees the existence of the corresponding inverse.

**Lemma D.2.** *Let $\rho$ be a probability measure over $\mathcal{X}$, and $L_\rho^2(\mathcal{X})$ be the set of all functions that is square-integrable on $\mathcal{X}$ with respect to $\rho$. We have $\mathcal{H} \subset L_\rho^2(\mathcal{X})$. When we define the operator $\Gamma$ as $\Gamma g(x) = \int_{\mathcal{X}} K(x', x) g(x') \mathrm{d}\rho(x')$, we have $\Gamma g \in \mathcal{H}$ when $g \in L_\rho^2(\mathcal{X})$. Moreover, when $\Gamma$ is regarded as an operator from $\mathcal{H}$ to $\mathcal{H}$, it is non-negative definite.*

*Proof.* See Appendix §J.5 for a detailed proof. $\square$

The following lemma shows that the operator norm of the inverse operator of the empirical operator can be computed efficiently.

**Lemma D.3.** *Let $\mathcal{H}$ be an RKHS. For $\mathcal{X}_n = \{x_1, \ldots, x_n\}$, we define $\Gamma[\mathcal{X}_n] : \mathcal{H} \to \mathcal{H}$ as*

$$\Gamma[\mathcal{X}_n]g(x) = \lambda g(x) + \sum_{i=1}^{n} g(x_i)K(x_i, x).$$

*Here $K$ is the kernel of the RKHS. We denote by $k$ the feature map of the RKHS. We have*

$$\|k(x)\|_{\Gamma^{-1}[\mathcal{X}_n]}^2 = \Big(K(x,x) - k[\mathcal{X}_n](x)^\top \big(\lambda I + K[\mathcal{X}_n]\big)^{-1} k[\mathcal{X}_n](x)\Big)\Big/\lambda.$$

*Here $k[\mathcal{X}_n](x) = (K(x_1, x), \ldots, K(x_n, x))^\top \in \mathbb{R}^n$ and $K[\mathcal{X}_n] = [K(x_{\tau_1}, x_{\tau_2})]_{\tau_1, \tau_2=1}^n \in \mathbb{R}^{n \times n}$.*

*Proof.* See Appendix §J.6 for a detailed proof. $\square$

We end this subsection by bounding the operator norm of the operators $\Gamma_e[\phi, \mathcal{D}, \lambda]$ and $\Gamma_p[\phi, \rho, \lambda, \tau]$.

**Lemma D.4.** *For any $g \in \mathcal{H}$, we have*

$$\lambda \langle g, g \rangle_{\mathcal{H}} \leq \big\langle g, \Gamma_e[\phi, \mathcal{D}, \lambda]g \big\rangle_{\mathcal{H}} \leq \Big(\lambda + \sum_{(s,a,r,s') \in \mathcal{D}} K\big(\phi(s,a), \phi(s,a)\big)\Big) \langle g, g \rangle_{\mathcal{H}},$$

$$\lambda \langle g, g \rangle_{\mathcal{H}} \leq \big\langle g, \Gamma_p[\phi, \rho, \lambda, \tau]g \big\rangle_{\mathcal{H}} \leq \Big\{\lambda + \tau \mathbb{E}_{(s,a) \sim \rho}\Big[K\big(\phi(s,a), \phi(s,a)\big)\Big]\Big\} \langle g, g \rangle_{\mathcal{H}}.$$

*Proof.* See Appendix §J.7 for a detailed proof. $\square$

# E    PROOF OF THEOREM 5.3

We first state the theorem again with clearer description on the choice of the parameters.

**Theorem** (Theorem 5.3, restated). *For an MDP with neural dynamics that satisfies Assumptions 5.1 and 5.2 with $\gamma < 1/2$, we set*

$$\lambda = Cn^{\gamma/(1+\gamma)}m\log(48HRN/\delta), \qquad \beta = 2(H+1)\sqrt{4\lambda v^2/c_{\min}^2 + 10r_{\max}^2 \zeta^2 |\mathcal{A}|}.$$

*in Algorithm 2 (ELNF), where $m$ is the dimension of the image of the feature maps. Suppose we can obtain the exact solution to the optimization problems (4.3) and (4.4). We have*

$$J(\pi^*; r^*, \mathcal{P}^*) - J(\widehat{\pi}; r^*, \mathcal{P}^*) \leq CH^5 |\mathcal{A}|^2 N^{(2\gamma-1)/(2+2\gamma)} \big(\log(48HRN/\delta)\big)^{5/2} (\log N)^{m+1}$$

*with probability at least $1 - \delta$, where $\widehat{\pi}$ is the policy returned by Algorithm 2 (ELNF). Here $C$ is a constant only depends on the dimension $m$, the bound of the feature maps $R$, and $C_{\mathrm{net}}$ in Assumption 5.2, and $\zeta$ is defined as*

$$\zeta = \sqrt{C_2 \log(20HRN/\delta)N^{\gamma/(1+\gamma)}},$$

*where $C_2$ is a constant depends on the regularity parameters and the parameters of the NN class.*

*Proof.* We define the normalization function $c[\phi_h^*, \psi_{h+1}^*](\phi_h^*(s_h, a_h))$ as

$$c[\phi_h^*, \psi_{h+1}^*]\big(s_h, a_h\big) = \int_{s_{h+1} \in \mathcal{S}} \exp\Big(-\big\|\phi_h^*(s_h, a_h) - \psi_{h+1}^*(s_{h+1})\big\|_2^2 /2\Big)\mathrm{d}s_{h+1}. \tag{E.1}$$

By (3.4) and (E.1), the transition kernel $\mathcal{P}_h^*$ can be written as

$$\mathcal{P}_h^*(s_{h+1} \mid s_h, a_h) = \exp\Big(-\big\|\phi_h^*(s_h, a_h) - \psi_{h+1}^*(s_{h+1})\big\|_2^2 /2\Big)\Big/c[\phi_h^*, \psi_{h+1}^*]\big(s_h, a_h\big).$$

We define the bound on the normalization constant as

$$c_{\max} = \sup_{\substack{\phi \in \Phi, \psi \in \Psi \\ (s,a) \in \mathcal{S} \times \mathcal{A}}} c[\phi, \psi](s,a), \quad c_{\min} = \inf_{\substack{\phi \in \Phi, \psi \in \Psi \\ (s,a) \in \mathcal{S} \times \mathcal{A}}} c[\phi, \psi](s,a), \quad r_{\max} = c_{\max}/c_{\min}.$$

We have $c_{\max} \le v$, $c_{\min} \ge v\exp(-2R^2)$, and $r_{\max} \le \exp(2R^2)$ when Assumption 5.1 holds. Here $v$ is the Lebesgue measure of $\mathcal{S}$.

Since the kernel function $K(x_1, x_2) = \exp(\|x_1 - x_2\|_2^2 / 2)$ is a positive-definite kernel, it induces an RKHS. We denote by $\mathcal{H}$ the RKHS induced by $K$, and denote by $k$ the corresponding feature map. This allows us to corporate techniques in RKHS into our analysis. The following lemma reveals the relationship between the bonus we define and $\mathcal{H}$.

**Lemma E.1.** *For a finite dataset* $\mathcal{D} = \{(s_\tau, a_\tau, r_\tau, s_{\tau+1})\}_{\tau=1}^n$ *and* $g \in \mathcal{H}$, *we define the empirical operator on* $\mathcal{H}$ $\Gamma_e[\phi, \mathcal{D}, \lambda]$ *as*

$$\Gamma_e[\phi, \mathcal{D}, \lambda]g(x) = \lambda g(x) + \sum_{(s,a,r,s') \in \mathcal{D}} g(\phi(s,a))K(\phi(s,a), x), \qquad \text{(E.2)}$$

*and denote by* $\Gamma_e^{-1}[\phi, \mathcal{D}, \lambda]$ *the corresponding inverse operator. Then we have*

$$u_h^n(s,a) = \min\left\{ 2H + 2, \beta \left\| k(\widehat{\phi}_h^n(s,a)) \right\|_{\Gamma_e^{-1}[\widehat{\phi}_h^n, \mathcal{D}_{h,1}^n, \lambda]} \right\}.$$

*Proof.* We conclude the proof by directly applying Lemma D.3. $\qquad \square$

In the sequel, we introduce two good events. We first define distributions and the population operator, which we use in the definitions of the good events. For an index $n \in [N]$ and $i \in \{0, 1, 2\}$, we denote by $\vartheta_{h,i}^n$ the distribution of $(s_h, a_h)$ when $s_1 = s_{\text{init}}$, the state $s_{\bar{h}+1} \sim \mathcal{P}_{\bar{h}}^*(\cdot \mid s_{\bar{h}}, a_{\bar{h}})$, the action $a_{\bar{h}} \sim \pi_{\bar{h}}^n(\cdot \mid s_{\bar{h}})$ for $\bar{h} \in [h - i]$, and $a_{\bar{h}} \sim \mathcal{U}(\mathcal{A})$ for $\bar{h} \in \{h - i + 1, \dots, h\}$, where $\mathcal{U}(\mathcal{A})$ is the uniform distribution over the action set $\mathcal{A}$. We further define the measure $\rho_{h,i}^n$ as

$$\rho_{h,i}^n(s,a) = \sum_{\tau=1}^n \vartheta_{h,i}^\tau(s,a)/n, \qquad \forall (s,a) \in \mathcal{S} \times \mathcal{A}. \qquad \text{(E.3)}$$

For a distribution $\rho$ over $\mathcal{S} \times \mathcal{A}$, we define the population operator as

$$\Gamma_p[\phi, \rho, \lambda, \tau]g(x) = \lambda g(x) + \tau \mathbb{E}_{(s,a) \sim \rho} g(\phi(s,a))k(\phi(s,a), x), \qquad \text{(E.4)}$$

and denote by $\Gamma_p^{-1}[\phi, \rho, \lambda, \tau]$ the corresponding inverse operator. For the empirical operator and the population operator, we have $\Gamma_e[\phi, \mathcal{D}, \lambda]g \in \mathcal{H}$ and $\Gamma_p[\phi, \rho, \lambda, \tau]g \in \mathcal{H}$ when $g \in \mathcal{H}$. The operators in (E.2) are also positive-definite and self-adjoint for any dataset $\mathcal{D}$ when $\lambda, \tau > 0$, which guarantees the existence of its inverse. We denote by $\mathcal{E}_1$ that the bonus defined by the empirical operator is bounded from the above and below by the operator defined by the population operator, that is,

$$\frac{1}{2} \left\| k(\phi(s,a)) \right\|_{\Gamma_p^{-1}[\phi, \rho_{h,i}^n, \lambda, n]} \le \left\| k(\phi(s,a)) \right\|_{\Gamma_e^{-1}[\phi, \mathcal{D}_{h,i}^n, \lambda]} \le 2 \left\| k(\phi(s,a)) \right\|_{\Gamma_p^{-1}[\phi, \rho_{h,i}^n, \lambda, n]} \quad \text{(E.5)}$$

for any $\phi \in \Phi$, $(s,a) \in \mathcal{S} \times \mathcal{A}$, $i \in \{0, 1, 2\}$, and $(h, n) \in [H] \times [N]$. The data we collected contains enough information for us to design the bonus for exploration when $\mathcal{E}_1$ holds. We denote by $\mathcal{E}_2$ that

$$\mathbb{E}_{(s,a) \sim \rho_{h,i}^n} \left[ \text{TV}[\widehat{\mathcal{P}}_h^n(\cdot \mid s,a), \mathcal{P}_h^*(\cdot \mid s,a)]^2 \right] \le \zeta^2/n, \qquad \text{(E.6)}$$

$$\mathbb{E}_{(s,a) \sim \rho_{h,i}^n} \left[ |\widehat{r}_h^n(s,a) - r_h^*(s,a)|^2 \right] \le \zeta^2/n \qquad \text{(E.7)}$$

for all $(h, n) \in [H] \times [N]$, $i \in \{1, 2\}$. Here $\text{TV}[p_1(\cdot), p_2(\cdot)]$ is the total variance divergence of two distributions. Our estimators of the model are accurate in the sense that the population risk of our estimators are small when $\mathcal{E}_2$ holds. The following lemmas show that $\mathcal{E}_1$ and $\mathcal{E}_2$ hold with high probability.

**Lemma E.2** (Concentration of Inverse Covariance). *Under Assumption 5.2, when we set* $\lambda = C_1 m N^{\gamma/(1+\gamma)} \log(48HRN/\delta)$, *Event* $\mathcal{E}_1$ *defined in* (E.5) *holds with probability at least* $1 - \delta/4$. *Here* $C_1$ *is a constant only depends on the parameters of the NN class.*

*Proof.* See Appendix §F for a detailed proof. $\qquad \square$

**Lemma E.3** (Estimation Error). *We assume that Assumption 5.2 holds with $\gamma < 2$. Event $\mathcal{E}_2$ defined in (E.6) and (E.7) holds with probability at least $1 - \delta/5$ when*

$$\zeta = \sqrt{C_2 \log(20HRN/\delta) N^{\gamma/(1+\gamma)}}.$$

*Here $C_2$ is a constant depends on the regularity parameters and the parameters of the NN class.*

*Proof.* See Appendix §G for a detailed proof. □

Lemmas E.2 and E.3 show that good events $\mathcal{E}_1$ and $\mathcal{E}_2$ hold with high probability. In the following part of the proof, we condition on Event $\mathcal{E}_1$ and $\mathcal{E}_2$. We also define the effective dimension as follows.

**Definition E.4** (Effective Dimension). *Let $\mathcal{H}$ be an RKHS and $K$ be the corresponding kernel. For $\mathcal{X}_n = \{x_1, \ldots, x_n\}$, we define the matrix $K[\mathcal{X}_n] = [K(x_{\tau_1}, x_{\tau_2})]_{\tau_1, \tau_2 = 1}^n$, and define*

$$\Lambda_1(n, \lambda_0) = \sup_{\mathcal{X}_n \subset \mathcal{X}} \log \det(I_n + K[\mathcal{X}_n]/\lambda_0).$$

*Let $k$ be the feature map of $\mathcal{H}$, and $\Delta(\mathcal{X})$ be the set of distributions over $\mathcal{X}$, we define*

$$\Lambda_2(n, \lambda_0) = \sup_{\rho \in \Delta(\mathcal{X})} n \mathbb{E}_{x \sim \rho} \Big[ \|k(x)\|_{\Gamma_p^{-1}[\rho, \lambda_0, n]}^2 \Big].$$

*Here the population operator $\Gamma_p : \mathcal{H} \to \mathcal{H}$ is defined as*

$$\Gamma_p[\rho, \lambda_0, n] f(x) = \lambda_0 f(x) + n \mathbb{E}_{x_0 \sim \rho} \big[ K(x, x_0) f(x_0) \big].$$

*We define $d_{\text{eff}} = \max\{\Lambda_1(N + 1, \lambda), \Lambda_2(N, \lambda)\}$.*

The effective dimension $\Lambda_1$ is the maximum information gain in Srinivas et al. (2009), which is closely related to Gaussian process regression. The effective dimension $\Lambda_2$ is also closely related to the dimension in finite-dimension RKHSs. See Appendix §H for a brief discussion. The effective dimension $d_{\text{eff}}$ is closely related to the sample complexity.

We are now ready to present the proof of Theorem 5.3. Our goal is to bound $J(\pi^*; r^*, \mathcal{P}^*) - J(\widehat{\pi}; r^*, \mathcal{P}^*)$. Since the policy $\widehat{\pi}$ returned by Algorithm 2 (ELNF) is the mixture of $\pi^1, \ldots, \pi^N$, we have

$$J(\pi^*; r^*, \mathcal{P}^*) - J(\widehat{\pi}; r^*, \mathcal{P}^*) = \frac{1}{N} \sum_{n=1}^N \big[ J(\pi^*; r^*, \mathcal{P}^*) - J(\pi^{n+1}; r^*, \mathcal{P}^*) \big]. \tag{E.8}$$

We decompose the suboptimality as

$$\sum_{n=1}^N \big[ J(\pi^*; r^*, \mathcal{P}^*) - J(\widehat{\pi}; r^*, \mathcal{P}^*) \big] = \underbrace{\sum_{n=1}^N z_{n,1}}_{\text{Term (a)}} + \underbrace{\sum_{n=1}^N z_{n,2}}_{\text{Term (b)}} + \underbrace{\sum_{n=1}^N z_{n,3}}_{\text{Term (c)}}, \tag{E.9}$$

where $u^n = \{u_h^n\}_{h=1}^H$ is the bonus we define in (4.5) and $\{z_{n,j}\}_{(n,h) \in [N] \times [3]}$ are defined as

$$z_{n,1} = J(\pi^*; r^*, \mathcal{P}^*) - J(\pi^*; r^n + u^n, \mathcal{P}^n),$$
$$z_{n,2} = J(\pi^*; r^n + u^n, \mathcal{P}^n) - J(\pi^{n+1}; r^n + u^n, \mathcal{P}^n),$$
$$z_{n,3} = J(\pi^{n+1}; r^n + u^n, \mathcal{P}^n) - J(\pi^{n+1}; r^*, \mathcal{P}^*).$$

In the sequel, we bound the terms from the above in (E.9) separately.

**Term (a).** To bound Term (a), we introduce the following characterization of the bonus.

**Lemma E.5** (Almost Optimistic for the Planning Phase). *Following the same condition of Theorem 5.3, when condition on the good events $\mathcal{E}_1, \mathcal{E}_2$ defined in (E.5), (E.6) and (E.7), we have*

$$J(\pi; \widehat{r}^n + u^n, \mathcal{P}^n) - J(\pi; r, \mathcal{P}^*) \geq -(H + 1)|\mathcal{A}|\zeta/\sqrt{n}.$$

*holds for any policy $\pi$. Here the bonus $u^n = \{u_h^n\}_{h=1}^H$ is defined in (4.5).*

*Proof.* See Appendix §I.1 for a detailed proof. □

By Lemma E.5, we have Term (a) $\leq \sum_{n=1}^{N}(H+1)|\mathcal{A}|\zeta/\sqrt{n} \leq 8H|\mathcal{A}|\zeta\sqrt{N}$.

**Term (b).** The following lemma bounds Term (b) from the above.

**Lemma E.6.** *For the policy $\pi^{n+1}$ returned by Algorithm 4 (Planning Algorithm), we have*
$$J(\pi^*; r^n + u^n, \mathcal{P}^n) - J(\pi^{n+1}; r^n + u^n, \mathcal{P}^n) \leq 0.$$

*Proof.* See Appendix §I.2 for a detailed proof. □

By Lemma E.6, we have Term (b) $= \sum_{n=1}^{N}[J(\pi^*; r^n + u^n, \mathcal{P}^n) - J(\pi^{n+1}; r^n + u^n, \mathcal{P}^n)] \leq 0$.

**Term (c).** We have the following lemma, which bounds Term (c) from the above.

**Lemma E.7** (Bounded optimistism). *Following the same condition of Theorem 5.3, we have*
$$\sum_{n=1}^{N}\left[J(\pi^{n+1}; r^n + u^n, \mathcal{P}^n) - J(\pi^{n+1}; r^*, \mathcal{P}^*)\right] \leq 46H^2|\mathcal{A}|\zeta\beta_1\sqrt{d_{\mathrm{eff}}N}\log(10H/\delta)$$

*with probability at least $1 - \delta/2$. Here $\beta_1 = (4H^2 + 6H + 2)\sqrt{4\lambda v^2/c_{\min}^2 + 4r_{\max}^2|\mathcal{A}|\beta^2\zeta^2 d_{\mathrm{eff}}}$ and $\zeta$ is defined in Lemma E.3.*

*Proof.* See Appendix §I.3 for a detailed proof. □

We denote by $\mathcal{E}_3$ the event defined by Lemma E.7. In the following part of the proof, we condition on Event $\mathcal{E}_3$. By Lemma E.7, we have Term (c) $\leq 46H^2|\mathcal{A}|\zeta\beta_1\sqrt{d_{\mathrm{eff}}N}\log(10H/\delta)$. Combining the upper bounds of the terms in (E.9), we have
$$\sum_{n=1}^{N}\left[J(\pi^*; r^*, \mathcal{P}^*) - J(\pi^{n+1}; r^*, \mathcal{P}^*)\right] \leq H|\mathcal{A}|\zeta\sqrt{N}\left(8 + 46H\beta_1\sqrt{d_{\mathrm{eff}}}\log(10H/\delta)\right), \quad \text{(E.10)}$$

By the definition of $\beta_1$ in Lemma E.7, we have $\beta_1 \leq 24H^2 v c_{\max}/c_{\min}^2(\sqrt{\lambda} + \zeta\beta\sqrt{d_{\mathrm{eff}}|\mathcal{A}|})$. By the definition of $\beta$ in Theorem 5.3, we have $\beta \leq 24H v c_{\max}/c_{\min}^2(\sqrt{\lambda} + \zeta\sqrt{|\mathcal{A}|})$. Therefore, we have
$$\beta_1 \leq 600H^3\zeta v^2 c_{\max}^2/c_{\min}^4\sqrt{d_{\mathrm{eff}}|\mathcal{A}|}\left(\sqrt{\lambda} + \sqrt{|\mathcal{A}|}\right) \quad \text{(E.11)}$$

Combining (E.10), (E.11) with the value of $\lambda$ in Theorem 5.3, we have
$$\sum_{n=1}^{N}\left[J(\pi^*; r^*, \mathcal{P}^*) - J(\pi^{n+1}; r^*, \mathcal{P}^*)\right] \leq 36000H^5 v^2 c_{\max}^2|\mathcal{A}|^2\zeta^2 d_{\mathrm{eff}}\sqrt{\lambda N}\log(10H/\delta)/c_{\min}^4$$
$$\leq C_3 H^5|\mathcal{A}|^2 d_{\mathrm{eff}}N^{(1+4\gamma)/(2+2\gamma)}\left(\log(48HRN/\delta)\right)^{5/2} \quad \text{(E.12)}$$

where $C_3$ is a constant that only depends on the dimension of the feature $m$, the bound of the feature maps $R$, and $C_{\mathrm{net}}$ in Assumption 5.2. The following lemma bounds $d_{\mathrm{eff}}$ from the above.

**Lemma E.8.** *For the Gaussian kernel $K(x_1, x_2) = \exp(-\|x_1 - x_2\|_2^2/2)$, we have*
$$d_{\mathrm{eff}} \leq C_4(\log N)^{m+1}.$$

*Here $C_4$ is a constant that only depends on the dimension $m$ and the radius $R$.*

*Proof.* See Appendix §H.1 for a detailed proof. □

Combining (E.8), (E.12) with Lemma E.8, we have
$$J(\pi^*; r^*, \mathcal{P}^*) - J(\widehat{\pi}; r^*, \mathcal{P}^*) \leq C_5 H^5|\mathcal{A}|^2 N^{(2\gamma-1)/(2+2\gamma)}\left(\log(48HRN/\delta)\right)^{5/2}(\log N)^{m+1},$$

where $C_5$ is a constant that only depends on the dimension of the feature $m$, the bound of the feature maps $R$, and $C_{\mathrm{net}}$ in Assumption 5.2. Thus, we conclude the proof of Theorem 5.3. □

## F    CONCENTRATION OF THE INVERSE COVARIANCE

In this section, we provide the proof of Lemma E.2, which shows that Event $\mathcal{E}_1$ defined in (E.5) happens with high probability. We first show that we can prove Lemma E.2 by the concentration of covariance using Lemma F.2. We then prove the concentration of the covariance for a fixed feature map using the concentration inequality in Lemma F.3. Next, we take a union bound to prove the uniform concentration for a covering of the feature map class, and use the property of the covering to prove the uniform concentration of the covariance for the whole feature map class, which concludes the proof of Lemma F.2.

*Proof.* We first introduce the filtration we use for our analysis.

**Definition F.1** (Filtration). *For any $n \in [N]$, we define $\mathcal{F}_n$ as the $\sigma$-algebra generated by the trajectories in the first $n$ loops of Algorithm 2 (*`ELNF`*).*

By taking a union bound, we only need to show that for a fix $n \in [N]$ and $i \in \{0, 1, 2\}$, we have

$$\left\| k\big(\phi(s,a)\big) \right\|^2_{\Gamma_p^{-1}[\phi, \rho_{h,i}^n, \lambda, n]} / 4 \leq \left\| k\big(\phi(s,a)\big) \right\|^2_{\Gamma_e^{-1}[\phi, \mathcal{D}_{h,i}^n, \lambda]} \leq 4 \left\| k\big(\phi(s,a)\big) \right\|^2_{\Gamma_p^{-1}[\phi, \rho_{h,i}^n, \lambda, n]}$$

for any $\phi \in \Phi$ and $(s,a) \in \mathcal{S} \times \mathcal{A}$ with probability $1 - \delta/(24HN)$. Since we have $\|\phi(s,a)\|_2 \leq R$, it remains to show that with probability $1 - \delta/(24HN)$, we have

$$\left\| k(x) \right\|^2_{\Gamma_p^{-1}[\phi, \rho_{h,i}^n, \lambda, n]} / 4 \leq \left\| k(x) \right\|^2_{\Gamma_e^{-1}[\phi, \mathcal{D}_{h,i}^n, \lambda]} \leq 4 \left\| k(x) \right\|^2_{\Gamma_p^{-1}[\phi, \rho_{h,i}^n, \lambda, n]}$$

for any $x \in \mathbb{R}^m$ with $\|x\|_2 \leq R$. We first prove that $\|k(x)\|^2_{\Gamma_e^{-1}[\phi, \mathcal{D}_{h,i}^n, \lambda]} \leq 4 \|k(x)\|^2_{\Gamma_p^{-1}[\phi, \rho_{h,i}^n, \lambda, n]}$ for any $x \in \mathbb{R}^m$ with $\|x\|_2 \leq R$ with probability at least $1 - \delta/(48HN)$. The following lemma allows us to prove the concentration of the inverse covariance by the concentration of the covariance.

**Lemma F.2.** *Let $\mathcal{H}$ be a Hilbert space, and $A$, $B$ be two positive-definite and self-adjoint bounded linear operators on $\mathcal{H}$. Suppose $\langle x, Ax \rangle_{\mathcal{H}} \geq \langle x, Bx \rangle_{\mathcal{H}}$ for all $x \in \mathcal{D}$, we have $\langle x, A^{-1}x \rangle_{\mathcal{H}} \leq \langle x, B^{-1}x \rangle_{\mathcal{H}}$ when $B^{-1/2}C^{-1}B^{-1/2}x \in \mathcal{D}$. Here $C = (B^{-1/2}AB^{-1/2})^{1/2}$.*

*Proof.* See Appendix §J.8 for a detailed proof.  □

For simplicity, we define $\Gamma_1 = \Gamma_p[\phi, \rho_{h,i}^n, \lambda, n]$, $\Gamma_2 = \Gamma_e[\phi, \mathcal{D}_{h,i}^n, \lambda]$,

$$\mathcal{H}_0 = \left\{ k(x) \mid x \in \mathbb{R}^m, \text{ and } \|x\|_2 \leq R \right\}, \text{ and } \mathcal{H}_1 = \left\{ \Gamma k(x) \mid x \in \mathbb{R}^m, \text{ and } \|x\|_2 \leq R \right\}, \quad \text{(F.1)}$$

where $\Gamma = \Gamma_1^{-1/2}(\Gamma_1^{-1/2}\Gamma_2\Gamma_1^{-1/2})^{-1/2}\Gamma_1^{-1/2}$. By Lemma F.2, it remains to show that $\|g\|^2_{\Gamma_1} \leq 4\|g\|^2_{\Gamma_2}$ for any $g \in \mathcal{H}_1$ with probability at least $1 - \delta/(48HN)$. By the definition of the population operator, we have

$$\|g\|^2_{\Gamma_1} = \|g\|^2_{\Gamma_p[\phi, \rho_{h,i}^n, \lambda, n]} = \lambda \|g\|^2_{\mathcal{H}} + \sum_{\tau=1}^{n} \mathbb{E}_{(s,a) \sim \varrho_{h,i}^\tau} \left[ g^2\big(\phi(s,a)\big) \right].$$

Similarly, we have $\|g\|^2_{\Gamma_2} = \lambda \|g\|^2_{\mathcal{H}} + \sum_{\tau=1}^{n} g^2(\phi(s_{h,i}^\tau, a_{h,i}^\tau))$. Therefore, we can prove $\|g\|^2_{\Gamma_1} \leq 4\|g\|^2_{\Gamma_2}$ by the concentration of $g^2$. We define

$$\mathcal{N}_1 = \left| \mathcal{C}\big(\mathcal{H}_1, \epsilon_1, \|\cdot\|_\infty\big) \right|, \qquad \mathcal{N}_2 = \left| \mathcal{C}\big(\Phi, \epsilon_2, \|\cdot\|_{\infty,2}\big) \right|, \quad \text{(F.2)}$$

for simplicity. Here $\mathcal{C}$ denote the covering sets we defined in notations, $\epsilon_1$ and $\epsilon_2$ are tuning parameters. By Bernstein inequality, we have the following lemma, which can be used to show the concentration of $g^2$.

**Lemma F.3.** *Suppose that $\{\mathcal{F}_\tau\}_{\tau=0}^n$ is a filtration and $\{(s^\tau, a^\tau)\}_{\tau=1}^n$ is a $\mathcal{S} \times \mathcal{A}$-value stochastic process adapted to this filtration. We denote by $\varrho^\tau$ the distribution of $(s^\tau, a^\tau)$ when condition on $\mathcal{F}_{\tau-1}$. For any fix $g \in \mathcal{H}$ and $\phi \in \Phi$, The following inequality holds with probability at least $1 - \delta/2$.*

$$\sum_{\tau=1}^n g^2\big(\phi(s^\tau, a^\tau)\big) \le 2\sum_{\tau=1}^n \mathbb{E}_{(s,a)\sim\varrho^\tau}\Big[g^2\big(\phi(s,a)\big)\Big] + 2\log(2/\delta)\langle g, g\rangle_{\mathcal{H}}.$$

*We also have the following inequality with probability at least $1 - \delta/2$.*

$$\sum_{\tau=1}^n \mathbb{E}_{(s,a)\sim\varrho^\tau}\Big[g^2\big(\phi(s,a)\big)\Big]/2 \le \sum_{\tau=1}^n g^2\big(\phi(s^n, a^n)\big) + 2\log(2/\delta)\langle g, g\rangle_{\mathcal{H}}.$$

*Proof.* See Appendix §J.9 for a detailed proof. $\qquad\square$

For any $g \in \mathcal{C}(\mathcal{H}_1, \epsilon_1, \|\cdot\|_\infty)$ and $\phi \in \mathcal{C}(\Phi, \epsilon_2, \|\cdot\|_{\infty,2})$, by Lemma F.3, we have

$$\sum_{\tau=1}^n \mathbb{E}_{(s,a)\sim\varrho_{h,i}^\tau}\Big[g^2\big(\phi(s,a)\big)\Big]/2 \le \sum_{\tau=1}^n g^2\big(\phi(s_{h,i}^\tau, a_{h,i}^\tau)\big) + 2\log(48HN\mathcal{N}_1\mathcal{N}_2/\delta)\langle g, g\rangle_{\mathcal{H}} \quad \text{(F.3)}$$

with probability at least $1 - \delta/(48HN\mathcal{N}_1\mathcal{N}_2)$. Here $\mathcal{N}_1$ and $\mathcal{N}_2$ are defined in (F.2), $\varrho_{h,i}^\tau$ is the distribution of $(s_{h,i}^\tau, a_{h,i}^\tau)$ when condition on $\mathcal{F}_{\tau-1}$. By taking a union bound, we have (F.3) holds for all $g \in \mathcal{C}(\mathcal{H}_1, \epsilon_1, \|\cdot\|_\infty)$ and $\phi \in \mathcal{C}(\Phi, \epsilon_2, \|\cdot\|_{\infty,2})$ with probability at least $1 - \delta/(48HN)$. Therefore, we have $\mathbb{P}(\mathcal{E}_{n,h,i}) \ge 1 - \delta/(48HN)$ when we define $\mathcal{E}_{n,h,i}$ as the event that (F.3) holds for all $g \in \mathcal{C}(\mathcal{H}_1, \epsilon_1, \|\cdot\|_\infty)$ and $\phi \in \mathcal{C}(\Phi, \epsilon_2, \|\cdot\|_{\infty,2})$.

In the following part of the proof, we condition on Event $\mathcal{E}_{n,h,i}$. For an arbitrary $g \in \mathcal{H}_1$ and $\phi \in \Phi$, we choose $g_0 \in \mathcal{C}(\mathcal{H}_1, \epsilon_1, \|\cdot\|_\infty)$ and $\phi_0 \in \mathcal{C}(\Phi, \epsilon_2, \|\cdot\|_{\infty,2})$ such that $\sup_{x\in\mathcal{X}} |g(x) - g_0(x)| \le \epsilon_1$ and $\sup_{(s,a)\in\mathcal{S}\times\mathcal{A}} \|\phi(s,a) - \phi_0(s,a)\|_2 \le \epsilon_2$. We decompose the difference in the expectation by

$$\sum_{\tau=1}^n \mathbb{E}_{(s,a)\sim\varrho_{h,i}^\tau}\Big[g_0^2\big(\phi_0(s,a)\big)\Big] - \sum_{\tau=1}^n \mathbb{E}_{(s,a)\sim\varrho_{h,i}^\tau}\Big[g^2\big(\phi(s,a)\big)\Big] \quad \text{(F.4)}$$

$$= \sum_{\tau=1}^n \mathbb{E}_{(s,a)\sim\varrho_{h,i}^\tau}\Big[\Big(g_0\big(\phi_0(s,a)\big) + g\big(\phi(s,a)\big)\Big)\Big(g_0\big(\phi_0(s,a)\big) - g\big(\phi(s,a)\big)\Big)\Big].$$

Since $\sup_{x\in\mathcal{X}} |g(x) - g_0(x)| \le \epsilon_1$, we have

$$\Big|g_0\big(\phi_0(s,a)\big) - g\big(\phi(s,a)\big)\Big| \le \epsilon_1 + \Big|g\big(\phi_0(s,a)\big) - g\big(\phi(s,a)\big)\Big| \quad \text{(F.5)}$$

for any $(s,a) \in \mathcal{S} \times \mathcal{A}$. Combining (F.5) with the reproducing property of $\mathcal{H}$, we have

$$\Big|g_0\big(\phi_0(s,a)\big) - g\big(\phi(s,a)\big)\Big| \le \epsilon_1 + \Big|\Big\langle g, k\big(\phi_0(s,a)\big) - k\big(\phi(s,a)\big)\Big\rangle_{\mathcal{H}}\Big| \quad \text{(F.6)}$$

$$\le \epsilon_1 + \|g\|_{\mathcal{H}} \Big\|k\big(\phi_0(s,a)\big) - k\big(\phi(s,a)\big)\Big\|_{\mathcal{H}},$$

where the last inequality follows Cauchy-Schwarz inequality. For the kernel feature map $k$, we have

$$\Big\|k\big(\phi_0(s,a)\big) - k\big(\phi(s,a)\big)\Big\|_{\mathcal{H}}^2 = \big\|k\big(\phi_0(s,a)\big)\big\|_{\mathcal{H}}^2 + \big\|k\big(\phi(s,a)\big)\big\|_{\mathcal{H}}^2 - 2\Big\langle k\big(\phi_0(s,a)\big), k\big(\phi(s,a)\big)\Big\rangle_{\mathcal{H}}$$

$$= 2\Big(1 - \exp\big(-\big\|\phi_0(s,a) - \phi(s,a)\big\|^2/2\big)\Big) \le \big\|\phi_0(s,a) - \phi(s,a)\big\|^2 \le \epsilon_2^2. \quad \text{(F.7)}$$

Combining (F.6) with (F.7), we have $|g_0(\phi_0(s,a)) - g(\phi(s,a))| \le \epsilon_1 + \epsilon_2\|g\|_{\mathcal{H}}$, and

$$\Big|g_0\big(\phi_0(s,a)\big) + g\big(\phi(s,a)\big)\Big| \le \epsilon_1 + \epsilon_2\|g\|_{\mathcal{H}} + 2\|g\|_{\mathcal{H}}. \quad \text{(F.8)}$$

We plug (F.8) into (F.4) and have

$$\sum_{\tau=1}^n \mathbb{E}_{(s,a)\sim\varrho_{h,i}^\tau}\Big[g^2\big(\phi(s,a)\big)\Big] - \sum_{\tau=1}^n \mathbb{E}_{(s,a)\sim\varrho_{h,i}^\tau}\Big[g_0^2\big(\phi_0(s,a)\big)\Big] \quad \text{(F.9)}$$

$$\le n\Big[(\epsilon_2^2 + 2\epsilon_2)\langle g, g\rangle_{\mathcal{H}} + (2\epsilon_1\epsilon_2 + 2\epsilon_1)\|g\|_{\mathcal{H}} + \epsilon_1^2\Big].$$

The following lemma provide an upper bound and a lower bound of $\langle g, g\rangle_{\mathcal{H}}$ for $g \in \mathcal{H}_1$.

**Lemma F.4.** *For any $g \in \mathcal{H}$ and the operator $\Gamma$ defined in (F.1), we have*

$$\lambda \langle g, g \rangle_{\mathcal{H}}/(\lambda + n)^3 \leq \langle \Gamma g, \Gamma g \rangle_{\mathcal{H}} \leq (\lambda + n) \langle g, g \rangle_{\mathcal{H}}/\lambda^3.$$

*Proof.* See Appendix §J.10 for a detailed proof. □

Combining the definition of $\mathcal{H}_1$ in (F.1) with Lemma F.4, we have

$$\langle g, g \rangle_{\mathcal{H}} = \langle \Gamma k(x), \Gamma k(x) \rangle_{\mathcal{H}} \geq \lambda \langle k(x), k(x) \rangle_{\mathcal{H}}/(\lambda + n)^3 = \lambda/(\lambda + n)^3, \tag{F.10}$$

where the operator $\Gamma$ is defined in Lemma F.4. Therefore, we plug (F.10) into (F.9) and have

$$\sum_{\tau=1}^{n} \mathbb{E}_{(s,a) \sim \varrho_{h,i}^{\tau}} \Big[ g^2 \big( \phi(s, a) \big) \Big] - \sum_{\tau=1}^{n} \mathbb{E}_{(s,a) \sim \varrho_{h,i}^{\tau}} \Big[ g_0^2 \big( \phi_0(s, a) \big) \Big] \tag{F.11}$$

$$\leq n \big[ (\epsilon_2^2 + 2\epsilon_2) + (\lambda + n)^{3/2} (2\epsilon_1 \epsilon_2 + 2\epsilon_1)/\lambda^{1/2} + \epsilon_1^2 (\lambda + n)^3/\lambda \big] \langle g, g \rangle_{\mathcal{H}}.$$

By the same method that induces (F.11), we have

$$\sum_{\tau=1}^{n} g_0^2 \big( \phi_0(s_{h,i}^{\tau}, a_{h,i}^{\tau}) \big) - \sum_{\tau=1}^{n} g^2 \big( \phi(s_{h,i}^{\tau}, a_{h,i}^{\tau}) \big) \tag{F.12}$$

$$\leq n \big[ (\epsilon_2^2 + 2\epsilon_2) + (\lambda + n)^{3/2} (2\epsilon_1 \epsilon_2 + 2\epsilon_1)/\lambda^{1/2} + \epsilon_1^2 (\lambda + n)^3/\lambda \big] \langle g, g \rangle_{\mathcal{H}}.$$

By the definition of Event $\mathcal{E}_{n,h,i}$ in (F.3), we have

$$\sum_{\tau=1}^{n} \mathbb{E}_{(s,a) \sim \varrho_{h,i}^{\tau}} \Big[ g_0^2 \big( \phi(s, a) \big) \Big] \Big/ 2 \leq \sum_{\tau=1}^{n} g_0^2 \big( \phi(s_{h,i}^{\tau}, a_{h,i}^{\tau}) \big) + 2 \log(48 H N \mathcal{N}_1 \mathcal{N}_2/\delta) \tag{F.13}$$

for $g_0 \in \mathcal{C}(\mathcal{H}_1, \epsilon_1, \|\cdot\|_\infty)$ when $\mathcal{E}_{n,h,i}$ holds. We plug (F.11), (F.12) into (F.13), and have

$$\sum_{\tau=1}^{n} \mathbb{E}_{(s,a) \sim \varrho_{h,i}^{\tau}} \Big[ g^2 \big( \phi(s, a) \big) \Big] \Big/ 2 \tag{F.14}$$

$$\leq \sum_{\tau=1}^{n} g^2 \big( \phi(s_{h,i}^{\tau}, a_{h,i}^{\tau}) \big) + \big( 2 \log(48 H N \mathcal{N}_1 \mathcal{N}_2/\delta) + \kappa[n, \epsilon_1, \epsilon_2] \big) \langle g, g \rangle_{\mathcal{H}}$$

when condition on Event $\mathcal{E}_{n,h,i}$ defined in (F.13). Here $\kappa[n, \epsilon_1, \epsilon_2]$ is defined as

$$\kappa[n, \epsilon_1, \epsilon_2] = 3n \big[ (\epsilon_2^2 + 2\epsilon_2) + (\lambda + n)^{3/2} (2\epsilon_1 \epsilon_2 + 2\epsilon_1)/\lambda^{1/2} + \epsilon_1^2 (\lambda + n)^3/\lambda \big]/2 \tag{F.15}$$

By the definition of $\kappa[n, \epsilon_1, \epsilon_2]$ in (F.15), we have

$$\kappa[n, \epsilon_1, \epsilon_2] \leq 5 n^{\gamma/(1+\gamma)} + 6(\lambda + n)^{3/2}/(n^{3/2} \lambda^{3/2}) + 2(\lambda + n)^3/(n^4 \lambda^3) \leq 38 n^{\gamma/(1+\gamma)} \tag{F.16}$$

when we set $\epsilon_1 = 1/(n^{5/2} \lambda)$ and $\epsilon_2 = n^{-1/(1+\gamma)}$. Combining (F.14) with (F.16), we have

$$\sum_{\tau=1}^{n} \mathbb{E}_{(s,a) \sim \varrho_{h,i}^{\tau}} \Big[ g^2 \big( \phi(s, a) \big) \Big] \Big/ 2 \tag{F.17}$$

$$\leq \sum_{\tau=1}^{n} g^2 \big( \phi(s_{h,i}^{\tau}, a_{h,i}^{\tau}) \big) + \big( 2 \log \mathcal{N}_1 + 2 \log \mathcal{N}_2 + 2 \log(48 H N/\delta) + 38 n^{\gamma/(1+\gamma)} \big) \langle g, g \rangle_{\mathcal{H}}.$$

It remains to bound $\log \mathcal{N}_1$ and $\log \mathcal{N}_2$ from the above. The following lemma bounds $\mathcal{N}_1$ from the above.

**Lemma F.5.** *For the set $\mathcal{H}_1$ defined in (F.1), we have $\mathcal{N}(\mathcal{H}_1, \epsilon, \|\cdot\|_\infty) \leq (R^2(\lambda + n)/(\lambda^3 \epsilon^2))^{m/2}$.*

*Proof.* See Appendix §J.11 for a detailed proof. □

By the definition of $\mathcal{N}_1$ in (F.2) and Lemma F.5, we have $2 \log \mathcal{N}_1 \leq 12m \log(nR)$. By the definition of $\mathcal{N}_1$ in (F.2) and Assumption 5.2, we have $2 \log \mathcal{N}_2 \leq C_{\mathrm{cic},1} n^{\gamma/(1+\gamma)} \log n$, where $C_{\mathrm{cic},1}$ is a constant only depends on $C_{\mathrm{net}}$ in Assumption 5.2. Therefore, by (F.17), we have

$$\sum_{\tau=1}^{n} \mathbb{E}_{(s,a)\sim\varrho_{h,i}^{\tau}}\Big[g^2\big(\phi(s,a)\big)\Big]/2$$

$$\leq \sum_{\tau=1}^{n} g^2\big(\phi(s_{h,i}^{\tau}, a_{h,i}^{\tau})\big) + 53 C_{\mathrm{cic},1} m N^{\gamma/(1+\gamma)} \log(48HNR/\delta)\langle g,g\rangle_{\mathcal{H}}$$

when condition on $\mathcal{E}_{n,h,i}$ in (F.3). By choosing $\lambda = 106 C_{\mathrm{cic},1} m N^{\gamma/(1+\gamma)} \log(48HNR/\delta)$, we have

$$\sum_{\tau=1}^{n} \mathbb{E}_{(s,a)\sim\varrho_{h,i}^{\tau}}\Big[g^2\big(\phi(s,a)\big)\Big]/2 + \lambda\langle g,g\rangle_{\mathcal{H}}/2 \leq \sum_{\tau=1}^{n} g^2\big(\phi(s_{h,i}^{\tau}, a_{h,i}^{\tau})\big) + \lambda\langle g,g\rangle_{\mathcal{H}} \qquad \text{(F.18)}$$

for any $g \in \mathcal{H}_1$ and $\phi \in \Phi$ when condition on $\mathcal{E}_{n,h,i}$. By the same method inducing (F.18), we have

$$\sum_{\tau=1}^{n} g^2\big(\phi(s_{h,i}^{\tau}, a_{h,i}^{\tau})\big) + \lambda\langle g,g\rangle_{\mathcal{H}} \leq 2\sum_{\tau=1}^{n} \mathbb{E}_{(s,a)\sim\varrho_{h,i}^{\tau}}\Big[g^2\big(\phi(s,a)\big)\Big] + 2\lambda\langle g,g\rangle_{\mathcal{H}}$$

for any $g \in \mathcal{H}_1$ and $\phi \in \Phi$ when condition on $\mathcal{E}_{n,h,i}$, which concludes the proof of Lemma E.2. $\square$

## G  PROOF OF LEMMA E.3

In this section, we provide the proof of Lemma E.3, which shows that Event $\mathcal{E}_2$ defined in (E.6) and (E.7) happens with high probability. We conclude the proof of Lemma E.3 by combining Lemma G.1 and Lemma G.2, which provide upper bounds of the estimation errors of the reward and the transition estimation. Lemma G.1 is proven by the standard technique of bounding generalization error. Lemma G.2 is proven by the same method of Theorem 7.4 in Van de Geer (2000). To prove Lemma G.2, we show that the estimation error in the transition is closely related to the uniform convergence over a function class induced by the transition kernel, and then prove the uniform convergence over the function class using empirical process theory.

*Proof.* The following lemma bounds the population risk of the estimators of the reward from the above.

**Lemma G.1.** *Let $\{\mathcal{F}_\tau\}_{\tau=0}^{n}$ be a filtration and $\{x_\tau\}_{\tau=1}^{n}$ be a $\mathcal{X}$-valued stochastic process adapted to this filtration. Suppose $r_\tau$ is a $\mathcal{F}_{\tau+1}$-measurable random variable with $\mathbb{E}[r_\tau \mid \mathcal{F}_\tau] = r^*(x_\tau)$, where $r^* \in \mathcal{R}$ is an unknown function and $\mathcal{R}$ is a known function class. We define the estimator of $r^*$ as*

$$\widehat{r} = \operatorname*{argmin}_{r\in\mathcal{R}} \sum_{\tau=1}^{n} \big(r_\tau - r(x_\tau)\big)^2.$$

*We also define the population risk as $\mathrm{Risk}(r) = \sum_{\tau=1}^{n} \mathbb{E}[(r(x_\tau) - r^*(x_\tau))^2 \mid \mathcal{F}_{\tau-1}]/n$, and then have*

$$\mathrm{Risk}(\widehat{r}) \leq 16 \log\big(\mathcal{N}(\mathcal{R}, \epsilon, \|\cdot\|_\infty)/\delta\big)/n + 12\epsilon$$

*with probability at least $1 - \delta$ for any fix $\epsilon, \delta > 0$.*

*Proof.* See Appendix §G.1 for a detailed proof. $\square$

We first apply Lemma G.1 with $\epsilon = 1/n^{-1/(1+\gamma)}$, we then have

$$\mathbb{E}_{(s,a)\sim\rho_{h,1}^{n}}\Big[|\widehat{r}_h^n(s,a) - r_h^*(s,a)|^2\Big] + \mathbb{E}_{(s,a)\sim\rho_{h,2}^{n}}\Big[|\widehat{r}_h^n(s,a) - r_h^*(s,a)|^2\Big]$$

$$\leq 24 n^{-1/(1+\gamma)} + 32\big(\log(20HN/\delta) + C_{\mathrm{net}} n^{\gamma/(1+\gamma)} \log(n)/(1+\gamma)\big)/n$$

$$\leq C_{\mathrm{net},1} n^{-1/(1+\gamma)} \log(20HN/\delta)$$

for any $(h, n) \in [H] \times [N]$ with probability at least $1 - \delta/(20HN)$ by Assumption 5.2. Here $C_{\text{net},1} = 56 + 32C_{\text{net}}/(1+\gamma)$ is a constant only depends on the parameter of the neural network classes. By taking a union bound, we have

$$\mathbb{E}_{(s,a) \sim \rho_{h,i}^n} \left[ \left| \widehat{r}_h^{n+1}(s,a) - r_h^*(s,a) \right|^2 \right] \leq C_{\text{net},1} n^{-1/(1+\gamma)} \log(20HN/\delta)$$

holds for all $(i, h, n) \in [2] \times [H] \times [N]$ with probability at least $1 - \delta/20$.

The following lemma bounds the population risk of the estimators of the transition kernel from the above.

**Lemma G.2.** *Let* $\{\mathcal{F}_\tau\}_{\tau=0}^n$ *be a filtration and let* $\{(S_\tau, A_\tau, S_\tau')\}_{\tau=1}^n$ *be a* $\mathcal{S} \times \mathcal{A} \times \mathcal{S}$-value stochastic process adapted to this filtration. Let $\rho_\tau$ be the distribution of $(S_\tau, A_\tau)$ when condition on $\mathcal{F}_{\tau-1}$. We assume that $S_\tau' \sim \mathcal{P}^*(\cdot \mid S_\tau, A_\tau)$ when condition on $(S_\tau, A_\tau)$ and $\mathcal{F}_{\tau-1}$. We estimate $\mathcal{P}^*$ by

$$\widehat{\mathcal{P}} = \underset{\mathcal{P} \in \mathcal{M}}{\operatorname{argmax}} \sum_{\tau=1}^n \log \mathcal{P}(S_\tau' \mid S_\tau, A_\tau).$$

*Let* $\mathcal{N}(\epsilon, \mathcal{M}, \|\cdot\|_\infty)$ *be the covering number of the transition class* $\mathcal{M}$, *and define* $H_\infty(\epsilon, \mathcal{M}) = \log \mathcal{N}(\epsilon, \mathcal{M}, \|\cdot\|_\infty)$. *Let $G$ be a function satisfies (1).* $G(\epsilon)/\epsilon^2$ *is non-increasing, and (2)*

$$G(\epsilon) \geq \max \left\{ 8 \int_{\epsilon^2/2^{17}}^{\epsilon/2^3} H_\infty^{1/2} \big( (2v)^{-1/2} u, \bar{\mathcal{G}} \big) \mathrm{d}u, \epsilon \right\},$$

*where* $\bar{\mathcal{G}} = \{ \bar{g}_\mathcal{P} \mid \bar{g}_\mathcal{P}(s, a, s') = \sqrt{(\mathcal{P}(s' \mid s, a) + \mathcal{P}^*(s' \mid s, a))/2}, \mathcal{P} \in \mathcal{M} \}$. *Suppose* $\epsilon_n$ *satisfies* $\sqrt{n} \epsilon_n^2 \geq C_9 G(\epsilon_n)$, *where* $C_9$ *is an absolute constant. Then for any* $\epsilon \geq \max\{\epsilon_n, 1/\sqrt{n}\}$, *we have*

$$\mathbb{P} \left( \frac{1}{n} \sum_{\tau=1}^n \mathbb{E}_{(s,a) \sim \rho_\tau} \left[ \mathrm{TV}^2 \big( \widehat{\mathcal{P}}(\cdot \mid s, a), \mathcal{P}^*(\cdot \mid s, a) \big) \right] \geq \epsilon^2 \right) \leq C_9 \exp \big( -n\epsilon^2/C_9^2 \big).$$

*Proof.* See Appendix §G.2 for a detailed proof. □

The following lemma bounds the covering number of $\bar{\mathcal{G}}$ in Lemma G.2 from the above.

**Lemma G.3.** *For two feature map classes $\Phi$ and $\Psi$, we define the density class as*

$$\mathcal{M} = \left\{ \mathcal{P} : \mathcal{P}(s' \mid s, a) = \exp \big( -\|\phi(s, a) - \psi(s')\|_2^2/2 \big)/c[\phi, \psi](s, a) \mid \phi \in \Phi, \psi \in \Psi \right\},$$

*where the normalization function $c[\phi, \psi](s, a)$ is defined in (E.1). We then have*

$$H_\infty(\delta, \bar{\mathcal{G}}) \leq H_2 \left( \frac{\exp(-R^2) c_{\min}^2 \delta}{R \sqrt{c_{\max}} (v + c_{\max})}, \Phi, \|\cdot\|_{\infty,2} \right) + H_2 \left( \frac{\exp(-R^2) c_{\min}^2 \delta}{R \sqrt{c_{\max}} (v + c_{\max})}, \Psi, \|\cdot\|_{\infty,2} \right),$$

*where $\bar{\mathcal{G}}$ is defined in Lemma G.2.*

*Proof.* See Appendix §G.3 for a detailed proof. □

By Lemma G.3, when we define $\bar{\mathcal{G}}$ in the same way as Lemma G.2, we have

$$H_\infty(\epsilon, \bar{\mathcal{G}}) \leq H_2(C_{\text{reg},1}\epsilon, \Phi) + H_2(C_{\text{reg},1}\epsilon, \Psi)$$
$$\leq 2C_{\text{net}} \big( 1 - \log C_{\text{reg},1} + \log(1/\epsilon) \big)/(C_{\text{reg},1}^\gamma \epsilon^\gamma) \leq C_{\text{reg},2}^2 \big( 1 + \log(1/\epsilon) \big)/\epsilon^\gamma,$$

where $C_{\text{reg},1} = \exp(-R^2) c_{\min}^2/\big( R\sqrt{c_{\max}}(v + c_{\max}) \big)$, and

$$C_{\text{reg},2} = \sqrt{2C_{\text{net}}(1 - \log C_{\text{reg},1})/C_{\text{reg},1}^\gamma}$$

are constants only depend on the regularity parameters. We have

$$\int_{\epsilon^2/2^{17}}^{\epsilon/8} H_\infty^{1/2}(u, \bar{\mathcal{G}}) \mathrm{d}u \leq \int_{\epsilon^2/2^{17}}^{\epsilon/8} C_{\text{reg},2} \sqrt{1 + \log(1/u)}/u^{\gamma/2} \mathrm{d}u$$
$$\leq C_{\text{reg},2} \sqrt{\log(2^{19}/\epsilon^2)} \epsilon^{1-\gamma/2}/(1 - \gamma/2)$$

when $\gamma < 2$. Therefore, we set $G(\epsilon) = C_{\text{reg},2}\sqrt{\log(2^{15}/\epsilon^2)}\epsilon^{1-\gamma/2}/(1-\gamma/2)$. By such definition, the solution of $\sqrt{n}\epsilon_n^2 = CG(\epsilon_n)$ should satisfy $\epsilon_n \leq C_{\text{reg},3}n^{-1/(2+\gamma)}\log n$, where $C_{\text{reg},3}$ is a constant only depends on the regularity parameters. Therefore, when we choose $\epsilon = C_{\text{reg},4}n^{-1/(2+\gamma)}\sqrt{\log(20HN/\delta)}$, where $C_{\text{reg},4}$ is a constant only depends on the regularity parameters, we have

$$\frac{1}{2}\sum_{i=1}^{2}\mathbb{E}_{(s,a)\sim\rho_{h,i}^{\tau}}\Big[\text{TV}^2\big(\widehat{\mathcal{P}}_h^n(\cdot \mid s,a), \mathcal{P}_h^*(\cdot \mid s,a)\big)\Big] \leq C_{\text{reg},4}^2 n^{-2/(2+\gamma)}\log(20HN/\delta)$$

for any $(h,n) \in \times[H] \times [N]$ with probability $1 - \delta/(20HN)$. By taking a union bound, (E.6) holds for all $(i,h,n) \in [2] \times [H] \times [N]$ with probability at least $1 - \delta/20$, which concludes the proof of Lemma E.3.

$\square$

## G.1 Proof of Lemma G.1

*Proof.* We first define $\epsilon_\tau = r_\tau - r^*(x_\tau)$ and $\overline{\text{Risk}}(r) = \sum_{\tau=1}^{n}[r_\tau - r(x_\tau)]^2/n$. By the definition, the noise $\epsilon_\tau$ is a $\mathcal{F}_{\tau+1}$-measurable random variable with $\mathbb{E}[\epsilon_\tau \mid \mathcal{F}_\tau] = 0$. We have

$$\big[r_\tau - r(x_\tau)\big]^2 = \epsilon_\tau^2 + 2\epsilon_\tau\big[r^*(x_\tau) - r(x_\tau)\big] + \big[r^*(x_\tau) - r(x_\tau)\big]^2 \tag{G.1}$$

for any fix $r \in \mathcal{R}$. We also have $\overline{\text{Risk}}(r^*) = \sum_{\tau=1}^{n}\epsilon_\tau^2$. Since $\mathbb{E}[\epsilon_\tau \mid \mathcal{F}_\tau] = 0$, by (G.1), we have

$$\mathbb{E}\Big[\big(r_\tau - r(x_\tau)\big)^2 - \big(r_\tau - r^*(x_\tau)\big)^2 - \big(r^*(x_\tau) - r(x_\tau)\big)^2 \mid \mathcal{F}_{\tau-1}\Big] = 0,$$

$$\text{and } \text{Var}\Big[\big(r_\tau - r(x_\tau)\big)^2 - \big(r_\tau - r^*(x_\tau)\big)^2 - \big(r^*(x_\tau) - r(x_\tau)\big)^2 \mid \mathcal{F}_{\tau-1}\Big]$$

$$= \text{Var}\Big[2\epsilon_\tau(r^*(x_\tau) - r(x_\tau)) \mid \mathcal{F}_{\tau-1}\Big] \leq 4\mathbb{E}\Big[\big(r^*(x_\tau) - r(x_\tau)\big)^2 \mid \mathcal{F}_{\tau-1}\Big].$$

Applying Lemma J.4 with $\lambda = 1/4$, we have

$$\overline{\text{Risk}}(r) - \overline{\text{Risk}}(r^*) - \text{Risk}(r) \geq -(e-2)\sum_{\tau=1}^{n}\mathbb{E}\Big[\big(r^*(x_\tau) - r(x_\tau)\big)^2 \mid \mathcal{F}_{\tau-1}\Big]\Big/n - 4\log(1/\delta)/n$$

with probability at least $1 - \delta$. By the definition of the population risk $\text{Risk}(r)$, we have

$$\overline{\text{Risk}}(r) - \overline{\text{Risk}}(r^*) - (3-e)\text{Risk}(r) \geq -4\log(1/\delta)/n \tag{G.2}$$

with probability at least $1 - \delta$. Equation (G.2) shows the concentration of the risk for a fix $r \in \mathcal{R}$. For the uniform convergence, we define $\mathcal{C}$ as the $\epsilon$-covering set of $\mathcal{R}$ with infinity norm, and have

$$\overline{\text{Risk}}(r) - \overline{\text{Risk}}(r^*) - \text{Risk}(r)/4 \geq -4\log(\mathcal{N}/\delta)/n \tag{G.3}$$

for all $r \in \mathcal{C}$ with probability at least $1 - \delta$ by taking a union bound. Here the covering number $\mathcal{N} = |\mathcal{C}|$. Therefore, when we denote by $\mathcal{E}_{\mathcal{R}}$ the event that (G.3) holds for all $r \in \mathcal{C}$, we have $\mathbb{P}(\mathcal{E}_{\mathcal{R}}) \geq 1 - \delta$. In the following part of the proof, we condition on Event $\mathcal{E}_{\mathcal{R}}$. For an arbitrary $r \in \mathcal{R}$, we choose $r' \in \mathcal{R}$ such that $\|r - r'\|_\infty \leq \epsilon$. First, we have

$$\overline{\text{Risk}}(r) - \text{Risk}(r)/4 - \overline{\text{Risk}}(r') + \text{Risk}(r')/4 = \frac{1}{n}\sum_{\tau=1}^{n}\big(2r_\tau - r(x_\tau) - r'(x_\tau)\big)\big(r'(x_\tau) - r(x_\tau)\big)$$

$$+ \frac{1}{4n}\sum_{\tau=1}^{n}\mathbb{E}\Big[\big(2r^*(x_\tau) - r(x_\tau) - r'(x_\tau)\big)\big(r(x_\tau) - r'(x_\tau)\big) \mid \mathcal{F}_{\tau-1}\Big]$$

By the definition of the covering and the boundedness of the reward, we have $\overline{\text{Risk}}(r) - \text{Risk}(r)/4 - \overline{\text{Risk}}(r') + \text{Risk}(r')/4 \geq -3\epsilon$. Therefore, we have

$$\overline{\text{Risk}}(r) - \text{Risk}(r)/4 - \overline{\text{Risk}}(r^*) \geq -4\log(\mathcal{N}/\delta)/n - 3\epsilon$$

for all $r \in \mathcal{R}$ when condition on $\mathcal{E}_{\mathcal{R}}$ defined in (G.3). Let $\widehat{r}$ be the minimizer of $\overline{\text{Risk}}(r)$, we have

$$-\text{Risk}(\widehat{r}) \geq 4\overline{\text{Risk}}(\widehat{r}) - \text{Risk}(\widehat{r}) - 4\overline{\text{Risk}}(r^*) \geq -16\log(\mathcal{N}/\delta)/n - 12\epsilon$$

when condition on Event $\mathcal{E}_{\mathcal{R}}$. Since $\mathbb{P}(\mathcal{E}_{\mathcal{R}}) \geq 1 - \delta$, we conclude the proof of Lemma G.1.

$\square$

## G.2  PROOF OF LEMMA G.2

*Proof.* We denote by H the Hellinger divergence of two probability measures, which is defined as

$$H^2\big(\mathcal{P}_1(\cdot), \mathcal{P}_2(\cdot)\big) = \frac{1}{2} \int_{\mathcal{S}} \Big(\sqrt{\mathcal{P}_1(s)} - \sqrt{\mathcal{P}_2(s)}\Big)^2 ds.$$

Since we have $\mathrm{TV}^2(\mathcal{P}_1, \mathcal{P}_2) \le 8H^2(\mathcal{P}_1, \mathcal{P}_2)$, it remains to bound the population risk from the above in terms of the Hellinger divergence. We now require the following lemmas, which connect the Hellinger divergence with the uniform law of large number (ULLN).

**Lemma G.4** (Lemma 4.2 in Van de Geer (2000))**.** *For the Hellinger divergence* H*, two probability measures* $\widehat{\mathcal{P}}(\cdot \mid s, a)$ *and* $\mathcal{P}^*(\cdot \mid s, a)$*, we have*

$$16H^2\Big(\big(\widehat{\mathcal{P}}(\cdot \mid s, a) + \mathcal{P}^*(\cdot \mid s, a)\big)/2, \mathcal{P}^*(\cdot \mid s, a)\Big) \ge H^2\big(\widehat{\mathcal{P}}(\cdot \mid s, a), \mathcal{P}^*(\cdot \mid s, a)\big).$$

**Lemma G.5.** *When we define* $\bar{\mathcal{P}}(s' \mid s, a) = (\widehat{\mathcal{P}}(s' \mid s, a) + \mathcal{P}^*(s' \mid s, a))/2$*, we have*

$$2 \sum_{\tau=1}^{n} \mathbb{E}\Big[H^2\big(\bar{\mathcal{P}}(\cdot \mid S_\tau, A_\tau), \mathcal{P}^*(\cdot \mid S_\tau, A_\tau)\big) \mid \mathcal{F}_{\tau-1}\Big]$$

$$\le \sum_{\tau=1}^{n} \log \frac{\bar{\mathcal{P}}(S'_\tau \mid S_\tau, A_\tau)}{\mathcal{P}^*(S'_\tau \mid S_\tau, A_\tau)} - \sum_{\tau=1}^{n} \mathbb{E}\bigg[\log \frac{\bar{\mathcal{P}}(S'_\tau \mid S_\tau, A_\tau)}{\mathcal{P}^*(S'_\tau \mid S_\tau, A_\tau)} \mid \mathcal{F}_{\tau-1}\bigg].$$

*Proof.* We conclude the proof by directly applying Lemma 4.1 in Van de Geer (2000). $\square$

We define the function $g_{\mathcal{P},1}$, the functional $\nu_{n,1}, \nu_{n,1}$ as follows,

$$g_{\mathcal{P},1}(s, a, s') = \frac{1}{2} \log\Big(\big(\mathcal{P}(s' \mid s, a) + \mathcal{P}^*(s' \mid s, a)\big)/\big(2\mathcal{P}^*(s' \mid s, a)\big)\Big), \tag{G.4}$$

$$\nu_{n,1}(g) = \frac{1}{\sqrt{n}} \sum_{\tau=1}^{n} \Big\{g(s_\tau, a_\tau, s'_\tau) - \mathbb{E}_{(s,a)\sim\rho_\tau, s'\sim\mathcal{P}^*(\cdot \mid s, a)}\big[g(s, a, s')\big]\Big\},$$

$$\nu_{n,2}(\mathcal{P}) = \frac{1}{n} \sum_{\tau=1}^{n} \mathbb{E}_{(s,a)\sim\rho_\tau}\bigg[H^2\Big(\big(\mathcal{P}(s' \mid s, a) + \mathcal{P}^*(s' \mid s, a)\big)/2, \mathcal{P}^*(\cdot \mid s, a)\Big)\bigg]. \tag{G.5}$$

By Lemma G.5, we have $\nu_{n,1}(g_{\widehat{\mathcal{P}},1}) - \sqrt{n}\nu_{n,2}(\widehat{\mathcal{P}}) \ge 0$. By Lemma G.4, we can prove Lemma G.2 by showing that $\nu_{n,2}(\widehat{\mathcal{P}}) \le \epsilon^2/128$ holds with high probability. Therefore, we only need to prove that

$$\mathbb{P}\Big(\sup_{\mathcal{P}\in\mathcal{M}, \nu_{n,2}(\mathcal{P})\ge\epsilon^2/128} \nu_{n,1}(g_{\mathcal{P},1}) - \sqrt{n}\nu_{n,2}(\mathcal{P}) \ge 0\Big) \le C\exp\big(-n\epsilon^2/C^2\big)$$

for some absolute constant $C$. Since the Hellinger distance is bounded from the above by 1, we have

$$\mathbb{P}\Big(\sup_{\mathcal{P}\in\mathcal{M}, \nu_{n,2}(\mathcal{P})\ge\epsilon^2/128} \nu_{n,1}(g_{\mathcal{P},1}) - \sqrt{n}\nu_{n,2}(\mathcal{P}) \ge 0\Big) \tag{G.6}$$

$$\le \sum_{s=1}^{S} \mathbb{P}\Big(\sup_{g\in\mathcal{G}(2^{s-4}\epsilon)} \nu_{n,1}(g) \ge \sqrt{n}2^{2s-10}\epsilon^2\Big),$$

where $S = \min\{s : 2^{s-4}\epsilon > 1\}$ and $\mathcal{G}(\epsilon) = \{g_{\mathcal{P},1} \mid \mathcal{P} \in \mathcal{M}, \nu_{n,2}(\mathcal{P}) \le \epsilon^2\}$. Therefore, we can prove Lemma G.2 by the uniform law of large number on the function class $\mathcal{G}(\epsilon)$. We introduce the definition of the bracketing and a related ULLN in martingale processes as follows.

**Definition G.6** ($\delta$-entropy with the bracketing)**.** *Let* $\{\mathcal{F}_\tau\}_{\tau=0}^{n}$ *be a filtration and* $\{X_\tau\}_{\tau=1}^{n}$ *be a* $\mathcal{X}$*-valued random process adapted to this filtration. For* $0 < \delta \le R$*, let* $\mathcal{N}_{B,M}(\delta, \mathcal{G}, \{X_\tau\}_{\tau=1}^{n}, \{\mathcal{F}_\tau\}_{\tau=0}^{n})$ *be the smallest value of* $N$ *for which there exists a non-random collection* $\{[g_j^L, g_j^U]\}_{j=1}^{N}$*, such that (1) for all* $g \in \mathcal{G}$*, there exists a non-random* $j(g) \in [N]$*, such that* $g_{j(g)}^L(x) \le g(x) \le g_{j(g)}^U(x)$ *for all* $x \in \mathcal{X}$*, and (2).* $\chi_M(g_j^U - g_j^L, \{X_\tau\}_{\tau=1}^{n}, \{\mathcal{F}_\tau\}_{\tau=0}^{n}) \le \delta$ *for*

*all $j \in [N]$. We define $\delta$-entropy with the bracketing $H_{B,M}$ as $H_{B,M}(\delta, \mathcal{G}, \{X_\tau\}_{\tau=1}^n, \{\mathcal{F}_\tau\}_{\tau=0}^n) = \log \mathcal{N}_{B,M}(\delta, \mathcal{G}, \{X_\tau\}_{\tau=1}^n, \{\mathcal{F}_\tau\}_{\tau=0}^n)$. Here $\chi_M(g, \{X_\tau\}_{\tau=1}^n, \{\mathcal{F}_\tau\}_{\tau=0}^n)$ is defined as*

$$\chi_M^2\big(g, \{X_\tau\}_{\tau=1}^n, \{\mathcal{F}_\tau\}_{\tau=0}^n\big) = 2M^2 \sum_{\tau=1}^n \mathbb{E}\Big[\exp\big(|g(X_\tau)|/M\big) - 1 - |g(X_\tau)|/M \,\big|\, \mathcal{F}_{\tau-1}\Big]\Big/n.$$

**Lemma G.7.** *Let $\{\mathcal{F}_\tau\}_{\tau=0}^n$ be a filtration and $\{X_\tau\}_{\tau=1}^n$ be a $\mathcal{X}$-valued random process adapted to this filtration. Suppose we have $\sup_{g \in \mathcal{G}} \chi_M(g) \leq R$ for the function class $\mathcal{G}$. We set the values of the parameters $M$, $\varsigma_0$, $\varsigma_1$, $\varsigma_2$, and $C_2$ such that the following inequalities hold.*

$$\varsigma_0 \leq \varsigma_2 \sqrt{n} R^2/M, \qquad \varsigma_0 \leq 8\sqrt{n} R, \qquad \varsigma_1^2 \geq C_2^2(\varsigma_2 + 1), \tag{G.7}$$

$$\varsigma_0 \geq \varsigma_1 \max\Big\{ \int_{\varsigma_0/(64\sqrt{n})}^R H_{B,M}^{1/2}\big(u, \mathcal{G}, \{X_\tau\}_{\tau=1}^n, \{\mathcal{F}_\tau\}_{\tau=0}^n\big) \mathrm{d}u, R \Big\}, \tag{G.8}$$

*where $H_{B,M}$ is defined in Definition G.6 and $C_2$ is an absolute constant. We have*

$$\mathbb{P}\Bigg( \sup_{g \in \mathcal{G}} \frac{1}{\sqrt{n}} \Big| \sum_{\tau=1}^n \Big\{ g(X_\tau) - \mathbb{E}\big[g(X_\tau) \mid \mathcal{F}_{\tau-1}\big] \Big\} \Big| \geq \varsigma_0 \Bigg) \leq C_2 \exp\Big( -\frac{\varsigma_0^2}{C_2^2 \varsigma_2 R^2} \Big).$$

*Proof.* We conclude the proof by directly applying Theorem 8.13 in Van de Geer (2000). $\qquad\square$

By the following lemma, we have $\chi_1(g) \leq 4\epsilon$ for $g \in \mathcal{G}(\epsilon)$.

**Lemma.** *For the function $g_{\mathcal{P},1}, \nu_{n,2}(\mathcal{P})$ defined in (G.4), (G.5). We have $\chi_1^2(g_{\mathcal{P},1}) \leq 16\nu_{n,2}(\mathcal{P})$.*

*Proof.* We conclude the proof by directly applying Lemma 7.2 in Van de Geer (2000). $\qquad\square$

Therefore, we can apply Lemma G.7 on $\mathcal{G}(2^{s-4}\epsilon)$ with $\varsigma_0 = \sqrt{n} 2^{2s-10}\epsilon^2$, $\varsigma_1 = 4C_2$, $\varsigma_2 = 15$, $M = 1$, and $R = 2^{s-2}\epsilon$. Our selection of parameters satisfies (G.7). To validate (G.8), we need to bound the generalized entropy of $\mathcal{G}(\epsilon)$ from the above, which depends on the distribution of the corresponding stochastic process. The following lemma decouples such dependency.

**Lemma G.8.** *We assume that $0 \leq \mathcal{P}^1(s' \mid s, a) \leq \mathcal{P}^2(s' \mid s, a)$, and define $g^i$, $\bar{\mathcal{P}}^i$ as*

$$\bar{\mathcal{P}}_i(s, a, s') = \sqrt{\frac{\mathcal{P}^i(s' \mid s, a) + \mathcal{P}^*(s' \mid s, a)}{2}}, \quad g^i(s, a, s') = \frac{1}{2} \log \frac{\bar{\mathcal{P}}_i^2(s, a, s')}{\mathcal{P}^*(s' \mid s, a)} \mathbb{1}_{\{\mathcal{P}^*(s'|s,a)>0\}},$$

*for $i = 1, 2$. Let $v$ be the Lebesgue measure of $\mathcal{S}$. We have*

$$\chi_1\Big(g^2 - g^1, \big\{(S_\tau, A_\tau, S_\tau')\big\}_{\tau=1}^n, \{\mathcal{F}_\tau\}_{\tau=0}^n\Big) \leq \sqrt{2v} \sup_{(s,a,s') \in \mathcal{S} \times \mathcal{A} \times \mathcal{S}} \big(\bar{\mathcal{P}}_2(s,a,s') - \bar{\mathcal{P}}_1(s,a,s')\big).$$

*Proof.* See G.7 for a detailed proof. $\qquad\square$

Combining Lemma G.8, Definition G.6, with the definition of $\mathcal{G}(\epsilon)$ in (G.6), we have

$$H_{B,1}\Big(u, \mathcal{G}(\epsilon), \big\{(S_\tau, A_\tau, S_\tau')\big\}_{\tau=1}^n, \{\mathcal{F}_\tau\}_{\tau=0}^n\Big) \leq H_\infty\Big((8v)^{-1/2} u, \bar{\mathcal{G}}\Big), \tag{G.9}$$

where $H_\infty(\epsilon, \mathcal{G})$ is the $\epsilon$-log-covering number of $\mathcal{G}$ with respect to the infinity norm, and $\bar{\mathcal{G}}$ is defined in Lemma G.2. When $\epsilon_n$ satisfies the condition in Lemma G.2 with $C_9 = 2^{10} C_2$ and $\epsilon \geq \epsilon_n$, we have

$$\sqrt{n} 2^{2s-10}\epsilon^2 \geq C_2 G(2^s \epsilon) \tag{G.10}$$

$$\geq C_2 \max\Big\{ 8 \int_{2^{2s-17}\epsilon^2}^{2^{s-3}\epsilon} H_\infty^{1/2}\big((2v)^{-1/2} u, \bar{\mathcal{G}}\big) \mathrm{d}u, 2^s \epsilon \Big\}.$$

Combining (G.9) with (G.10), we have

$$\sqrt{n} 2^{2s-10}\epsilon^2 \geq 4C_2 \max\Big\{ \int_{2^{2s-16}\epsilon^2}^{2^{s-2}\epsilon} H_{B,1}^{1/2}\Big(u, \mathcal{G}(2^{s-4}\epsilon), \big\{(S_\tau, A_\tau, S_\tau')\big\}_{\tau=1}^n, \{\mathcal{F}_\tau\}_{\tau=0}^n\Big) \mathrm{d}u, 2^{s-2}\epsilon \Big\}$$

for all $\epsilon \geq \epsilon_n$ and $s \geq 0$, which validates (G.8) in Lemma G.7. Therefore, by Lemma G.7, we have

$$\mathbb{P}\Big(\sup_{g \in \mathcal{G}(2^{s-4}\epsilon)} \nu_{n,1}(g) \geq \sqrt{n} 2^{2s-10} \epsilon^2\Big) \leq C_2 \exp\big(-(n2^{2s-16}\epsilon^2)/(15C_2^2)\big).$$

Therefore, by (G.6), when $\epsilon > 1/\sqrt{n}$ and $C_3$ is an absolute constant that large enough, we have

$$\mathbb{P}\Big(\sup_{\mathcal{P} \in \mathcal{M}, \nu_{n,2}(\mathcal{P}) > \epsilon^2/128} \nu_n(g_{\mathcal{P},1}) - \sqrt{n}\nu_{n,2}(\mathcal{P})\Big) \leq C_2 \sum_{s=0}^{\infty} \exp\big(-(n2^{2s-16}\epsilon^2)/(15C_2^2)\big)$$
$$\leq C_3 \exp\big(-n\epsilon^2/C_3^2\big).$$

We conclude the proof of Lemma G.2 by setting $C_9$ in Lemma G.2 by the maximum of $C_3$ and $2^{10}C_2$.

$\square$

### G.3  PROOF OF LEMMA G.3

*Proof.* We use the following lemmas to connect the covering number of different function classes and bound the covering number of $\bar{\mathcal{G}}$ from the above using the covering number of $\Psi$ and $\Psi$.

**Lemma G.9.** *We have $\mathcal{N}(\delta, \bar{\mathcal{G}}, \|\cdot\|_{\infty}) \leq \mathcal{N}(4\delta\sqrt{c}, \mathcal{M}, \|\cdot\|_{\infty})$, where $\bar{\mathcal{G}}$ is defined in Lemma G.2 and $c = \inf_{\mathcal{P} \in \mathcal{M}, s', s \in \mathcal{S}, a \in \mathcal{A}} \mathcal{P}(s' \mid s, a)$.*

*Proof.* See Appendix §G.4 for a detailed proof. $\square$

**Lemma G.10.** *Suppose the function class $\mathcal{M}$ is defined as*

$$\mathcal{M} = \Big\{\mathcal{P}_u(s' \mid s, a)/c[\mathcal{P}_u](s, a) \mid \mathcal{P}_u \in \mathcal{M}_u\Big\},$$

*where the normalization function $c[\mathcal{P}_u](s, a)$ is defined as $c[\mathcal{P}_u](s, a) = \int_{s' \in \mathcal{S}} \mathcal{P}_u(s' \mid s, a)\mathrm{d}(s')$. We assume that $c_{\max} \geq c[\mathcal{P}_u](s, a) \geq c_{\min}$ and $\mathcal{P}_u(s' \mid s, a) \leq 1$ for any $\mathcal{P}_u \in \mathcal{M}_u$. We have*

$$\mathcal{N}\big(\delta, \mathcal{M}, \|\cdot\|_{\infty}\big) \leq \mathcal{N}\big(c_{\min}^2\delta/(v + c_{\max}), \mathcal{M}_u, \|\cdot\|_{\infty}\big).$$

*Proof.* See Appendix §G.5 for a detailed proof. $\square$

**Lemma G.11.** *For two feature map classes $\Phi$ and $\Psi$, we define the function class $\mathcal{M}_u$ as*

$$\mathcal{M}_u = \Big\{\exp\big(-\|\phi(s, a) - \psi(s')\|_2^2/2\big) \mid \phi \in \Phi, \psi \in \Psi\Big\}.$$

*We have $\mathcal{N}(\delta, \mathcal{M}_u, \|\cdot\|_{\infty}) \leq \mathcal{N}(\delta/(4R), \Phi, \|\cdot\|_{\infty,2}) \cdot \mathcal{N}(\delta/(4R), \Psi, \|\cdot\|_{\infty,2})$, where $R$ is the bound of the feature maps.*

*Proof.* See Appendix §G.6 for a detailed proof. $\square$

For the density class $\mathcal{M}$, we have $\inf_{\mathcal{P} \in \mathcal{M}, s', s \in \mathcal{S}, a \in \mathcal{A}} \mathcal{P}(s' \mid s, a) \geq \exp(-2R^2)/c_{\max}$, where $R$ bounds the norm of the feature maps from the above. Combining Lemma G.9, G.10, with G.11, we have

$$\mathcal{N}\big(\delta, \mathcal{M}', \|\cdot\|_{\infty}\big) \leq \mathcal{N}\big(c_{\mathrm{cover}}\delta, \Phi, \|\cdot\|_{\infty,2}\big)\mathcal{N}\big(c_{\mathrm{cover}}\delta, \Psi, \|\cdot\|_{\infty,2}\big), \tag{G.11}$$

where $c_{\mathrm{cover}} = \exp(-R^2)c_{\min}^2/(r\sqrt{c_{\max}}(v + c_{\max}))$. We conclude the proof of Lemma G.3 by taking logarithms of both sides of (G.11). $\square$

### G.4 PROOF OF LEMMA G.9

*Proof.* For $\mathcal{P}_1', \mathcal{P}_2' \in \mathcal{G}'$, we have $\mathcal{P}_i'(s' \mid s, a) = \sqrt{(\mathcal{P}_i(s' \mid s, a) + \mathcal{P}^*(s' \mid s, a))/2}$ for $i = 1, 2$, $s', s \in \mathcal{S}$ and $a \in \mathcal{A}$ for some $\mathcal{P}_1, \mathcal{P}_2 \in \mathcal{M}$. Therefore, we have

$$\mathcal{P}_1'(s' \mid s, a) - \mathcal{P}_2'(s' \mid s, a) \tag{G.12}$$
$$= \frac{\mathcal{P}_1(s' \mid s, a) - \mathcal{P}_2(s' \mid s, a)}{\sqrt{2}\big(\sqrt{\mathcal{P}_1(s' \mid s, a) + \mathcal{P}^*(s' \mid s, a)} + \sqrt{\mathcal{P}_2(s' \mid s, a) + \mathcal{P}^*(s' \mid s, a)}\big)}$$

Combining (G.12) with the fact that $\mathcal{P}(s' \mid s, a) \geq c$ for any $\mathcal{P} \in \mathcal{M}$, we have

$$\big|\mathcal{P}_1'(s' \mid s, a) - \mathcal{P}_2'(s' \mid s, a)\big| \leq \big|\mathcal{P}_1(s' \mid s, a) - \mathcal{P}_2(s' \mid s, a)\big|/4\sqrt{c}. \tag{G.13}$$

Let $\mathcal{C}$ be a $4\sqrt{c}\epsilon$-covering set of $\mathcal{M}$. We define $\mathcal{C}'$ by

$$\mathcal{C}' = \Big\{\bar{g}_\mathcal{P} \mid \bar{g}_\mathcal{P}(s, a, s') = \sqrt{\big(\mathcal{P}(s' \mid s, a) + \mathcal{P}^*(s' \mid s, a)\big)/2},\ \mathcal{P} \in \mathcal{C}\Big\}.$$

By (G.13), $\mathcal{C}'$ is an $\epsilon$-covering set of $\bar{\mathcal{G}}$, and $|\mathcal{C}'| = \mathcal{N}(4\delta\sqrt{c}, \mathcal{M}, \|\cdot\|_\infty)$. Thus, we conclude the proof of Lemma G.9. □

### G.5 PROOF OF LEMMA G.10

*Proof.* For $\mathcal{P}^1, \mathcal{P}^2 \in \mathcal{M}$, we have $\mathcal{P}^i(s' \mid s, a) = \mathcal{P}_u^i(s' \mid s, a)/c[\mathcal{P}_u^i](s, a)$ holds for all $i \in [2]$, $s', s \in \mathcal{S}$ and $a \in \mathcal{A}$ for some $\mathcal{P}_u^1, \mathcal{P}_u^2 \in \mathcal{M}_u$. Therefore, we have

$$\mathcal{P}^1(s' \mid s, a) - \mathcal{P}^2(s' \mid s, a) = \frac{c[\mathcal{P}_u^1](s, a)\mathcal{P}_u^1(s' \mid s, a) - c[\mathcal{P}_u^1](s, a)\mathcal{P}_u^2(s' \mid s, a)}{c[\mathcal{P}_u^1](s, a)c[\mathcal{P}_u^1](s, a)} \tag{G.14}$$
$$+ \frac{c[\mathcal{P}_u^1](s, a)\mathcal{P}_u^2(s' \mid s, a) - c[\mathcal{P}_u^2](s, a)\mathcal{P}_u^2(s' \mid s, a)}{c[\mathcal{P}_u^1](s, a)c[\mathcal{P}_u^2](s, a)}.$$

Combining (G.14) with $c_{\max} \geq c[\mathcal{P}_u](s, a) \geq c_{\min}$ for all $\mathcal{P}_u \in \mathcal{M}_u$, we have

$$\big|\mathcal{P}^1(s' \mid s, a) - \mathcal{P}^2(s' \mid s, a)\big| \leq c_{\max}\big|\mathcal{P}_u^1(s' \mid s, a) - \mathcal{P}_u^2(s' \mid s, a)\big|/c_{\min}^2 \tag{G.15}$$
$$+ \mathcal{P}_u^2(s' \mid s, a)\big|c[\mathcal{P}_u^1](s, a) - c[\mathcal{P}_u^2](s, a)\big|/c_{\min}^2.$$

By the definition of the normalization function $c[\mathcal{P}_u](s, a)$, we have

$$\big|c[\mathcal{P}_u^1](s, a) - c[\mathcal{P}_u^2](s, a)\big| \leq \int_\mathcal{S} \big|\mathcal{P}_u^1(s' \mid s, a) - \mathcal{P}_u^2(s' \mid s, a)\big|\mathrm{d}s' \leq v\big\|\mathcal{P}_u^1 - \mathcal{P}_u^2\big\|_\infty, \tag{G.16}$$

where $v$ is the Lebesgue measure of $\mathcal{S}$. Combining (G.15), (G.16) with the fact that $\mathcal{P}_u(s' \mid s, a) \leq 1$ for all $\mathcal{P}_u \in \mathcal{M}_u$, we have $\big|\mathcal{P}^1(s' \mid s, a) - \mathcal{P}^2(s' \mid s, a)\big| \leq \big\|\mathcal{P}_u^1 - \mathcal{P}_u^2\big\|_\infty (v + c_{\max})/c_{\min}^2$. Let $\mathcal{C}$ be a $c_{\min}^2\epsilon/(v + c_{\max})$-covering set of $\mathcal{M}_u$. We define $\mathcal{C}'$ by

$$\mathcal{C}' = \Big\{\mathcal{P}_u(s' \mid s, a)/c[\mathcal{P}_u](s, a) \mid \mathcal{P}_u \in \mathcal{C}\Big\}.$$

The set $\mathcal{C}'$ is an $\epsilon$-covering set of $\mathcal{M}$, and $|\mathcal{C}'| = \mathcal{N}(c_{\min}^2\delta/(v + c_{\max}), \mathcal{M}_u, \|\cdot\|_\infty)$. Thus, we conclude the proof of Lemma G.10.

□

### G.6 PROOF OF LEMMA G.11

*Proof.* First, for any $x, y > 0$, we have

$$\big|\exp(-x^2) - \exp(-y^2)\big| = \Big|\int_{x^2}^{y^2} \exp(-u)\mathrm{d}u\Big| \leq \big|x^2 - y^2\big|. \tag{G.17}$$

We set $x = \|\phi_1(s,a) - \psi_1(s')\|_2 / \sqrt{2}$ and $y = \|\phi_2(s,a) - \psi_2(s')\|_2 / \sqrt{2}$ in (G.17) and have

$$\left| \exp\left(- \|\phi_1(s,a) - \psi_1(s')\|_2^2 / 2\right) - \exp\left(- \|\phi_2(s,a) - \psi_2(s')\|_2^2 / 2\right) \right| \tag{G.18}$$

$$\leq \frac{1}{2} \left| \|\phi_1(s,a) - \psi_1(s')\|_2^2 - \|\phi_1(s,a) - \psi_2(s')\|_2^2 \right|$$

$$+ \frac{1}{2} \left| \|\phi_1(s,a) - \psi_2(s')\|_2^2 - \|\phi_2(s,a) - \psi_2(s')\|_2^2 \right|$$

by triangle inequality. For the first term in (G.18), we have

$$\left| \|\phi_1(s,a) - \psi_1(s')\|_2^2 - \|\phi_1(s,a) - \psi_2(s')\|_2^2 \right| \tag{G.19}$$

$$\leq \|\psi_2(s') - \psi_1(s')\|_2 \|2\phi_1(s,a) - \psi_1(s') - \psi_2(s')\|_2$$

Combining (G.19) with the boundedness of the feature maps, we have

$$\left| \|\phi_1(s,a) - \psi_1(s')\|_2^2 - \|\phi_1(s,a) - \psi_2(s')\|_2^2 \right| \leq 4R \|\psi_2(s') - \psi_1(s')\|_2 \,.$$

Similarly, we have $\left| \|\phi_1(s,a) - \psi_2(s')\|_2^2 - \|\phi_2(s,a) - \psi_2(s')\|_2^2 \right| \leq 4R \|\phi_2(s,a) - \phi_1(s,a)\|_2 \,.$
By (G.18), we have

$$\left| \exp\left(- \|\phi_1(s,a) - \psi_1(s')\|_2^2 / 2\right) - \exp\left(- \|\phi_2(s,a) - \psi_2(s')\|_2^2 / 2\right) \right|$$

$$\leq 4R \max\left\{ \|\psi_2(s') - \psi_1(s')\|_2 , \|\phi_2(s,a) - \phi_1(s,a)\|_2 \right\}. \tag{G.20}$$

Let $\mathcal{C}_1$ be a $\epsilon/(4r)$-covering set of $\Phi$, and $\mathcal{C}_2$ be a $\epsilon/(4r)$-covering set of $\Psi$. We define $\mathcal{C}$ by

$$\mathcal{C} = \left\{ \exp\left(- \|\phi(s,a) - \psi(s')\|_2^2 / 2\right) \mid \phi \in \mathcal{C}_1, \psi \in \mathcal{C}_2 \right\}.$$

By (G.20), $\mathcal{C}$ is an $\epsilon$-covering set of $\mathcal{M}_u$, and $|\mathcal{C}| = \mathcal{N}(\delta/(4R), \Phi, \|\cdot\|_{\infty,2}) \cdot \mathcal{N}(\delta/(4R), \Psi, \|\cdot\|_{\infty,2})$.
Thus, we conclude the proof of Lemma G.11. □

## G.7 Proof of Lemma G.8

*Proof.* Since $2(\exp(x) - 1 - x) \leq (\exp(x) - 1)^2$ when $x \geq 0$, we have

$$\chi_1^2\left(g^U - g^L, \{(s_\tau, a_\tau, s'_\tau)\}_{\tau=1}^n\right) \leq \frac{1}{n} \sum_{\tau=1}^n \mathbb{E}\left[ \left( \exp\left(g^U(s_\tau, a_\tau, s'_\tau) - g^L(s_\tau, a_\tau, s'_\tau)\right) - 1 \right)^2 \bigg| \mathcal{F}_{\tau-1} \right]$$

$$= \frac{1}{n} \sum_{\tau=1}^n \mathbb{E}\left[ \int_{\mathcal{S}} \left( \sqrt{\frac{\mathcal{P}^*(s' \mid s_\tau, a_\tau) + \mathcal{P}^U(s' \mid s_\tau, a_\tau)}{\mathcal{P}^*(s' \mid s_\tau, a_\tau) + \mathcal{P}^L(s' \mid s_\tau, a_\tau)}} - 1 \right)^2 \mathcal{P}^*(s' \mid s_\tau, a_\tau) \mathrm{d}s' \bigg| \mathcal{F}_{\tau-1} \right]$$

$$\leq \frac{2}{n} \sum_{\tau=1}^n \mathbb{E}\left[ \int_{\mathcal{S}} \left( \bar{\mathcal{P}}^U(s_\tau, a_\tau, s') - \bar{\mathcal{P}}^L(s_\tau, a_\tau, s') \right)^2 \mathrm{d}s' \bigg| \mathcal{F}_{\tau-1} \right]. \tag{G.21}$$

We also have

$$\mathbb{E}\left[ \int_{\mathcal{S}} \left( \bar{\mathcal{P}}^U(s_\tau, a_\tau, s') - \bar{\mathcal{P}}^L(s_\tau, a_\tau, s') \right)^2 \mathrm{d}s' \bigg| \mathcal{F}_{\tau-1} \right] \tag{G.22}$$

$$\leq v \sup_{(s,a,s') \in \mathcal{S} \times \mathcal{A} \times \mathcal{S}} \left( \bar{\mathcal{P}}^U(s,a,s') - \bar{\mathcal{P}}^L(s,a,s') \right)^2.$$

Thus, we conclude the proof of Lemma G.8 by combining (G.21) with (G.22). □

## H Discussion on the Effective Dimension

We first provide the following lemma, which shows the relation between $\Lambda_2$ in Definition E.4 and the dimension of a space in the case of finite dimension.

**Lemma H.1.** *We have $\Lambda[n, \lambda_0] \leq d$ when $\mathcal{X} = \mathbb{R}^d$ and $K(x_1, x_2) = x_1^\top x_2$.*

*Proof.* For any $f \in \mathcal{H}$, there exists $x_f \in \mathbb{R}^d$, such that $f(x) = x_f^\top x$. We also have $k(x) = x$. By the definition of $\Gamma_p$, we have $\Gamma_p[\rho, \lambda_0, n]f(x) = \lambda_0 x_f^\top x + n x_f^\top \mathbb{E}_{x_1 \sim \rho}[x_1 x_1^\top]x$. Therefore, the operator $\Gamma_p[\rho, \lambda_0, n]$ can be written as $\lambda_0 I_d + n\mathbb{E}_{x_1 \sim \rho}[x_1 x_1^\top]$. By the property of the matrix trace, we have

$$n\mathbb{E}_{x \sim \rho}\Big[\langle k(x), \Gamma_p^{-1}[\rho, \lambda_0, n]k(x)\rangle_{\mathcal{H}}\Big] = n\mathbb{E}_{x \sim \rho}\Big[\mathrm{tr}\Big(x^\top \big\{\lambda_0 I_d + n\mathbb{E}_{x_1 \sim \rho}[x_1 x_1^\top]\big\}^{-1} x\Big)\Big]$$

$$= n\mathbb{E}_{x \sim \rho}\Big[\mathrm{tr}\Big(\big\{\lambda_0 I_d + n\mathbb{E}_{x_1 \sim \rho}[x_1 x_1^\top]\big\}^{-1} x x^\top\Big)\Big]$$

for any $\rho \in \Delta(\mathcal{X})$. By the exchangeability of the expectation and the trace, we have

$$n\mathbb{E}_{x \sim \rho}\Big[\langle k(x), \Gamma_p^{-1}[\rho, \lambda_0, n]k(x)\rangle_{\mathcal{H}}\Big] = n\,\mathrm{tr}\Big(\big\{\lambda_0 I_d + n\mathbb{E}_{x_1 \sim \rho}[x_1 x_1^\top]\big\}^{-1} \mathbb{E}_{x \sim \rho}[x x^\top]\Big)$$

$$= \mathrm{tr}(I_d) - \lambda_0\,\mathrm{tr}\Big(\big\{\lambda_0 I_d + N\mathbb{E}_{x_1 \sim \rho}[x_1 x_1^\top]\big\}^{-1}\Big) \le d.$$

Since $\rho$ can be an arbitrary distribution over $\mathcal{X}$, we conclude the proof of Lemma H.1. $\square$

## H.1 PROOF OF LEMMA E.8

To bound the effective dimension from the above, we construct an upper bound of the effective dimension by the eigenvalue of the operator in the RKHS, and then use the characterization of the eigenvalue in Lemma H.2 to obtain the upper bound. Lemma H.2 can be proven by Theorem A of Belkin (2018), and we provide the proof in Appendix §H.2 here for the completeness of our paper.

*Proof.* In the following part of the proof, we bound $\Lambda_1$ and $\Lambda_2$ from the above separately.

**Upper Bound of $\Lambda_1$.** Let $\mathcal{X}_{N+1} = \{x_1, \ldots, x_{N+1}\}$ be a subset of $\mathcal{X}$. We define $K_{N+1} = [K(x_{\tau_1}, x_{\tau_2})]_{\tau_1, \tau_2 = 1}^{N+1}$. Let $\lambda_j$ be the $j$-th eigenvalue of $K_{N+1}$ and $\alpha_j = (\alpha_{1,j}, \ldots, \alpha_{N+1,j})^\top$ denote the corresponding eigenvector, we have

$$\log\det(I_{N+1} + K_{N+1}/\lambda) = \sum_{i=1}^{N+1} \log(1 + \lambda_i/\lambda). \tag{H.1}$$

Therefore, we can bound $\log\det(I_{N+1} + K_{N+1}/\lambda)$ from the above by bounding $\{\lambda_i\}_{i=1}^{N+1}$ from the above. We define the operator $\Gamma_0$ as $\Gamma_0 f(x) = \sum_{i=1}^{N+1} f(x_i)K(x_i, x)/(N+1)$. The function $\sum_{i=1}^{N+1} \alpha_{i,j} K(x_i, x)$ is an eigenfunction of $\Gamma_0$ and $\lambda_j/(N+1)$ is the corresponding eigenvalue. The following lemma bounds the eigenvalue of $\Gamma_0$ from the above.

**Lemma H.2.** *Suppose $\mathcal{X} \subset \bar{\mathcal{X}} \subset \mathbb{R}^m$, and $\bar{\mathcal{X}}$ is a cube with side length $l \ge 1/\sqrt{m}$, and $\rho$ is a distribution over $\mathcal{X}$, and the operator $\Gamma : L_2^\rho(\mathcal{X}) \to L_2^\rho(\mathcal{X})$ is defined as*

$$\Gamma f(x) = \int_{x \in \mathcal{X}} f(t)K(t, x)\mathrm{d}\rho(t),$$

*where $K(t, x) = \exp(-\|t - x\|_2^2/2)$. It holds that $\lambda_i(\Gamma) \le C_{\mathrm{app},5}\exp(-C_{\mathrm{app},4}i^{1/m})$. Here $C_{\mathrm{app},5}$ and $C_{\mathrm{app},4}$ are two constants only depend on the side length $l$ and the dimension $m$.*

*Proof.* See Appendix §H.2 for a detailed proof. $\square$

Applying Lemma H.2 on $\Gamma_0$, we have $\lambda_j \le NC_{\mathrm{app},5}\exp(-C_{\mathrm{app},4}j^{1/m})$, where $C_{\mathrm{app},4}, C_{\mathrm{app},4}$ are two constants that only depend on $m$ and $R$. Therefore, by (H.1), we have

$$\log\det(I_{N+1} + K_{N+1}/\lambda) \le e\sum_{j=1}^{N_{\mathrm{mid}}} \log\big(1 + NC_{\mathrm{app},5}\exp(-C_{\mathrm{app},4}j^{1/m})/\lambda\big) \tag{H.2}$$

$$+ e\sum_{j=N_{\mathrm{mid}}+1}^{N} \log\big(1 + NC_{\mathrm{app},5}\exp(-C_{\mathrm{app},4}j^{1/m})/\lambda\big).$$

We have $\sum_{j=1}^{N_{\mathrm{mid}}} \log(1 + NC_{\mathrm{app},5}\exp(-C_{\mathrm{app},4}j^{1/m})/\lambda) \le N_{\mathrm{mid}}\log(1 + NC_{\mathrm{app},5}/\lambda)$. The following lemma bounds the second term from the above.

**Lemma H.3.** *If $m$ is a positive integer, we have $\int_x^\infty t^m e^{-t} \mathrm{d}t \leq 2m! x^m e^{-x}$ when $x \geq 1$.*

*Proof.* When we define $b_m = \int_x^\infty t^m e^{-t} \mathrm{d}t \leq 2x^m e^{-x}$, we have

$$b_m = -\int_x^\infty t^m de^{-t} = x^m e^{-x} + m \int_x^\infty t^{m-1} e^{-t} \mathrm{d}t = x^m e^{-x} + m b_{m-1}.$$

By induction, we have $b_m = m! e^{-x} \sum_{m_0=0}^m x^{m_0}/m_0! \leq 2m! x^m e^{-x}$, which concludes the proof of Lemma H.3. $\square$

Since $\log(1 + x) \leq x$, by the lemma above, we have

$$\sum_{j=N_{\mathrm{mid}}+1}^N \log\big(1 + N C_{\mathrm{app},5} \exp(-C_{\mathrm{app},4} j^{1/m})/\lambda\big) \leq \sum_{j=N_{\mathrm{mid}}+1}^N N C_{\mathrm{app},5} \exp(-C_{\mathrm{app},4} j^{1/m})/\lambda$$

$$\leq N m C_{\mathrm{app},5} \int_{C_{\mathrm{app},4} N_{\mathrm{mid}}^{1/m}}^\infty j^{m-1} \exp(-j) \mathrm{d}j/(C_{\mathrm{app},4}^m \lambda)$$

$$\leq 2N m! C_{\mathrm{app},5} N_{\mathrm{mid}}^{(m-1)/m} \exp(-C_{\mathrm{app},4} N_{\mathrm{mid}}^{1/m})/(\lambda C_{\mathrm{app},4}). \tag{H.3}$$

By (H.2) and (H.3), we have $\log \det(I_{N+1} + K_{N+1}/\lambda) \leq C_4 (\log N/C_{\mathrm{app},4})^{m+1}$ when we choose $N_{\mathrm{mid}} = (\log N/C_{\mathrm{app},4})^m$. Here $C_4$ is a constant that only depends on the dimension $m$ and the bound of the feature maps $R$. Since $\mathcal{X}_{N+1}$ can be any subset of $\mathcal{X}$ with $|\mathcal{X}_{N+1}| = N + 1$, we have $\Lambda_1(N + 1, \lambda) \leq C_4 (\log N/C_{\mathrm{app},4})^{m+1}$.

**Upper Bound of $\Lambda_2$.** In order to bound $\Lambda_2$ from the above, we need to choose an appropriate representation of the RKHS. We define the integral operator $\Gamma_0 : L_2^\rho(\mathcal{X}) \to L_2^\rho(\mathcal{X})$ as $\Gamma_0 f(x) = \mathbb{E}_{x_0 \sim \rho}[f(x_0) K(x_0, x)]$. Since $\rho$ is a probability measure and the Gaussian kernel $K$ is bounded from the above, $\Gamma_0$ is compact and self-adjoint. Therefore, by spectral theorem, there exists $\{e_j\}_{j=1}^\infty$ such that it is both the eigenfunction of the operator $\Gamma_0$ and the orthonormal basis of the space $L_2^\rho(\mathcal{X})$. We then define $\mathcal{H}$ as

$$\mathcal{H} = \Big\{ f : f \in L_2^\rho(\mathcal{X}), f(x) = \sum_{j=1}^\infty \alpha_j e_j(x) \mid \sum_{j=1}^\infty \alpha_j^2/\lambda_j < \infty \Big\},$$

where $\lambda_j$ is the eigenvalue corresponding to $e_j$. We also know that $K(x, y) = \sum_{j=1}^\infty \lambda_j e_j(x) e_j(y)$. For $f(x) = \sum_{j=1}^\infty \alpha_j e_j(x)$ and $g(x) = \sum_{j=1}^\infty \beta_j e_j(x)$, we define the inner product on $\mathcal{H}$ as $\langle f, g \rangle_{\mathcal{H}} = \sum_{j=1}^\infty \alpha_j \beta_j/\lambda_j$. Then the space $\mathcal{H}$ is an RKHS with the kernel $K$, $e_j/\sqrt{\lambda_j}$ is an eigenfunction of $\Gamma$ with eigenvalue $\lambda_j$, and $\{e_j/\sqrt{\lambda_j}\}_{j=1}^\infty$ is an orthonormal basis of $\mathcal{H}$. We represent $k(x)$ using the orthonormal basis and have

$$\|k(x)\|_{\Gamma^{-1}}^2 = \langle k(x), \Gamma^{-1} k(x) \rangle_{\mathcal{H}} = \Big\langle \sum_{j=1}^\infty \lambda_j e_j(x) e_j, \sum_{j=1}^\infty \lambda_j e_j(x) e_j/(\lambda + N\lambda_j) \Big\rangle_{\mathcal{H}}$$

$$= \sum_{j=1}^\infty \lambda_j e_j^2(x)/(\lambda + N\lambda_j)$$

when we define $\Gamma f(x) = \lambda f(x) + N\Gamma_0 f(x)$. Therefore, using Fubini's Theorem, we have

$$N\mathbb{E}_{x \sim \rho}\big[\|k(x)\|_{\Gamma^{-1}}^2\big] = \sum_{j=1}^\infty N\lambda_j \mathbb{E}_{x \sim \rho}\big[e_j^2(x)\big]/(\lambda + N\lambda_j) = \sum_{j=1}^\infty N\lambda_j/(\lambda + N\lambda_j). \tag{H.4}$$

By Lemma H.2, we have $\lambda_j \leq C_{\text{app},5} \exp(-C_{\text{app},4} j^{1/m})$, where $C_0$ and $C_1$ are two constants only depend on the side length $l$ and the dimension $m$. We combine the bound on $\lambda_j$ with (H.4) and have

$$
\begin{aligned}
N\mathbb{E}_{x\sim\rho}\big[\|k(x)\|_{\Gamma^{-1}}^2\big] &\leq \sum_{j=1}^{\infty} \frac{NC_{\text{app},5}\exp(-C_{\text{app},4}j^{1/m})}{NC_{\text{app},5}\exp(-C_{\text{app},4}j^{1/m})+\lambda} \\
&= \sum_{j=1}^{N_{\text{mid}}} \frac{NC_{\text{app},5}\exp(-C_{\text{app},4}j^{1/m})}{NC_{\text{app},5}\exp(-C_{\text{app},4}j^{1/m})+\lambda} + \sum_{j=N_{\text{mid}}+1}^{\infty} \frac{NC_{\text{app},5}\exp(-C_{\text{app},4}j^{1/m})}{NC_{\text{app},5}\exp(-C_{\text{app},4}j^{1/m})+\lambda} \\
&\leq N_{\text{mid}} + NC_{\text{app},5}\int_{N_{\text{mid}}}^{\infty}\exp(-C_{\text{app},4}j^{1/m})\mathrm{d}j/\lambda.
\end{aligned}
$$

By Lemma H.3, we have $\int_{N_{\text{mid}}}^{\infty}\exp(-C_{\text{app},4}j^{1/m})\mathrm{d}j \leq 2m! N_{\text{mid}}^{1-1/m}\exp(-C_{\text{app},4}N_{\text{mid}}^{1/m})/C_{\text{app},4}$. Therefore, we choose $N_{\text{mid}} = (\log N/C_{\text{app},4})^m$ and have

$$
N\mathbb{E}_{x\sim\rho}[\|k(x)\|_{\Gamma^{-1}}^2] \leq (\log N/C_{\text{app},4})^m + 2m! C_{\text{app},5}(\log N)^{m-1}/(C_{\text{app},4}\lambda) \leq \quad C_4(\log N)^m,
$$

where $C_4 = (1 + 2m! C_{\text{app},5}C_{\text{app},4}^{m-1})/C_{\text{app},5}^m$. By the definition, $C_4$ is a constant that only depends on the side length $l$ and the dimension $m$.

We conclude the proof of Lemma E.8 by combining the upper bound of $\Lambda_1$ and $\Lambda_2$. $\qquad\square$

## H.2 PROOF OF LEMMA H.2

*Proof.* We require the following two lemmas. The first lemma allows us to bound the eigenvalues by the bound on the residuals of the approximation, and the second lemma bounds the residuals.

**Lemma H.4** (Lemma 1 in Belkin (2018)). *Suppose $\Gamma : \mathcal{H} \to \mathcal{H}$ is a self-adjoint operator on a Hilbert space $\mathcal{H}$, and $\Gamma_n$ is a finite-rank operator with rank $n$, such that $\|\Gamma - \Gamma_n\|_{\text{op}} \leq \epsilon$. Here $\|A\|_{\text{op}} = \sup_{x\in\mathcal{H}/\{0\}}\|Ax\|_{\mathcal{H}}/\|x\|_{\mathcal{H}}$. It holds that all eigenvalues of the operator $\Gamma$ except for at most $n$ (counting multiplicity) are smaller than $\epsilon$.*

**Lemma H.5.** *Suppose $\mathcal{X} \subset \mathbb{R}^m$ is a cube with side length $l \geq 1/\sqrt{m}$, and $\Gamma : V \to \mathcal{H}$ is a (not necessarily linear) map from a Hilbert space $V$ to an RKHS $\mathcal{H}$ of functions on $\mathbb{R}^m$. There exists a map $\Gamma_n$ from the space $V$ to an $n$-dimensional linear subspace $\mathcal{H}_n \subset \mathcal{H}$, such that*

$$
\|\Gamma - \Gamma_n\|_{V\to L_2^{\rho}(\mathcal{X})} \leq C_{\text{app},2}\exp(-C_{\text{app},3}n^{1/m})\|\Gamma\|_{V\to\mathcal{H}}.
$$

*Here $C_{\text{app},2}$ and $C_{\text{app},3}$ are two positive constants that only depend on the side length $l$ and the dimension $m$, and $\|\Gamma\|_{V\to L_2^{\rho}(\mathcal{X})} = \sup_{v\in V/\{0\}}\|\Gamma v\|_{L_2^{\rho}(\mathcal{X})}/\|v\|_V$.*

**Remark.** *Since $\mathcal{H}$ is a subset of $L_2^{\rho}(\mathcal{X})$, we can view $\Gamma$ as an operator from the space $V$ to the space $L_2^{\rho}(\mathcal{X})$ and investigate its operator norm accordingly.*

*Proof.* See Appendix §H.3 for a detailed proof. $\qquad\square$

First, by Lemma H.4, we have $\lambda_i(\Gamma) \leq \inf_{\text{rank}(\Gamma_{i-1})=i-1}\|\Gamma - \Gamma_{i-1}\|_{\text{op}}$. By Lemma H.5, we have

$$
\inf_{\text{rank}(\Gamma_{i-1})=i-1}\|\Gamma - \Gamma_{i-1}\|_{\text{op}} = \inf_{\text{rank}(\Gamma_{i-1})=i-1}\|\Gamma - \Gamma_{i-1}\|_{L_2^{\rho}(\mathcal{X})\to L_2^{\rho}(\mathcal{X})} \tag{H.5}
$$

$$
\leq C_{\text{app},2}\exp\big(-C_{\text{app},3}(i-1)^{1/m}\big)\|\Gamma\|_{L_2^{\rho}(\mathcal{X})\to\mathcal{H}} \leq C_{\text{app},2}\exp\big(-C_{\text{app},4}i^{1/m}\big)\|\Gamma\|_{L_2^{\rho}(\mathcal{X})\to\mathcal{H}}
$$

for $i > 1$. Here $C_{\text{app},4} = C_{\text{app},3}2^{-1/m}$. It remains to bound $\|\Gamma\|_{L_2^{\rho}(\mathcal{X})\to\mathcal{H}}$ from the above. For elements $e \in \mathcal{H}$ and $e' \in L_2^{\rho}(\mathcal{X})$, we have

$$
\begin{aligned}
\langle e, \Gamma e'\rangle_{\mathcal{H}} &= \Big\langle e, \int_{x\in\mathcal{X}} e'(t)K(t,\cdot)\mathrm{d}\rho(t)\Big\rangle_{\mathcal{H}} = \int_{x\in\mathcal{X}}\langle e, K(t,\cdot)\rangle_{\mathcal{H}}e'(t)\mathrm{d}\rho(t) = \int_{x\in\mathcal{X}}e(t)e'(t)\mathrm{d}\rho(t) \\
&\leq \|e\|_{L_2^{\rho}(\mathcal{X})}\|e'\|_{L_2^{\rho}(\mathcal{X})} \leq \|e\|_{\mathcal{H}}\|e'\|_{L_2^{\rho}(\mathcal{X})},
\end{aligned}
$$

where the last inequality is derived from (H.10). Therefore, we have

$$\|\Gamma e'\|_{\mathcal{H}} = \sup_{e \in \mathcal{H}} \langle e, \Gamma e' \rangle_{\mathcal{H}} / \|e\|_{\mathcal{H}} \le \|e'\|_{L_2^\rho(\mathcal{X})},$$

which implies $\|\Gamma\|_{L_2^\rho(\mathcal{X}) \to \mathcal{H}} \le 1$. Therefore, we have $\lambda_i(\Gamma) \le C_{\mathrm{app},2} \exp(-C_{\mathrm{app},4} i^{1/m})$ for $i > 1$ by (H.5). For $i = 1$, we have

$$\begin{aligned}
\lambda_1(\Gamma) &\le \sup_{x \in L_2^\rho(\mathcal{X})/\{0\}} \|\Gamma x\|_{L_2^\rho(\mathcal{X})} / \|x\|_{L_2^\rho(\mathcal{X})} \\
&\le \sup_{x \in L_2^\rho(\mathcal{X})/\{0\}} \|\Gamma x\|_{\mathcal{H}} / \|x\|_{L_2^\rho(\mathcal{X})} = \|\Gamma\|_{L_2^\rho(\mathcal{X}) \to \mathcal{H}} \le 1.
\end{aligned}$$

Therefore, we have $\lambda_i(\Gamma) \le C_{\mathrm{app},5} \exp(-C_{\mathrm{app},4} i^{1/m})$ for all integer $i$ when we set $C_{\mathrm{app},5} = C_{\mathrm{app},2} \exp(C_{\mathrm{app},4})$, which concludes the proof of Lemma H.2.

$\square$

## H.3    PROOF OF LEMMA H.5

*Proof.* We prove Lemma H.5 by constructing an operator that satisfies the condition in this lemma. For $n \ge (3\gamma_m \exp(8m\gamma_m + 2)m^{3/2}l^2)^m$, we have

$$\max_{x \in \mathcal{X}} \min_{x' \in \mathcal{X}_n} \|x - x'\|_2 = \sqrt{m} l n^{-1/m} \le 1/(3\gamma_m \exp(8m\gamma_m + 2)ml)$$

when $\gamma_m = 4^m m!$ and $\mathcal{X}_n = (x_1, \dots, x_n)$ is an $m$-dimensional grid of $\mathcal{X}$. By Theorem 6.10 of Wendland (2004), the kernel $K(x_1, x_2) = \exp(-\|x_1 - x_2\|_2^2 / 2)$ is positive-definite. Therefore, the matrix $K[\mathcal{X}_n]$ is invertible, and $\sum_{i=1}^n \alpha_{ij_1} K(x_i, x_{j_2}) = \mathbb{1}_{j_1 = j_2}$ when $K[\mathcal{X}_n]\alpha_j = e_j$, $\alpha_{ij}$ is the $i$-th element of $\alpha_j$, and

$$K[\mathcal{X}_n] = \left[ K(x_{\tau_1}, x_{\tau_2}) \right]_{\tau_1, \tau_2 = 1}^n, \tag{H.6}$$

$$k[\mathcal{X}_n](x) = \left( K(x, x_1), \dots, K(x, x_n) \right)^\top. \tag{H.7}$$

The following lemma allows us to construct an operator that satisfies the condition in Lemma H.5.

**Lemma H.6.** *Let $\mathcal{H}$ be the RKHS induced by the kernel $K$, where $K : \mathbb{R}^m \times \mathbb{R}^m \to \mathbb{R}$ is defined as $K(x, x') = \exp(-\|x - x'\|_2^2 / 2)$. Suppose $\mathcal{X} \subset \mathbb{R}^m$ is a cube with side length $l \ge \sqrt{2/m}$, and*

$$\max_{x \in \mathcal{X}} \min_{x' \in \mathcal{X}_n} \|x - x'\|_2 = \iota \le 1/(3 \cdot 4^m m! \exp(m2^{2m+3}m! + 2)ml),$$

*for the set $\mathcal{X}_n \subset \mathcal{X}$ with $|\mathcal{X}_n| = n$. We define $u_j(x) = \sum_{i=1}^n \alpha_{ij} K(x_i, x)$, where $\alpha_j = K^{-1}[\mathcal{X}_n]e_j$, $\alpha_{ij}$ is the $i$-th element of $\alpha_j$, and the matrix $K[\mathcal{X}_n]$ is defined in (H.6). We also define $S_{\mathcal{X}_n}$ as $S_{\mathcal{X}_n}f(x) = \sum_{i=1}^n f(x_i)u_i(x)$ for all $f \in \mathcal{H}$. For an operator $\Gamma : \mathcal{H} \to L_2^\rho(\mathcal{X})$, where $\rho$ is a probability measure over $\mathcal{X}$, we define $\|\Gamma\|_{\mathcal{H} \to L_2^\rho(\mathcal{X})} = \sup_{f \in \mathcal{H}/\{0\}} \|\Gamma f\|_{L_2^\rho(\mathcal{X})} / \|f\|_{\mathcal{H}}$. We have*

$$\|S_0 - S_{\mathcal{X}_n}\|_{\mathcal{H} \to L_2^\rho(\mathcal{X})} \le 4l\sqrt{m} \exp(-C_{\mathrm{app},1}/(2\iota)),$$

*where $\gamma_m = 4^m \cdot m!$, $C_{\mathrm{app},1} = l/(3\gamma_m)$, and $S_0 : \mathcal{H} \to L_2^\rho(\mathcal{X})$ is defined as $S_0 f(x) = f(x)$.*

*Proof.* See Appendix §H.4 for a detailed proof. $\square$

Let $S_0$ and $S_{\mathcal{X}_n}$ be the operator defined in Lemma H.6, we have $\Gamma - S_{\mathcal{X}_n} = (S_0 - S_{\mathcal{X}_n}) \circ \Gamma$. Combining the definition of the norm with Lemma H.6, we have

$$\begin{aligned}
\|\Gamma - S_{\mathcal{X}_n} \circ \Gamma\|_{V \to L_2^\rho(\mathcal{X})} &\le \|\Gamma\|_{V \to \mathcal{H}} \|S_0 - S_{\mathcal{X}_n}\|_{\mathcal{H} \to L_2^\rho(\mathcal{X})} \tag{H.8} \\
&\le 4l\sqrt{m} \exp(-C_{\mathrm{app},1} n^{1/m}/(2l)) \|\Gamma\|_{V \to \mathcal{H}}.
\end{aligned}$$

Here $C_{\mathrm{app},1} = l/(3\gamma_m)$. Therefore, we also have

$$\|\Gamma - \Gamma_n\|_{V \to L_2^\rho(\mathcal{X})} \le 4l\sqrt{m} \exp(-C_{\mathrm{app},1} n^{1/m}/(2l)) \|\Gamma\|_{V \to \mathcal{H}}$$

when we define $\Gamma_n = S_{\mathcal{X}_n} \circ \Gamma$ in the case that $n \geq (3\gamma_m \exp(8m\gamma_m + 2)m^{3/2}l^2)^m$. Since the rank of $S_{\mathcal{X}_n}$ does not exceed $n$, the rank of $\Gamma_n$ does not exceed $n$. When $n < (3\gamma_m \exp(8m\gamma_m + 2)m^{3/2}l^2)^m$, we define $\Gamma_n$ as $\Gamma_n f(x) = 0$ for all $x \in \mathcal{X}$ and $v \in V$. We then have $\|\Gamma\|_{V \to L_2^\rho(\mathcal{X})} = \|\Gamma - \Gamma_n\|_{V \to L_2^\rho(\mathcal{X})}$. We first show that $\|\Gamma\|_{V \to L_2^\rho(\mathcal{X})}$ is bounded from the above by $\|\Gamma\|_{V \to \mathcal{H}}$. By the reproducing property of the space $\mathcal{H}$, we have

$$\|g\|_{L_2^\rho(\mathcal{X})}^2 = \int_{x \in \mathcal{X}} g^2(x) \mathrm{d}\rho(x) = \int_{x \in \mathcal{X}} \left( \langle g, k(x) \rangle_{\mathcal{H}} \right)^2 \mathrm{d}\rho(x) \tag{H.9}$$

for any $g \in \mathcal{H}$. Combining (H.9) with Cauchy-Schwarz inequality, we have

$$\|g\|_{L_2^\rho(\mathcal{X})}^2 \leq \int_{x \in \mathcal{X}} \langle g, g \rangle_{\mathcal{H}} \langle k(x), k(x) \rangle_{\mathcal{H}} \mathrm{d}\rho(x) = \int_{x \in \mathcal{X}} \langle g, g \rangle_{\mathcal{H}} \mathrm{d}\rho(x) = \|g\|_{\mathcal{H}}^2 \tag{H.10}$$

for any $g \in \mathcal{H}$ when $\rho$ is a probability measure. By (H.10), we have

$$\|\Gamma\|_{V \to L_2^\rho(\mathcal{X})} = \sup_{v \in V/\{0\}} \|\Gamma v\|_{L_2^\rho(\mathcal{X})} / \|v\|_V \leq \sup_{v \in V/\{0\}} \|\Gamma v\|_{\mathcal{H}} / \|v\|_V = \|\Gamma\|_{V \to \mathcal{H}}. \tag{H.11}$$

Since $\|\Gamma - \Gamma_n\|_{V \to L_2^\rho(\mathcal{X})} = \|\Gamma\|_{V \to L_2^\rho(\mathcal{X})}$ when we define $\Gamma_n$ as $\Gamma_n v(x) = 0$, we have

$$\|\Gamma - \Gamma_n\|_{V \to L_2^\rho(\mathcal{X})} \leq \exp\left(C_{\mathrm{app},1} n^{1/m}/(2l)\right) \exp\left(-C_{\mathrm{app},1} n^{1/m}/(2l)\right) \|\Gamma\|_{V \to \mathcal{H}}. \tag{H.12}$$

by (H.11). Combining (H.12) with $n \leq (3\gamma_m \exp(8m\gamma_m + 2)m^{3/2}l^2)^m$, we have

$$\|\Gamma - \Gamma_n\|_{V \to L_2^\rho(\mathcal{X})} \leq \exp\left(m^{3/2}l^2 \exp(8m\gamma_m + 2)/2\right) \exp\left(-n^{1/m}/(6\gamma_m)\right) \|\Gamma\|_{V \to \mathcal{H}} \tag{H.13}$$

since $C_{\mathrm{app},1} = l/(3\gamma_m)$. Therefore, by combining (H.8) with (H.13), when we choose

$$C_{\mathrm{app},2} = \max\left\{ 4l\sqrt{m}, \exp\left(m^{3/2}l^2 \exp(8m\gamma_m + 2)/2\right) \right\}$$

and $C_{\mathrm{app},3} = C_{\mathrm{app},1}/(2l) = 1/(6\gamma_m)$, we have (1) both $C_{\mathrm{app},2}$ and $C_{\mathrm{app},3}$ are constants that only depend on the side-length $l$ and the dimension $m$, and (2) for any positive integer $n$, there exists an operator $\Gamma_n : V \to L_2^\rho(\mathcal{X})$ with finite rank $n$, such that

$$\|\Gamma - \Gamma_n\|_{V \to L_2^\rho(\mathcal{X})} \leq C_{\mathrm{app},2} \exp(-C_{\mathrm{app},3} n^{1/m}) \|\Gamma\|_{V \to \mathcal{H}}.$$

Thus, we conclude the proof of Lemma H.5. $\qquad\square$

## H.4 PROOF OF LEMMA H.6

*Proof.* By the definition in Lemma H.6 and the reproducing property of the kernel, we have

$$\left| f(x) - S_{\mathcal{X}_n} f(x) \right| = \left| \langle k(x), f \rangle_{\mathcal{H}} - \sum_{i=1}^n \langle k(x_i), f \rangle_{\mathcal{H}} u_i(x) \right| \tag{H.14}$$

$$= \left| \left\langle k(x) - \sum_{i=1}^n u_i(x) k(x_i), f \right\rangle_{\mathcal{H}} \right|.$$

Combining (H.14) with Cauchy-Schwarz inequality, we have

$$\left| f(x) - S_{\mathcal{X}_n} f(x) \right| \leq \left\| k(x) - \sum_{i=1}^n u_i(x) k(x_i) \right\|_{\mathcal{H}} \|f\|_{\mathcal{H}}.$$

Therefore, taking expectation with respect to the probability measure $\rho$, we have

$$\left\| (R - S_{\mathcal{X}_n}) \circ f \right\|_{L_\rho^2(\mathcal{X})} = \int_{\mathcal{X}} \left| f(x) - S_{\mathcal{X}_n, u} f(x) \right|^2 \mathrm{d}\rho(x) \tag{H.15}$$

$$\leq \sup_{x \in \mathcal{X}} \left\| k(x) - \sum_{i=1}^n u_i(x) k(x_i) \right\|_{\mathcal{H}}^2 \|f\|_{\mathcal{H}}^2.$$

The following lemma allows us to bound the term $\sup_{x \in \mathcal{X}} \|k(x) - \sum_{i=1}^n u_i(x) k(x_i)\|_{\mathcal{H}}^2$ from the above.

**Lemma H.7.** *For a fix $x \in \mathcal{X}$, a set $\mathcal{X}_n = \{x_\tau\}_{\tau=1}^n \subset \mathcal{X}$, and a vector $v = (v_1, \dots, v_n)^\top$, we have* $\min_{v \in \mathbb{R}^n} Q(v) = K(x, x) - k[\mathcal{X}_n](x)^\top K^{-1}[\mathcal{X}_n]k[\mathcal{X}_n](x)$ *when we define $Q(v)$ as*

$$Q(v) = K(x, x) - 2\sum_{i=1}^n v_i K(x_i, x) + \sum_{i=1}^n \sum_{j=1}^n v_i v_j K(x_i, x_j)$$

$$= K(x, x) - 2v^\top k[\mathcal{X}_n](x) + v^\top K[\mathcal{X}_n]v.$$

*Here the matrix $K[\mathcal{X}_n]$, the vector $k[\mathcal{X}_n](x)$ are defined in (H.6) and (H.7).*

*Proof.* We conclude the proof by using simple linear algebra. $\qquad\square$

Combining (H.15) with the lemma above, we can prove Lemma H.6 by bounding the term

$$\sup_{x \in \mathcal{X}} \inf_{v \in \mathbb{R}^n} K(x, x) - 2v^\top k[\mathcal{X}_n](x) + v^\top K[\mathcal{X}_n]v.$$

We bound the term using tools from the approximation theory.

**Lemma H.8** (Theorem 11.21 in Wendland (2004)). *Suppose that $\mathcal{X} \subset \mathbb{R}^m$ is a cube with side length $l$, and we have $\max_{x \in \mathcal{X}} \min_{x' \in \mathcal{X}_n} \|x - x'\|_2 \leq \iota$ for $\mathcal{X}_n = \{x_1, \dots, x_n\}$. We define $\gamma_m = 4^m m!$, $C_{\mathrm{app},1} = l/(3\gamma_m)$, and $q = [C_{\mathrm{app},1}/\iota]$, where $[x]$ denote the maximal integer that does not exceed $x$. Then there exists a sequence of function $\{v_i(x)\}_{i=1}^n \subset \mathcal{X} \to \mathbb{R}$ such that (1) for every $p \in \varpi_q(\mathbb{R}^m)$, we have $p(x) = \sum_{i=1}^n v_i(x)p(x_i)$ for all $x \in \mathcal{X}$, (2) we have $\sum_{i=1}^n |v_i(x)| \leq \exp(2m\gamma_m(q+1))$ for all $x \in \mathcal{X}$, and (3) we have $v_i(x) = 0$ when $\|x - x_i\|_2 > \sqrt{m}l$. Here $\varpi_q(\mathbb{R}^m)$ is the set of all polynomials in $\mathbb{R}^m$ with total degree no higher than $q$.*

By the lemma above, there exists a sequence of function $\{v_i\}_{i=1}^n$ such that (1) we have

$$p(x) = \sum_{i=1}^n v_i(x)p(x_i). \tag{H.16}$$

for all $x \in \mathcal{X}$ and $p \in \varpi_q(\mathbb{R}^m)$, and (2) we have $\sum_{i=1}^n |v_i(x)| \leq \exp(2m\gamma_m(q+1))$ for all $x \in \mathcal{X}$. For a fix $x_0$, we have $p(x - x_0) \in \varpi_q(\mathbb{R}^m)$ when $p(x) \in \varpi_q(\mathbb{R}^m)$. By (H.16), we have

$$p(0) = p(x_0 - x_0) = \sum_{i=1}^n v_i(x_0)p(x_i - x_0). \tag{H.17}$$

Similarly, we have $p(x_i - x) \in \varpi_q(\mathbb{R}^m)$ when $p(x) \in \varpi_q(\mathbb{R}^m)$. We apply (H.16) again and have

$$\sum_{i=1}^n v_i(x_0)p(x_i - x_0) = \sum_{i=1}^n \sum_{j=1}^n v_i(x_0)v_j(x_0)p(x_i - x_j) \tag{H.18}$$

for all $x_0 \in \mathcal{X}$ and $p \in \varpi_q(\mathbb{R}^m)$. Combining (H.17) with (H.18), we have

$$p(0) - 2\sum_{i=1}^n v_i(x_0)p(x_i - x_0) + \sum_{i=1}^n \sum_{j=1}^n v_i(x_0)v_j(x_0)p(x_i - x_j) = 0. \tag{H.19}$$

Combining (H.19) with Lemma H.7, we have

$$\left| K(x_0, x_0) - k[\mathcal{X}_n](x_0)^\top K^{-1}[\mathcal{X}_n]k[\mathcal{X}_n](x_0) \right| \leq K(x_0, x_0) - p(0) \tag{H.20}$$

$$- 2\sum_{i=1}^n v_i(x_0)\big[K(x_i, x_0) - p(x_i - x_0)\big] + \sum_{i=1}^n \sum_{j=1}^n v_i(x_0)v_j(x_0)\big[K(x_i, x_j) - p(x_i - x_j)\big]$$

for all $x_0 \in \mathcal{X}$ and $p \in \varpi_q(\mathbb{R}^m)$. By the definition of the kernel $K$, we have $K(x, x') = k_s(x - x')$ when we define $k_s(x) = \exp(-\|x\|_2^2/2)$. Therefore, by (H.20), we have

$$\left| K(x_0, x_0) - k[\mathcal{X}_n](x_0)^\top K^{-1}[\mathcal{X}_n]k[\mathcal{X}_n](x_0) \right| \tag{H.21}$$

$$\leq \sum_{i=1}^n \sum_{j=1}^n \left| v_i(x_0)v_j(x_0)\big[k_s(x_i - x_j) - p(x_i - x_j)\big] \right|$$

$$+ 2\sum_{i=1}^n \left| v_i(x_0)\big[k_s(x_i - x_0) - p(x_i - x_0)\big] \right| + \left| k_s(0) - p(0) \right|$$

for all $x_0 \in \mathcal{X}$ and $p \in \varpi_q(\mathbb{R}^m)$. Combining (H.21) with the fact that $\|x - x'\|_2 \leq \sqrt{m}l$ when $x, x' \in \mathcal{X}$, we have

$$\left| K(x_0, x_0) - k[\mathcal{X}_n](x_0)^\top K^{-1}[\mathcal{X}_n]k[\mathcal{X}_n](x_0) \right| \leq \|k_s - p\|_{L^\infty(B(0, \sqrt{m}l))} \tag{H.22}$$
$$+ 2\sum_{i=1}^n \left| v_i(x_0) \right| \|k_s - p\|_{L^\infty(B(0, \sqrt{m}l))} + \|k_s - p\|_{L^\infty(B(0, \sqrt{m}l))} \sum_{i=1}^n \sum_{j=1}^n \left| v_i(x_0)v_j(x_0) \right|.$$

Here $\|f\|_{L^\infty(B(0, \sqrt{m}l))} = \sup_{\|x\|_2 \leq \sqrt{m}l} |f(x)|$. By Lemma H.8, we have $\sum_{i=1}^n |v_i(x_0)| \leq \exp(2m\gamma_m(q+1))$ for all $x_0 \in \mathcal{X}$. Therefore, by (H.22), we have

$$\left| K(x_0, x_0) - k[\mathcal{X}_n](x_0)^\top K^{-1}[\mathcal{X}_n]k[\mathcal{X}_n](x_0) \right| \leq \left(1 + \exp\left(2m\gamma_m(q+1)\right)\right)^2 \varsigma_p \tag{H.23}$$

for all $x_0 \in \mathcal{X}$ and $p \in \varpi_q(\mathbb{R}^m)$. Here $\varsigma_p = \|k_s - p\|_{L^\infty(B(0, \sqrt{m}l))} = \sup_{\|x\|_2 \leq \sqrt{m}l} |k_s(x) - p(x)|$. Therefore, we only need to bound $\inf_{p \in \varpi_q(\mathbb{R}^m)} \|k_s - p\|_{L^\infty(B(0, \sqrt{m}l))}$ from the above. We have $p(\|x\|_2^2) \in \varpi_q(\mathbb{R}^m)$ when $p \in \varpi_{q/2}(\mathbb{R})$. Since we define $k_s(x) = \exp(-\|x\|_2^2/2)$, by (H.23), we have

$$\left| K(x_0, x_0) - k[\mathcal{X}_n](x_0)^\top K^{-1}[\mathcal{X}_n]k[\mathcal{X}_n](x_0) \right| \tag{H.24}$$
$$\leq 4\exp\left(4m\gamma_m(q+1)\right) \inf_{p \in \varpi_{q/2}(\mathbb{R})} \sup_{x \in [0, ml^2]} \left| \exp(-x/2) - p(x) \right|.$$

By Taylor's Theorem (with Lagrange Remainder), we have

$$\sup_{x \in [0, ml^2]} |\exp(-x/2) - p_t(x)| \leq (ml^2/2)^{[q/2]+1}/\left([q/2]+1\right)! \tag{H.25}$$

when we choose the polynomial $p_t$ as the Taylor polynomial of $\exp(-x/2)$ around zero of degree $[q/2]$. Without loss of generality, we assume $ml^2 > 2$. Since the degree $q$ is an integer, we have $[q/2] \geq (q-1)/2$ and $q \geq 1$ when $\iota \leq C_{\text{app},1}$. Combining (H.25) with Stirling's formula, we have

$$\sup_{x \in [0, ml^2]} \left| \exp(-x/2) - p_t(x) \right| \leq (ml^2/2)^{[q/2]+1} \frac{1}{([q/2]+1)!} \leq (eml^2/2)^{[q/2]+1} \left([q/2]+1\right)^{-[q/2]-1}$$
$$\leq (ml^2/2)^{[q/2]+1} \exp\left((q+1)/2\right)\left((q+1)/2\right)^{-(q+1)/2}. \tag{H.26}$$

We plug (H.26) into (H.24) and have

$$\left| K(x_0, x_0) - k[\mathcal{X}_n](x_0)^\top K^{-1}[\mathcal{X}_n]k[\mathcal{X}_n](x_0) \right| \leq 4l\sqrt{m}\left(\exp(8m\gamma_m + 1)ml^2/(q+1)\right)^{(q+1)/2}.$$

Since we choose $q = [C_{\text{app},1}/\iota]$, we have $q + 1 \geq C_{\text{app},1}/\iota$. Therefore, we have

$$\left| K(x_0, x_0) - k[\mathcal{X}_n](x_0)^\top K^{-1}[\mathcal{X}_n]k[\mathcal{X}_n](x_0) \right| \leq 4l\sqrt{m}\left(\exp(8m\gamma_m + 1)ml^2\iota/C_{\text{app},1}\right)^{(q+1)/2}$$
$$= 4l\sqrt{m}\left(3\gamma_m \exp(8m\gamma_m + 1)ml\iota\right)^{(q+1)/2}$$

where the second equation is induced by the definition of $C_{\text{app},1}$ in Lemma H.8. Therefore, when $\iota < 1/(3\gamma_m \exp(8m\gamma_m + 2)ml)$, we have

$$\left| K(x_0, x_0) - k[\mathcal{X}_n](x_0)^\top K^{-1}[\mathcal{X}_n]k[\mathcal{X}_n](x_0) \right| \leq 4l\sqrt{m}\exp\left(-(q+1)/2\right) \tag{H.27}$$
$$\leq 4l\sqrt{m}\exp\left(-C_{\text{app},1}/(2\iota)\right).$$

Combining (H.15) with (H.27), we have

$$\|(R - S_{\mathcal{X}_n}) \circ f\|_{L_2^\rho(\mathcal{X})} = \sqrt{\int_{\mathcal{X}} \left| f(x) - S_{\mathcal{X}_n}f(x) \right|^2 \mathrm{d}\rho(x)} \leq 4l\sqrt{m}\exp\left(-C_{\text{app},1}/(2\iota)\right) \|f\|_{\mathcal{H}}$$

for $f \in \mathcal{H}$. By the definition of the operator norm, we have

$$\|R - S_{\mathcal{X}_n}\|_{\mathcal{H} \to L_2^\rho(\mathcal{X})} = \sup_{f \in \mathcal{H}/\{0\}} \|(R - S_{\mathcal{X}_n}) \circ f\|_{L_2^\rho(\mathcal{X})} / \|f\|_{\mathcal{H}} \leq 4l\sqrt{m}\exp\left(-C_{\text{app},1}/(2\iota)\right),$$

which concludes the proof of Lemma H.6.

$\square$

# I PROOF OF LEMMAS IN APPENDIX §E

## I.1 PROOF OF LEMMA E.5

*Proof.* In the following part of the proof, we condition on the events good events $\mathcal{E}_1$ and $\mathcal{E}_2$. First, by Lemma J.1, we decompose the difference in value as follows,

$$J(\pi; \widehat{r}^n + u^n, \mathcal{P}^n) - J(\pi; r^*, \mathcal{P}^*) \tag{I.1}$$

$$= \sum_{h=1}^{H} \mathbb{E}_{\pi, \mathcal{P}^n} \left[ \mathbb{E}_{s'_{h+1} \sim \mathcal{P}^n_h(s_h, a_h)} \left[ V^\pi_{h+1}(s'_{h+1}; r^*, \mathcal{P}^*) \right] - \mathbb{E}_{s'_{h+1} \sim \mathcal{P}^*_h(s_h, a_h)} \left[ V^\pi_{h+1}(s'_{h+1}; r^*, \mathcal{P}^*) \right] \right]$$

$$+ \sum_{h=1}^{H} \mathbb{E}_{\pi, \mathcal{P}^n} \left[ u^n_h(s_h, a_h) + \widehat{r}^n_h(s_h, a_h) - r^*_h(s_h, a_h) \right].$$

Since the value function in (I.1) is bounded from the above by $H$, we have

$$J(\pi; \widehat{r}^n + u^n, \mathcal{P}^n) - J(\pi; r^*, \mathcal{P}^*) \tag{I.2}$$

$$\geq \sum_{h=1}^{H} \mathbb{E}_{\pi, \mathcal{P}^n} \left[ u^n_h(s_h, a_h) \right] - \sum_{h=1}^{H} \mathbb{E}_{\pi, \mathcal{P}^n} \left[ f_{h,r}(s_h, a_h) \right] - \sum_{h=1}^{H} \mathbb{E}_{\pi, \mathcal{P}^n} \left[ H f_{h,\mathcal{P}}(s_h, a_h) \right],$$

where $f_{h,\mathcal{P}}(s, a) = \| \mathcal{P}^n_h(\cdot \mid s, a) - \mathcal{P}^*_h(\cdot \mid s, a) \|_1$ and $f_{h,r}(s, a) = |\widehat{r}^n_h(s_h, a_h) - r^*_h(s_h, a_h)|$. In the following part of the proof, we bound the expectation of $f$ in (I.2) from the above. For $h = 1$, by the definition of Event $\mathcal{E}_1$ in (E.5) and $\rho^n_{1,1}$ in (E.3), we have

$$\mathbb{E}_{\pi, \mathcal{P}^n} \left[ f_{1,\mathcal{P}}(s_1, a_1) \right] \leq |\mathcal{A}| \mathbb{E}_{\rho^n_{1,1}} \left[ f_{1,\mathcal{P}}(s, a) \right] \leq |\mathcal{A}| \sqrt{\mathbb{E}_{\rho^n_{1,1}} \left[ f^2_{1,\mathcal{P}}(s, a) \right]} \leq |\mathcal{A}| \zeta / \sqrt{n}, \tag{I.3}$$

when condition on Event $\mathcal{E}_1$. Here the second inequality follows Cauchy-Schwarz inequality. By the same technique in (I.3), we have

$$\mathbb{E}_{\pi, \mathcal{P}^n} \left[ f_{1,r}(s_1, a_1) \right] \leq |\mathcal{A}| \zeta / \sqrt{n}.$$

For $h > 1$, we have $\| f_{h,\mathcal{P}} \|_\infty \leq 2$. Since we have $(s', a') \sim \rho^n_{h,2}$ when $(s, a) \sim \rho^n_{h-1,1}, s' \sim \mathcal{P}^*_{h-1}(\cdot \mid s, a), a' \sim \mathcal{U}(\mathcal{A})$, by Lemma J.2, we have

$$|\mathbb{E}_{\pi, \mathcal{P}^n} [f_{h,\mathbb{P}}(s_h, a_h)]| \leq \beta^n_h \mathbb{E}_{\pi, \mathcal{P}^n} \left[ \left\| k\big( \phi^n_{h-1}(s_{h-1}, a_{h-1}) \big) \right\|_{\Gamma^{-1}_P [\phi^n_{h-1}, \rho^n_{h-1,1}, \lambda, n]} \right],$$

where $\beta^n_h = \sqrt{4\lambda v^2 / c^2_{\min} + 2r^2_{\max}(n |\mathcal{A}| \mathbb{E}_{\rho^n_{h,1}} [f_{h,\mathcal{P}}(s, a)^2] + 4\zeta^2)}$. By the definition of $\mathcal{E}_1$ in (E.5), we have $n\mathbb{E}_{\rho^n_{h,1}}[f_{h,\mathcal{P}}(s, a)^2] \leq \zeta^2$ when condition on $\mathcal{E}_1$. We also have

$$\left\| k\big( \phi^n_h(s_h, a_h) \big) \right\|_{\Gamma^{-1}_P [\phi^n_h, \rho^n_{h,1}, \lambda, n]} \leq 2 \left\| k\big( \phi^n_h(s_h, a_h) \big) \right\|_{\Gamma^{-1}_e [\phi^n_h, \mathcal{D}^n_{h,1}, \lambda]}$$

when condition on Event $\mathcal{E}_2$ in Lemma E.2. By the definition of $u^n_h$ in (4.5), we have

$$\left| \mathbb{E}_{\pi, \mathcal{P}^n} \left[ f_{h,\mathbb{P}}(s_h, a_h) \right] \right| \leq \mathbb{E}_{\pi, \mathcal{P}^n} \left[ u^n_h(s_h, a_h) \right] / (H + 1).$$

By the exactly same method, we have $|\mathbb{E}_{\pi, \mathcal{P}^n} [f_{h,r}(s_h, a_h)]| \leq \mathbb{E}_{\pi, \mathcal{P}^n} [u^n_h(s_h, a_h)] / (H + 1)$. Therefore, we conclude the proof of Lemma E.5 by combing the bound above, (I.2), with (I.3).

$\square$

## I.2 PROOF OF LEMMA E.6

*Proof.* By the definition of $Q^n_h$ in Algorithm 4 (Planning Algorithm), we have $Q^n_h(s, a) = Q^n_h(s, a, \pi^{n+1}; \widehat{r}^n + u^n, \mathcal{P}^n)$ for all $(s, a) \in \mathcal{S} \times \mathcal{A}$, and $\pi^{n+1}$ is the greedy policy with regard to $Q^n_h$. By the definition of the value function, we have

$$V^{\pi^*}_h(s; r^n + u^n, \mathcal{P}^n) - V^n_h(s) = \left\langle Q^{\pi^*}_h(s, \cdot; r^n + u^n, \mathcal{P}^n) - Q^n_h(s, \cdot), \pi^*_h(\cdot \mid s) \right\rangle_{\mathcal{A}} \tag{I.4}$$

$$+ \left\langle Q^n_h(s, \cdot), \pi^*_h(\cdot \mid s) - \widehat{\pi}_h(\cdot \mid s) \right\rangle_{\mathcal{A}}.$$

Since $\pi^{n+1}$ is the greedy policy with regard to $Q_h^n$, we have $\langle Q_h^n(s,\cdot), \pi_h^*(\cdot \mid s) - \widehat{\pi}_h(\cdot \mid s)\rangle_\mathcal{A} \leq 0$. We also have $Q_h^{\pi^*}(s,a; r^n+u^n, \mathcal{P}^n) - Q_h^n(s,a) = \mathbb{E}_{s' \sim \mathcal{P}_h^n(\cdot|s,a)}[V_{h+1}^{\pi^*}(s'; r^n+u^n, \mathcal{P}^n) - Q_{h+1}^n(s')]$. Therefore, taking expectation over $\mathcal{P}^n$ and $\pi^*$ in both sides of (I.4), we have

$$\mathbb{E}_{\mathcal{P}^n, \pi^*}\big[V_h^{\pi^*}(s_h; r^n+u^n, \mathcal{P}^n) - V_h^n(s_h)\big] \leq \mathbb{E}_{\mathcal{P}^n, \pi^*}\big[V_{h+1}^{\pi^*}(s_{h+1}; r^n+u^n, \mathcal{P}^n) - V_{h+1}^n(s_{h+1})\big].$$

Using induction on $h$, we have

$$V_1^{\pi^*}(s_\text{init}; r^n+u^n, \mathcal{P}^n) - V_1^n(s_\text{init}) \leq \mathbb{E}_{\mathcal{P}^n, \pi^*}[V_{H+1}^{\pi^*}(s_{H+1}; r^n+u^n, \mathcal{P}^n) - V_{H+1}^n(s_{H+1})] = 0.$$

Since we have $V_1^\pi(s_\text{init}; r^n+u^n, \mathcal{P}^n) = J(\pi^*; r^n+u^n, \mathcal{P}^n)$ and $V_1^n(s_\text{init}) = J(\pi^{n+1}; r^n+u^n, \mathcal{P}^n)$, we conclude the proof of Lemma E.6. $\qquad\square$

### I.3 PROOF OF LEMMA E.7

*Proof.* The following lemma shows that the value function defined by the estimated model is bounded from the above by the value function defined by the true model.

**Lemma I.1** (Bounded optimism in Each Iteration). *Following the same condition with Theorem 5.3, when condition on the good events $\mathcal{E}_1$ and $\mathcal{E}_2$, which are defined in (E.5), (E.6) and (E.7), we have*

$$J(\pi; r^* + u^n + u^{*,n}, \mathcal{P}^*) - J(\pi; \widehat{r}^n + u^n, \mathcal{P}^n) \geq -(2H^2 + 3H + 1)|\mathcal{A}|\zeta/\sqrt{n}$$

*for any policy $\pi$, where the bonus $u^n = \{u_h^n\}_{h=1}^H$ is defined in Lemma E.5. Here the underlying bonus $u^{*,n} = \{u_h^{*,n}\}_{h=1}^H$ is defined as*

$$u_h^{*,n}(s,a) = \min\Big\{4H^2 + 6H + 2, \beta_1 \big\|k\big(\phi_h^*(s_h, a_h)\big)\big\|_{\Gamma_e^{-1}[\phi_h^*, \mathcal{D}_{h,0}^n, \lambda]}\Big\},$$

*where $\beta_1 = (4H^2 + 6H + 2)\sqrt{4\lambda v^2/c_\text{min}^2 + 4r_\text{max}^2|\mathcal{A}|\beta^2\zeta^2 d_\text{eff}}$ and $\beta$ is defined in Lemma E.5.*

*Proof.* See Appendix §I.4 for a detailed proof. $\qquad\square$

By Lemma I.1, and the definition of the expected total reward, we have

$$\sum_{n=1}^N \big[J(\pi^{n+1}; r^n+u^n, \mathcal{P}^n) - J(\pi^{n+1}; r^*, \mathcal{P}^*)\big] \tag{I.5}$$

$$\leq \sum_{n=1}^N (2H^2 + 3H + 1)|\mathcal{A}|\zeta/\sqrt{n} + \sum_{n=1}^N J(\pi^{n+1}; u^n + u^{*,n}, \mathcal{P}^*)$$

However, the bonus defined by the learned feature might vary in each episode, which make it difficult to bound from the above. The following lemma connects the bonus defined by the learned feature with the bonus defined by the true feature.

**Lemma I.2** (Bonus Equivalence for the True Model). *Following the same condition with Theorem 5.3, when condition on $\mathcal{E}_1$ and $\mathcal{E}_2$, which are defined in (E.5), (E.6), and (E.7), we have*

$$J(\pi; u^n, \mathcal{P}^*) \leq 2|\mathcal{A}|\beta d_\text{eff}^{1/2}/\sqrt{n} + J(\pi; u^{*,n}, \mathcal{P}^*)$$

*for any policy $\pi$ and $n \geq 2$, where the bonus $u^n = \{u_h^n\}_{h=1}^H$ is defined in Lemma E.5 and the underlying bonus $u^{*,n} = \{u_h^{*,n}\}_{h=1}^H$ is defined in Lemma I.1.*

*Proof.* See Appendix §I.5 for a detailed proof. $\qquad\square$

Combining (I.5) with Lemma I.2, we have

$$\sum_{n=1}^N \big[J(\pi^n; r^n+u^n, \mathcal{P}^n) - J(\pi^n; r^*, \mathcal{P}^*)\big] \tag{I.6}$$

$$\leq \sum_{n=1}^N (2H^2 + 3H + 1)|\mathcal{A}|\zeta/\sqrt{n} + \sum_{n=1}^N 2|\mathcal{A}|\beta d_\text{eff}^{1/2}/\sqrt{n} + 2\sum_{n=1}^N J(\pi^{n+1}; u^{*,n}, \mathcal{P}^*)$$

$$\leq 32H^2|\mathcal{A}|\zeta\sqrt{d_\text{eff}N} + 2\sum_{n=1}^N J(\pi^{n+1}; u^{*,n}, \mathcal{P}^*).$$

Therefore, it remains to bound $J(s_{\text{init}}, \pi; u^{*,n}, \mathcal{P}^*)$. Since the bonus $u^{*,n}$ are bounded by $4H^2 + 6H + 2$, it is a $12H^2$-subGaussian random variable. Therefore, by Hoeffding's inequality and the definition of the value function, we have

$$\sum_{n=1}^{N} J(\pi^{n+1}; u^{*,n}, \mathcal{P}^*) = \sum_{h=1}^{H} \sum_{n=1}^{N} \mathbb{E}_{\pi^{n+1}, \mathcal{P}^*}\left[u_h^{*,n}(s_h, a_h)\right] \tag{I.7}$$

$$\leq 12H^2 \log(10H/\delta)\sqrt{N} + \sum_{h=1}^{H} \sum_{n=1}^{N} u_h^{*,n}(s_{h,0}^{n+1}, a_{h,0}^{n+1})$$

with probability at least $1 - \delta/10$. Combining the definition of the underlying bonus in Lemma I.1 with Cauchy-Schwarz inequality, we have

$$\sum_{h=1}^{H} \sum_{n=1}^{N} u_h^{*,n}(s_{h,0}^{n+1}, a_{h,0}^{n+1}) \leq \beta_1 \sum_{h=1}^{H} \sum_{n=1}^{N} \left\| k\left(\phi_h^*(s_{h,0}^{n+1}, a_{h,0}^{n+1})\right) \right\|_{\Gamma_e^{-1}[\phi_h^*, \mathcal{D}_{h,0}^n, \lambda]} \tag{I.8}$$

$$\leq \beta_1 \sqrt{N} \sum_{h=1}^{H} \sqrt{\sum_{n=1}^{N} \left\| k\left(\phi_h^*(s_{h,0}^{n+1}, a_{h,0}^{n+1})\right) \right\|_{\Gamma_e^{-1}[\phi_h^*, \mathcal{D}_{h,0}^n, \lambda]}^2}.$$

For simplicity, we define $x_h^n = \phi_h^*(s_{h,0}^n, a_{h,0}^n)$, $K_h^n = [K(x_h^{\tau_1}, x_h^{\tau_2})]_{\tau_1, \tau_2 = 1}^n$, and

$$k_h^n = \left(K(x_h^1, x_h^{n+1}), \ldots, K(x_h^n, x_h^{n+1})\right).$$

By Lemma D.3, we have

$$\left\| k\left(\phi_h^*(s_{h,0}^{n+1}, a_{h,0}^{n+1})\right) \right\|_{\Gamma_e^{-1}[\phi_h^*, \mathcal{D}_{h,0}^n, \lambda]}^2 = \left(K(x_h^{n+1}, x_h^{n+1}) - k_h^{n,\top}(\lambda I + K_h^n)^{-1} k_h^n\right)/\lambda.$$

We also define $a_h^n = (K(x_h^{n+1}, x_h^{n+1}) - k_h^{n,\top}(\lambda I + K_h^n)^{-1} k_h^n)/\lambda$. We then have

$$\det(I + K_h^{n+1}/\lambda) = \det \begin{pmatrix} I + K_h^n/\lambda & k_h^n/\lambda \\ k_h^{n,\top}/\lambda & 1 + K(x_h^{n+1}, x_h^{n+1})/\lambda \end{pmatrix}$$

$$= \det \begin{pmatrix} I & 0 \\ b^\top & 1 \end{pmatrix} \det \begin{pmatrix} I + K_n/\lambda & 0 \\ 0 & 1 + a_n \end{pmatrix} \det \begin{pmatrix} I & b \\ 0 & 1 \end{pmatrix} = \det(I + K_h^n/\lambda)(1 + a_h^n),$$

where $b = (\lambda I + K_h^n)^{-1} k_h^n$. We also have $a_h^n \leq 1$. Since $x \leq e \log(1 + x)$ when $x \leq e - 1$, we have $a_h^n \leq e \log(I + K_h^{n+1}/\lambda) - e \log(I + K_h^n/\lambda)$, and $\sum_{n=1}^{N} a_h^n \leq e \log(1 + K_h^{N+1}/\lambda) \leq e d_{\text{eff}}$ by the definition of $d_{\text{eff}}$ in Definition E.4. Therefore, by (I.8), we have $\sum_{h=1}^{H} \sum_{n=1}^{N} u_h^{*,n}(s_{h,0}^{n+1}, a_{h,0}^{n+1}) \leq \beta_1 H \sqrt{eN d_{\text{eff}}}$. Combining (I.6) with (I.7), we have

$$\sum_{n=1}^{N} \left[J(\pi^n; r^n + u^n, \mathcal{P}^n) - J(\pi^n; r^*, \mathcal{P}^*)\right]$$

$$\leq 32H^2 |\mathcal{A}| \zeta \sqrt{d_{\text{eff}} N} + 12H^2 \log(10H/\delta)\sqrt{N} + \beta_1 H \sqrt{eN d_{\text{eff}}}$$

$$\leq 46H^2 |\mathcal{A}| \zeta \beta_1 \sqrt{d_{\text{eff}} N} \log(10H/\delta),$$

which concludes the proof of Lemma E.7.

$\square$

### I.4 PROOF OF LEMMA I.1

*Proof.* In the following part of the proof, we condition on $\mathcal{E}_1$ and $\mathcal{E}_2$. First, by Lemma J.1, we decompose the difference in value as follows,

$$J(\pi; r^* + u^n + u^{*,n}, \mathcal{P}^*) - J(\pi; \widehat{r}^n + u^n, \mathcal{P}^n) \tag{I.9}$$

$$= \sum_{h=1}^{H} \mathbb{E}_{\pi, \mathcal{P}^*}\left[\mathbb{E}_{s'_{h+1} \sim \mathcal{P}_h^*(s_h, a_h)}\left[V_{h+1}^\pi(s'_{h+1}; \widehat{r}^n + u^n, \mathcal{P}^n)\right] - \mathbb{E}_{s'_{h+1} \sim \mathcal{P}_h^n(s_h, a_h)}\left[V_{h+1}^\pi(s'_{h+1}; \widehat{r}^n + u^n, \mathcal{P}^n)\right]\right]$$

$$+ \sum_{h=1}^{H} \mathbb{E}_{\pi, \mathcal{P}^*}\left[u_h^{*,n}(s_h, a_h) + r_h^*(s_h, a_h) - \widehat{r}_h^n(s_h, a_h)\right].$$

Since $\|\widehat{r}^n + u^n\|_\infty \leq (2H + 3)$, the value function in (I.9) is bounded from the above, and we have

$$J(\pi; r^* + u^n + u^{*,n}, \mathcal{P}^*) - J(\pi; \widehat{r}^n + u^n, \mathcal{P}^n) \tag{I.10}$$

$$\geq \sum_{h=1}^H \mathbb{E}_{\pi,\mathcal{P}^*}\left[u_h^{*,n}(s_h, a_h)\right] - \sum_{h=1}^H \mathbb{E}_{\pi,\mathcal{P}^*}\left[f_{h,r}(s_h, a_h)\right] - \sum_{h=1}^H \mathbb{E}_{\pi,\mathcal{P}^*}\left[(2H^2 + 3H)f_{h,\mathcal{P}}(s_h, a_h)\right],$$

where $f_{h,\mathcal{P}}(s, a) = \|\mathcal{P}_h^n(\cdot \mid s, a) - \mathcal{P}_h^*(\cdot \mid s, a)\|_1$ and $f_{h,r}(s, a) = |\widehat{r}_h^n(s_h, a_h) - r_h^*(s_h, a_h)|$ are defined for simplicity. In the following part of the proof, we bound the expectation of $f$ in (I.10) from the above. For $h = 1$, by the definition of Event $\mathcal{E}_1$ in (E.5) and $\rho_{1,1}^n$ in (E.3), we have

$$\mathbb{E}_{\pi,\mathcal{P}^*}\left[f_{1,\mathcal{P}}(s_1, a_1)\right] \leq |\mathcal{A}| \, \mathbb{E}_{\rho_{1,1}^n}\left[f_{1,\mathcal{P}}(s, a)\right] \leq |\mathcal{A}| \sqrt{\mathbb{E}_{\rho_{1,1}^n}\left[f_{1,\mathcal{P}}^2(s, a)\right]} \leq |\mathcal{A}| \zeta/\sqrt{n} \tag{I.11}$$

when condition on $\mathcal{E}_1$. Here the second inequality follows Cauchy-Schwarz inequality. By the same technique in (I.11), we have

$$\mathbb{E}_{\pi,\mathcal{P}^*}\left[f_{1,r}(s_1, a_1)\right] \leq |\mathcal{A}| \zeta/\sqrt{n}.$$

For $h > 1$, we have $\|f_{h,\mathcal{P}}\|_\infty \leq 2$. Since $(s', a') \sim \rho_{h,1}^n$ when $(s, a) \sim \rho_{h-1,0}^n, s' \sim \mathcal{P}_{h-1}^*(\cdot \mid s, a), a' \sim \mathcal{U}(\mathcal{A})$, by Lemma J.3, we have

$$\left|\mathbb{E}_{\pi,\mathcal{P}^*}\left[f_{h,\mathbb{P}}(s_h, a_h)\right]\right| \leq \beta_h^{n,\prime}\mathbb{E}_{\pi,\mathcal{P}^*}\left[\left\|k(\phi_{h-1}^*(s_{h-1}, a_{h-1}), \cdot)\right\|_{\Gamma_p^{-1}[\phi_{h-1}^*, \rho_{h-1,0}^n, \lambda, n]}\right], \tag{I.12}$$

where $\beta_h^{n,\prime} = \sqrt{4\lambda v^2/c_{\min}^2 + r_{\max}^2 n \, |\mathcal{A}| \, \mathbb{E}_{(s_h, a_h) \sim \rho_{h,1}^n}[f_{h,\mathbb{P}}(s_h, a_h)^2]}$. By the definition of $\mathcal{E}_1$ in (E.5), we have $n\mathbb{E}_{(s_h, a_h) \sim \rho_{h,1}^n}[f_{h,\mathbb{P}}(s_h, a_h)^2] \leq \zeta^2$ when $\mathcal{E}_1$ holds. We also have

$$\left\|k\left(\phi_h^*(s_h, a_h)\right)\right\|_{\Gamma_p^{-1}[\phi_h^*, \rho_{h,0}^n, \lambda, n]} \leq 2\left\|k\left(\phi_h^*(s_h, a_h)\right)\right\|_{\Gamma_e^{-1}[\phi_h^*, \mathcal{D}_{h,0}^n, \lambda]}$$

when condition on Event $\mathcal{E}_2$ defined in (E.6), (E.7). By the definition of $u_h^{*,n}$ in Lemma I.1 and (I.12), we have

$$\left|\mathbb{E}_{\pi,\mathcal{P}^*}\left[f_{h,\mathbb{P}}(s_h, a_h)\right]\right| \leq \mathbb{E}_{\pi,\mathcal{P}^*}\left[u_{h-1}^{*,n}(s_{h-1}, a_{h-1})\right]/(2H^2 + 3H + 1).$$

By the same method, we have $|\mathbb{E}_{\pi,\mathcal{P}^*}[f_{h,r}(s_h, a_h)]| \leq \mathbb{E}_{\pi,\mathcal{P}^*}[u_{h-1}^{*,n}(s_{h-1}, a_{h-1})]/(2H^2 + 3H + 1)$. Therefore, we conclude the proof of Lemma I.1 by combing the bound above, (I.10), with (I.11). □

### I.5 PROOF OF LEMMA I.2

*Proof.* In the following part of the proof, we condition on the good events $\mathcal{E}_1$ and $\mathcal{E}_2$ defined in (E.5), (E.6), and (E.7). By the definition of the value function, we have

$$J(\pi; u^n, \mathcal{P}^*) = \underbrace{\mathbb{E}_\pi\left[u_1^n(s_{\text{init}}, a_h)\right]}_{\text{Term (a)}} + \underbrace{\sum_{h=2}^H \mathbb{E}_{\pi,\mathcal{P}^*}\left[u_h^n(s_h, a_h)\right]}_{\text{Term (b)}}. \tag{I.13}$$

Now we bound the terms in (I.13) from the above separately.

**Term (a).** By the definition of $\rho_{1,1}^n$ in (E.3), Algorithm 3 (Sampling Scheme), and Cauchy-Schwarz inequality, we have

$$\mathbb{E}_\pi\left[u_1^n(s_{\text{init}}, a_h)\right] \leq |\mathcal{A}| \, \mathbb{E}_{\rho_{1,1}^n}\left[u_1^n(s, a)\right] \leq |\mathcal{A}| \sqrt{\mathbb{E}_{\rho_{1,1}^n}\left[\left(u_1^n(s, a)\right)^2\right]}. \tag{I.14}$$

By the definition of $u_h^n$ in (4.5) and $d_{\text{eff}}$ in Definition E.4, when Event $\mathcal{E}_2$ defined in Lemma E.2 holds, we have

$$\mathbb{E}_{\rho_{h,1}^n}\left[\left(u_h^n(s, a)\right)^2\right] \leq \beta^2 \mathbb{E}_{\rho_{h,1}^n}\left[\left\|k\left(\phi_h^n(s, a)\right)\right\|_{\Gamma_e^{-1}[\phi_h^n, \mathcal{D}_{h,1}^n, \lambda]}^2\right]$$

$$\leq 4\beta^2 \mathbb{E}_{\rho_{h,1}^n}\left[\left\|k\left(\phi_h^n(s, a)\right)\right\|_{\Gamma_p^{-1}[\phi_h^n, \rho_{h,1}^n, \lambda, n]}^2\right] \leq 4\beta^2 d_{\text{eff}}/n.$$

Therefore, by (I.14), we have $\mathbb{E}_\pi[u_1^n(s_{\text{init}}, a_1)] \leq 2\,|\mathcal{A}|\,\beta d_{\text{eff}}^{1/2}/\sqrt{n}$.

**Term (b).** By the definition of $u_h^n$ in Lemma E.5, we have $\|u^n\|_\infty \leq 2H+2$. By the definition of $\rho_{h,0}^n$ and $\rho_{h,1}^n$ in (E.3), we have $(s', a') \sim \rho_{h,1}^n$ when $(s, a) \sim \rho_{h-1,0}^n$, $s' \sim \mathcal{P}_{h-1}^*(\cdot \mid s, a), a' \sim \mathcal{U}(\mathcal{A})$. Therefore, by Lemma J.3, when we condition on Event $\mathcal{E}_2$ defined in Lemma E.2, we have

$$
\begin{aligned}
\mathbb{E}_{\pi, \mathcal{P}^*}\left[\left(u_h^n(s_h, a_h)\right)^2\right] &\leq \beta_h^{n,*} \mathbb{E}_{\pi, \mathcal{P}^*}\left[\left\|k\big(\phi_{h-1}^*(s_{h-1}, a_{h-1})\big)\right\|_{\Gamma_p^{-1}[\phi_{h-1}^*, \rho_{h-1,0}^n, \lambda, n]}\right] \qquad (\text{I}.15) \\
&\leq 2\beta_h^{n,*} \mathbb{E}_{\pi, \mathcal{P}^*}\left[\left\|k\big(\phi_{h-1}^*(s_{h-1}, a_{h-1})\big)\right\|_{\Gamma_e^{-1}[\phi_{h-1}^*, \mathcal{D}_{h-1,0}^n, \lambda]}\right],
\end{aligned}
$$

where $\beta_h^{n,*} = \sqrt{4\lambda(H+1)^2 v^2/c_{\min}^2 + r_{\max}^2 n\,|\mathcal{A}|\,\mathbb{E}_{\rho_{h,1}^n}[u_h^n(s, a)^2]}$. By the definition of $u_h^n$ in (4.5), and $d_{\text{eff}}$ in Definition E.4, when we condition on Event $\mathcal{E}_2$ defined in Lemma E.2, we have

$$
\begin{aligned}
\mathbb{E}_{\rho_{h,1}^n}[u_h^n(s, a)^2] &\leq \beta^2 \mathbb{E}_{\rho_{h,1}^n}\left[\left\|k\big(\phi_h^n(s, a)\big)\right\|_{\Gamma_e^{-1}[\phi_h^n, \mathcal{D}_h^n, \lambda]}\right] \\
&\leq 2\beta^2 \mathbb{E}_{\rho_{h,1}^n}\left[\left\|k\big(\phi_h^n(s, a)\big)\right\|_{\Gamma_p^{-1}[\phi_h^n, \rho_{h,1}^n, \lambda, n]}\right] \leq 2\beta^2 d_{\text{eff}}/n.
\end{aligned}
$$

Therefore, we have $\beta_1 \geq 2\beta_h^{n,*}$, where $\beta_1$ is defined in Lemma I.1. By (I.15) and the definition of $u_{h-1}^{*,n}(s_{h-1}, a_{h-1})$ in Lemma I.1, we have $\mathbb{E}_{\pi, \mathcal{P}^*}[u_h^n(s_h, a_h)] \leq \mathbb{E}_{\pi, \mathcal{P}^*}[u_{h-1}^{*,n}(s_{h-1}, a_{h-1})]$.

We conclude the proof of Lemma I.2 by combining the bound on Term (a) and Term (b). □

# J    PROOF OF AUXILIARY LEMMAS

## J.1    PROOF OF LEMMA J.1

In this subsection, we provide the proof of the simulation lemma. We first state it below.

**Lemma J.1** (Simulation Lemma). *Let $V_h(s; r, \mathcal{P}, \pi)$ be the value function defined in (2.1). For the transition kernels $\mathcal{P} = \{\mathcal{P}_h\}_{h=1}^H$, $\mathcal{P}^* = \{\mathcal{P}_h^*\}_{h=1}^H$ and the reward functions $r = \{r_h\}_{h=1}^H$, we have*

$$
\begin{aligned}
V_h^\pi(s; r+u, \mathcal{P}) - V_h^\pi(s; r, \mathcal{P}^*) &= \sum_{\bar{h}=h}^H \mathbb{E}_{\pi, \mathcal{P}}\left[u_{\bar{h}}(s_{\bar{h}}, a_{\bar{h}}) \mid s_h = s\right] \\
&+ \sum_{\bar{h}=h}^H \mathbb{E}_{\pi, \mathcal{P}}\left[\mathbb{E}_{1, \bar{h}}\left[V_{\bar{h}+1}^\pi(s'_{\bar{h}+1}; r, \mathcal{P}^*)\right] - \mathbb{E}_{2, \bar{h}}\left[V_{\bar{h}+1}^\pi(s'_{\bar{h}+1}; r, \mathcal{P}^*)\right] \mid s_h = s\right],
\end{aligned}
$$

*where $\mathbb{E}_{1,h}[\cdot] = \mathbb{E}_{s'_{h+1} \sim \mathcal{P}_h(s_h, a_h)}[\cdot]$ and $\mathbb{E}_{2,h}[\cdot] = \mathbb{E}_{s'_{h+1} \sim \mathcal{P}_h^*(s_h, a_h)}[\cdot]$. Here $\mathbb{E}_{\pi, \mathcal{P}}$ is the expectation taken over the trajectory induced by the policy $\pi = \{\pi_h\}_{h=1}^H$ and the transition kernel $\mathcal{P}$.*

*Proof.* By the definition of $V_h(s; r, \mathcal{P}, \pi)$ in (2.1), we have

$$
\begin{aligned}
V_h^\pi(s; r+u, \mathcal{P}) - V_h^\pi(s; r, \mathcal{P}^*) &= V_h^\pi(s; u, \mathcal{P}) + V_h^\pi(s; r, \mathcal{P}) - V_h^\pi(s; r, \mathcal{P}^*) \qquad (\text{J}.1) \\
&= \sum_{\bar{h}=h}^H \mathbb{E}_{\pi, \mathcal{P}}\left[u_{\bar{h}}(s_{\bar{h}}, a_{\bar{h}}) \mid s_h = s\right] + V_h^\pi(s; r, \mathcal{P}) - V_h^\pi(s; r, \mathcal{P}^*).
\end{aligned}
$$

Here $\mathbb{E}_{\pi, \mathcal{P}}$ is the expectation taken over the trajectory induced by policy $\pi$ and the transition kernel $\mathcal{P}$. By the definition of the value function $V_{\bar{h}}(s; r, \mathcal{P}, \pi)$ in (2.1), we have

$$
\begin{aligned}
V_{\bar{h}}^\pi(s_{\bar{h}}^{\text{true}}; r, \mathcal{P}) &= \mathbb{E}_{\pi, \mathcal{P}}\left[r_{\bar{h}}(s_{\bar{h}}, a_{\bar{h}}) + V_{\bar{h}+1}^\pi(s_{\bar{h}+1}; r, \mathcal{P}) \mid s_{\bar{h}} = s_{\bar{h}}^{\text{true}}\right] \qquad (\text{J}.2) \\
&= \sum_{a \in \mathcal{A}} r_{\bar{h}}(s_{\bar{h}}^{\text{true}}, a) \pi_{\bar{h}}(a \mid s_{\bar{h}}^{\text{true}}) + \mathbb{E}_{\pi, \mathcal{P}}\left[V_{\bar{h}+1}^\pi(s_{\bar{h}+1}; r, \mathcal{P}) \mid s_{\bar{h}} = s_{\bar{h}}^{\text{true}}\right].
\end{aligned}
$$

By (J.2), we have

$$
\begin{aligned}
V_{\bar{h}}^\pi(s_{\bar{h}}^{\text{true}}; r, \mathcal{P}) &- V_{\bar{h}}^\pi(s_{\bar{h}}^{\text{true}}; r, \mathcal{P}^*) \qquad (\text{J}.3) \\
&= \mathbb{E}_{\pi, \mathcal{P}}\left[V_{\bar{h}+1}^\pi(s_{\bar{h}+1}; r, \mathcal{P}) \mid s_{\bar{h}} = s_{\bar{h}}^{\text{true}}\right] - \mathbb{E}_{\pi, \mathcal{P}^*}\left[V_{\bar{h}+1}^\pi(s_{\bar{h}+1}; r, \mathcal{P}^*) \mid s_{\bar{h}} = s_{\bar{h}}^{\text{true}}\right].
\end{aligned}
$$

Combining the property of the expectation with (J.3), we have

$$V_{\bar{h}}^{\pi}(s_{\bar{h}}^{\text{true}}; r, \mathcal{P}) - V_{\bar{h}}^{\pi}(s_{\bar{h}}^{\text{true}}; r, \mathcal{P}^*) = \mathbb{E}_{\pi, \mathcal{P}}\big[V_{\bar{h}+1}^{\pi}(s_{\bar{h}+1}; r, \mathcal{P}) - V_{\bar{h}+1}^{\pi}(s_{\bar{h}+1}; r, \mathcal{P}^*) \mid s_{\bar{h}} = s_{\bar{h}}^{\text{true}}\big]$$

$$+ \mathbb{E}_{\pi, \mathcal{P}}\big[V_{\bar{h}+1}^{\pi}(s_{\bar{h}+1}; r, \mathcal{P}^*) \mid s_{\bar{h}} = s_{\bar{h}}^{\text{true}}\big] - \mathbb{E}_{\pi, \mathcal{P}^*}\big[V_{\bar{h}+1}^{\pi}(s_{\bar{h}+1}; r, \mathcal{P}^*) \mid s_{\bar{h}} = s_{\bar{h}}^{\text{true}}\big]. \qquad (J.4)$$

Taking expectation in both sides of (J.4) and summing it up from $\bar{h} = h$ to $H$, we have

$$V_h^{\pi}(s; r, \mathcal{P}) - V_h^{\pi}(s; r, \mathcal{P}^*) \qquad (J.5)$$

$$= \sum_{\bar{h}=h}^{H} \mathbb{E}_{\pi, \mathcal{P}}\Big[\mathbb{E}_{1, \bar{h}}\big[V_{\bar{h}+1}^{\pi}(s_{\bar{h}+1}'; r, \mathcal{P}^*)\big] - \mathbb{E}_{2, \bar{h}}\big[V_{\bar{h}+1}^{\pi}(s_{\bar{h}+1}'; r, \mathcal{P}^*)\big] \mid s_h = s\Big],$$

where $\mathbb{E}_{1,h}[\cdot] = \mathbb{E}_{s_{h+1}' \sim \mathcal{P}_h(s_h, a_h)}[\cdot]$ and $\mathbb{E}_{2,h}[\cdot] = \mathbb{E}_{s_{h+1}' \sim \mathcal{P}_h^*(s_h, a_h)}[\cdot]$. We conclude the proof of Lemma J.1 by combining (J.1) with (J.5). □

## J.2 Proof of Lemma J.2

In this subsection, we provide the proof of the one-step backward inequality for the learned model. We first state this lemma below.

**Lemma J.2** (One-Step Backward for the Learned Model). *Let $\rho_{h-1}^n$ be a distribution over $\mathcal{S} \times \mathcal{A}$. We assume that the transition kernels $\mathcal{P}^n = \{\mathcal{P}_h^n\}_{h=1}^H$ and $\mathcal{P}^* = \{\mathcal{P}_h^*\}_{h=1}^H$ satisfy*

$$\mathcal{P}_h^n(s' \mid s, a) = \Big\langle k\big(\phi_h^n(s, a)\big), k\big(\psi_{h+1}^n(s')\big) \Big\rangle_{\mathcal{H}} \Big/ c[\phi_h^n, \psi_{h+1}^n](s, a),$$

*and* $\qquad \mathcal{P}_h^*(s' \mid s, a) = \Big\langle k\big(\phi_h^*(s, a)\big), k\big(\psi_{h+1}^*(s')\big) \Big\rangle_{\mathcal{H}} \Big/ c[\phi_h^*, \psi_{h+1}^*](s, a),$

*where $c[\phi, \psi](s, a)$ is the normalization function defined in (E.1) and $k : \mathcal{X} \to \mathcal{H}$ is the feature map of the RKHS $\mathcal{H}$. We assume further that $n\mathbb{E}_{(s,a) \sim \rho_{h-1}^n}[\text{TV}^2(\mathcal{P}_{h-1}^n(\cdot \mid s, a), \mathcal{P}_{h-1}^*(\cdot \mid s, a))] \leq \zeta^2$, and the non-negative function $g$ satisfies $\|g\|_{\infty} \leq B$. For any policy $\pi$, we have*

$$\Big|\mathbb{E}_{\pi, \mathcal{P}^n}\big[g(s_h, a_h)\big]\Big| \leq \mathbb{E}_{\pi, \mathcal{P}^n}\Big[\beta_l \big\|k\big(\phi_{h-1}^n(s_{h-1}, a_{h-1})\big)\big\|_{\Gamma_p^{-1}[\phi_{h-1}^n, \rho_{h-1}^n, \lambda, n]}\Big].$$

*Here the operator $\Gamma_p[\phi_{h-1}^n, \rho_{h-1}^n, \lambda, n]$ is defined in (E.4), $\mathbb{E}_{\pi, \mathcal{P}}$ is the expectation taken over the trajectory induced by the policy $\pi = \{\pi_h\}_{h=1}^H$ and the transition kernel $\mathcal{P}$, $\beta$ is defined as*

$$\beta_l = \sqrt{\lambda B^2 v^2 / c_{\min}^2 + 2r_{\max}^2 \Big(n|\mathcal{A}|\, \mathbb{E}_{\rho_{h-1}^n, \mathcal{P}_h^*, \mathcal{U}(\mathcal{A})}\big[g^2(s_h, a_h)\big] + B^2 \zeta^2\Big)},$$

*and the expectation $\mathbb{E}_{\rho_{h-1}^n, \mathcal{P}_h^*, \mathcal{U}(\mathcal{A})}[g(s_h, a_h)]$ is defined as*

$$\mathbb{E}_{\rho_{h-1}^n, \mathcal{P}_{h-1}^*, \mathcal{U}(\mathcal{A})}\big[g(s_h, a_h)\big] = \mathbb{E}_{(s_{h-1}, a_{h-1}) \sim \rho_{h-1}^n, s_h \sim \mathcal{P}_{h-1}^*(\cdot | s_{h-1}, a_{h-1}), a_h \sim \mathcal{U}(\mathcal{A})}\big[g(s_h, a_h)\big].$$

*Proof.* By the structure of the transition and the property of conditional expectation, we have

$$\Big|\mathbb{E}_{\pi, \mathcal{P}^n}\big[g(s_h, a_h)\big]\Big| \qquad (J.6)$$

$$= \Big|\mathbb{E}_{\pi, \mathcal{P}^n}\Big[\int_s \mathcal{P}_{h-1}^n(s \mid s_{h-1}, a_{h-1}) \sum_{a \in \mathcal{A}} \pi_h(a \mid s) g(s, a) \mathrm{d}s\Big]\Big|$$

$$= \Big|\mathbb{E}_{\pi, \mathcal{P}^n}\Big[\int_{s \in \mathcal{S}} K\big(\phi_{h-1}^n(s_{h-1}, a_{h-1}), \psi_h^n(s)\big) \bar{g}(s, \pi_h) \mathrm{d}s / c[\phi_{h-1}^n, \psi_h^n](s_{h-1}, a_{h-1})\Big]\Big|,$$

where the last equation follows from the structure of $\mathcal{P}_{h-1}^n$, and $\bar{g}(s, \pi)$ is defined as

$$\bar{g}(s, \pi) = \sum_{a \in \mathcal{A}} \pi(a \mid s) g(s, a). \qquad (J.7)$$

Combining (J.6) with the lower bound of the normalization constant, we have

$$\Big|\mathbb{E}_{\pi, \mathcal{P}^n}\big[g(s_h, a_h)\big]\Big| \leq \mathbb{E}_{\pi, \mathcal{P}^n}\Big[\Big|\int_{s \in \mathcal{S}} K\big(\phi_{h-1}^n(s_{h-1}, a_{h-1}), \psi_h^n(s)\big) \bar{g}(s, \pi_h) \mathrm{d}s\Big| \Big/ c_{\min}\Big]. \qquad (J.8)$$

For a bounded function $g$, we define the functional $\chi[\psi_h^n, \pi, g] : \mathcal{H} \to \mathbb{R}$ as

$$\chi[\psi_h^n, \pi, g]g' = \int_{s \in \mathcal{S}} \left\langle g', k\big(\psi_h^n(s)\big) \right\rangle_{\mathcal{H}} \sum_{a \in \mathcal{A}} \pi_h(a \mid s)g(s,a)\mathrm{d}s. \tag{J.9}$$

The functional $\chi[\psi_h^n, \pi, g]$ is linear. Moreover, for any $g' \in \mathcal{H}$, we have

$$\left| \chi[\psi_h^n, \pi, g]g' \right| \leq \int_{s \in \mathcal{S}} \|g'\|_{\mathcal{H}} \left\|k\big(\psi_h^n(s)\big)\right\|_{\mathcal{H}} \left| \sum_{a \in \mathcal{A}} \pi_h(a \mid s)g(s,a) \right| \mathrm{d}s \tag{J.10}$$

$$\leq \int_{s \in \mathcal{S}} \|g'\|_{\mathcal{H}} \, B\upsilon \mathrm{d}s = B\upsilon \|g'\|_{\mathcal{H}},$$

where $\upsilon$ is the Lebesgue measure of $\mathcal{S}$. Therefore, $\chi[\psi_h^n, \pi, g]$ is bounded by $B\upsilon$. By Riesz theorem, there exists an element $u[\psi_h^n, \pi, g] \in \mathcal{H}$ with $\|u[\psi_h^n, \pi, g]\|_{\mathcal{H}} \leq B\upsilon$, such that $\chi[\psi_h^n, \pi, g]g' = \langle u[\psi_h^n, \pi, g], g' \rangle_{\mathcal{H}}$ for any $g' \in \mathcal{H}$. Therefore, by (J.8) and Cauchy-Schwarz inequality, we have

$$\left| \mathbb{E}_{\pi, \mathcal{P}^n}[g(s_h, a_h)] \right| \leq \mathbb{E}_{\pi, \mathcal{P}^n} \left[ \left| \left\langle u[\psi_h^n, \pi, g], k\big(\phi_{h-1}^n(s_{h-1}, a_{h-1})\big) \right\rangle_{\mathcal{H}} \right| / c_{\min} \right] \tag{J.11}$$

$$\leq \mathbb{E}_{\pi, \mathcal{P}^n} \left[ \left\|k\big(\phi_{h-1}^n(s_{h-1}, a_{h-1})\big)\right\|_{\Gamma_p^{-1}[\phi_{h-1}^n, \rho_{h-1}^n, \lambda, n]} \|u[\psi_h^n, \pi, g]\|_{\Gamma_p[\phi_{h-1}^n, \rho_{h-1}^n, \lambda, n]} / c_{\min} \right],$$

where $\Gamma_p[\phi_{h-1}^n, \rho_{h-1}^n, \lambda, n]$ is defined in (E.4), and $c_{\min}$ is the lower bound of the normalization constant $c[\phi_{h-1}^n, \psi_h^n](s_{h-1}, a_{h-1})$. What remains is to bound the term $\|u[\psi_h^n, \pi, g]\|_{\Gamma_p[\phi_{h-1}^n, \rho_{h-1}^n, \lambda, n]}$ from the above. By the definition of $\Gamma_p[\phi_{h-1}^n, \rho_{h-1}^n, \lambda, n]$, we have

$$\|u[\psi_h^n, \pi, g]\|^2_{\Gamma_p[\phi_{h-1}^n, \rho_{h-1}^n, \lambda, n]} = \lambda \|u[\psi_h^n, \pi, g]\|^2_{\mathcal{H}} + n\mathbb{E}_{(s,a)\sim\rho_{h-1}^n} \left[ u^2[\psi_h^n, \pi, g]\big(\phi_{h-1}^n(s,a)\big) \right]$$

$$\leq \lambda B^2 \upsilon^2 + n\mathbb{E}_{(s,a)\sim\rho_{h-1}^n} \left[ u^2[\psi_h^n, \pi, g]\big(\phi_{h-1}^n(s,a)\big) \right]. \tag{J.12}$$

Since $u[\psi_h^n, \pi, g]$ is the representation of $\chi[\psi_h^n, \pi, g]$ defined in (J.9), we have

$$u[\psi_h^n, \pi, g]\big(\phi_{h-1}^n(s,a)\big) = \left\langle u[\psi_h^n, \pi, g], k\big(\phi_{h-1}^n(s,a)\big) \right\rangle_{\mathcal{H}} \tag{J.13}$$

$$= \int_{s \in \mathcal{S}} \left\langle k\big(\phi_{h-1}^n(s,a)\big), k\big(\psi_h^n(s')\big) \right\rangle_{\mathcal{H}} \sum_{a' \in \mathcal{A}} \pi_h(a' \mid s')g(s',a')\mathrm{d}s'.$$

Combining (J.13) with the structure of the transition kernel $\mathcal{P}_{h-1}^n$ in Lemma J.2, we have

$$u[\psi_h^n, \pi, g]\big(\phi_{h-1}^n(s,a)\big) = c[\phi_{h-1}^n, \psi_h^n](s,a) \int_{s \in \mathcal{S}} \mathcal{P}_{h-1}^n(s' \mid s,a) \sum_{a' \in \mathcal{A}} \pi_h(a' \mid s')g(s',a')\mathrm{d}s'$$

$$\leq c_{\max}\mathbb{E}_{s'\sim\mathcal{P}_{h-1}^n(\cdot|s,a),\pi_h}\big[g(s',a')\big] \tag{J.14}$$

since $c[\phi_{h-1}^n, \psi_h^n](s,a)$ is bounded from the above by $c_{\max}$. By the property of the sum of squares, we have

$$\mathbb{E}_{s'\sim\mathcal{P}_{h-1}^n(\cdot|s,a),\pi_h}\big[g(s',a')\big]^2 \leq 2\Big\{ \mathbb{E}_{s'\sim\mathcal{P}_{h-1}^*(\cdot|s,a),\pi_h}\big[g(s',a')\big] - \mathbb{E}_{s'\sim\mathcal{P}_{h-1}^n(\cdot|s,a),\pi_h}\big[g(s',a')\big] \Big\}^2$$

$$+ 2\mathbb{E}_{s'\sim\mathcal{P}_{h-1}^*(\cdot|s,a),\pi_h}\big[g(s',a')\big]^2. \tag{J.15}$$

By simple calculation, we have

$$\left| \mathbb{E}_{s'\sim\mathcal{P}_{h-1}^*(\cdot|s,a),\pi_h}\big[g(s',a')\big] - \mathbb{E}_{s'\sim\mathcal{P}_{h-1}^n(\cdot|s,a),\pi_h}\big[g(s',a')\big] \right| \tag{J.16}$$

$$\leq \int_{s'} \left| \mathcal{P}_{h-1}^*(s' \mid s,a) - \mathcal{P}_{h-1}^n(s' \mid s,a) \right| \sum_{a' \in \mathcal{A}} \pi_h(a' \mid s')g(s',a')\mathrm{d}s'.$$

$$\leq B \cdot \mathrm{TV}\big(\mathcal{P}_{h-1}^n(\cdot \mid s,a), \mathcal{P}_{h-1}^*(\cdot \mid s,a)\big),$$

where the last inequality follows from the definition of the total variance divergence. Combining (J.14), (J.15) with (J.16), we have

$$u[\psi_h^n, \pi, g]\big(\phi_{h-1}^n(s,a)\big)^2 = c^2[\phi_{h-1}^n, \psi_h^n](s,a)\mathbb{E}_{s'\sim\mathcal{P}_{h-1}^n(\cdot|s,a),\pi_h}\big[g(s',a')\big]^2 \tag{J.17}$$

$$\leq 2c_{\max}^2 \Big\{ \mathbb{E}_{s'\sim\mathcal{P}_{h-1}^*(\cdot|s,a),\pi_h}\big[g(s',a')\big]^2 + B^2\mathrm{TV}^2\big(\mathcal{P}_{h-1}^n(\cdot \mid s,a), \mathcal{P}_{h-1}^*(\cdot \mid s,a)\big) \Big\}.$$

We plug (J.17) into (J.12) and have

$$\left\|u[\psi_h^n, \pi, g]\right\|_{\Gamma_p[\phi_{h-1}^n, \rho_{h-1}^n, \lambda, n]}^2 \le \lambda B^2 v^2 + 2n\mathbb{E}_{(s,a)\sim\rho_{h-1}^n}\left[c_{\max}^2 \mathbb{E}_{s'\sim\mathcal{P}_{h-1}^*(\cdot|s,a),\pi_h}\left[g(s', a')\right]^2\right]$$
$$+ 2nB^2\mathbb{E}_{(s,a)\sim\rho_{h-1}^n}\left[c_{\max}^2 \mathrm{TV}^2\big(\mathcal{P}_{h-1}^n(\cdot \mid s, a), \mathcal{P}_{h-1}^*(\cdot \mid s, a)\big)\right]. \tag{J.18}$$

Therefore, when $n\mathbb{E}_{(s,a)\sim\rho_{h-1}^n}[\mathrm{TV}^2(\mathcal{P}_{h-1}^n(\cdot \mid s, a), \mathcal{P}_{h-1}^*(\cdot \mid s, a))] \le \zeta^2$ holds, we have

$$\left\|u[\psi_h^n, \pi, g]\right\|_{\Gamma_p[\phi_{h-1}^n, \rho_{h-1}^n, \lambda, n]}^2 \le \lambda B^2 v^2 + 2c_{\max}^2\left\{n\mathbb{E}_{\rho_{h-1}^n, \mathcal{P}_{h-1}^*, \pi_h}\left[g^2(s', a')\right] + B^2\zeta^2\right\} \tag{J.19}$$

by (J.18). Here the expectation is defined in Lemma J.2. By the definition of $\mathcal{U}(\mathcal{A})$, we have

$$\mathbb{E}_{\rho_{h-1}^n, \mathcal{P}_{h-1}^*, \pi_h}\left[g^2(s', a')\right] \le |\mathcal{A}|\,\mathbb{E}_{\rho_{h-1}^n, \mathcal{P}_{h-1}^*, \mathcal{U}(\mathcal{A})}\left[g^2(s', a')\right]. \tag{J.20}$$

Combining (J.11), (J.19) with (J.20), we have

$$\left|\mathbb{E}_{\pi, \mathcal{P}_{h-1}^n}[g(s_h, a_h)]\right| \le \mathbb{E}_{\pi, \mathcal{P}_{h-1}^n}\left[\beta_l \left\|k\big(\phi_{h-1}^n(s_{h-1}, a_{h-1})\big)\right\|_{\Gamma_p^{-1}[\phi_{h-1}^n, \rho_{h-1}^n, \lambda, n]}\right].$$

Here the expectation and $\beta_l$ are defined in Lemma J.2. Thus, we conclude the proof of Lemma J.2. $\qquad\square$

## J.3 PROOF OF LEMMA J.3

In this subsection, we provide the proof of the one-step backward inequality for the true model. We first state this lemma below.

**Lemma J.3** (One-Step Backward for the True Model). *Let $\rho_{h-1}^n$ be a distribution over $\mathcal{S} \times \mathcal{A}$. We assume that for the transition kernel $\mathcal{P}^n = \{\mathcal{P}_h^n\}_{h=1}^H$ and the transition kernel $\mathcal{P}^* = \{\mathcal{P}_h^*\}_{h=1}^H$ with*

$$\mathcal{P}_h^n(s' \mid s, a) = \left\langle k\big(\phi_h^n(s, a)\big), k\big(\psi_{h+1}^n(s')\big)\right\rangle_{\mathcal{H}}\Big/ c[\phi_h^n, \psi_{h+1}^n](s, a),$$

*and* $\qquad \mathcal{P}_h^*(s' \mid s, a) = \left\langle k\big(\phi_h^*(s, a)\big), k\big(\psi_{h+1}^*(s')\big)\right\rangle_{\mathcal{H}}\Big/ c[\phi_h^*, \psi_{h+1}^*](s, a).$

*We also assume that the non-negative function $g$ satisfies $\|g\|_\infty \le B$. Then for any policy $\pi$, we have*

$$\left|\mathbb{E}_{\pi, \mathcal{P}_h^*}\big[g(s_h, a_h)\big]\right| \le \mathbb{E}_{\pi, \mathcal{P}_h^*}\left[\beta_t \left\|k\big(\phi_{h-1}^*(s_{h-1}, a_{h-1})\big)\right\|_{\Gamma_p^{-1}[\phi_{h-1}^*, \rho_{h-1}^n, \lambda, n]}\right].$$

*Here the expectation $\mathbb{E}_{\rho_{h-1}^n, \mathcal{P}_h^*, \mathcal{U}(\mathcal{A})}[g(s_h, a_h)]$ is defined as*

$$\mathbb{E}_{\rho_{h-1}^n, \mathcal{P}_h^*, \mathcal{U}(\mathcal{A})}\big[g(s_h, a_h)\big] = \mathbb{E}_{(s_{h-1}, a_{h-1})\sim\rho_{h-1}^n, s_h\sim\mathcal{P}_h^*(\cdot|s_{h-1}, a_{h-1}), a_h\sim\mathcal{U}(\mathcal{A})}\big[g(s_h, a_h)\big],$$

*the operator $\Gamma_p[\phi_{h-1}^*, \rho_{h-1}^n, \lambda, n]$ is the operator defined in (E.4), and $\beta_t$ is defined as*

$$\beta_t = \sqrt{\lambda B^2 v^2/c_{\min}^2 + r_{\max}^2 n |\mathcal{A}|\,\mathbb{E}_{\rho_{h-1}^n, \mathcal{P}_h^*, \mathcal{U}(\mathcal{A})}\big[g^2(s_h, a_h)\big]}.$$

*Proof.* The proof is similar with the proof of Lemma J.2. Similar with (J.8), we have

$$\left|\mathbb{E}_{\pi, \mathcal{P}^*}\big[g(s_h, a_h)\big]\right| \le \mathbb{E}_{\pi, \mathcal{P}_h^*}\left[\left|\int_{s\in\mathcal{S}} K\big(\phi_{h-1}^*(s_{h-1}, a_{h-1}), \psi_h^*(s)\big)\bar{g}(s, \pi_h)\mathrm{d}s\right|\Big/c_{\min}\right], \tag{J.21}$$

where $\bar{g}(s, \pi_h)$ is defined in (J.7). By (J.10), the the linear functional $\chi[\psi_h^*, \pi, g]$ defined in (J.9) is bounded by $Bv$. Here $v$ is the Lebesgue measure of $\mathcal{S}$. By Riesz's theorem, there exists an element $u[\psi_h^*, \pi, g] \in \mathcal{H}$ with $\|u[\psi_h^*, \pi, g]\|_{\mathcal{H}} \le Bv$, such that $\chi[\psi_h^*, \pi, g]g' = \langle u[\psi_h^*, \pi, g], g'\rangle_{\mathcal{H}}$ holds for all $g' \in \mathcal{H}$. Combining this property with (J.21), we have

$$\left|\mathbb{E}_{\pi, \mathcal{P}_h^*}\big[g(s_h, a_h)\big]\right| \le \mathbb{E}_{\pi, \mathcal{P}_h^*}\left[\left|\left\langle u[\psi_h^*, \pi, g], k\big(\phi_{h-1}^*(s_{h-1}, a_{h-1})\big)\right\rangle_{\mathcal{H}}\right|\Big/c_{\min}\right] \tag{J.22}$$
$$\le \left|\mathbb{E}_{\pi, \mathcal{P}_h^*}\left[\left\|k\big(\phi_{h-1}^*(s_{h-1}, a_{h-1})\big)\right\|_{\Gamma_p^{-1}[\phi_{h-1}^*, \rho_{h-1}^n, \lambda, n]}\left\|u[\psi_h^*, \pi, g]\right\|_{\Gamma_p[\phi_{h-1}^*, \rho_{h-1}^n, \lambda, n]}\right]\right|\Big/c_{\min}.$$

Here $\Gamma_p[\phi_{h-1}^*, \rho_{h-1}^*, \lambda, n]$ is defined in (E.4), and $c_{\min}$ is the lower bound of the normalization constant $c[\phi_{h-1}^n, \psi_h^n](s_{h-1}, a_{h-1})$. By the definition of the operator , we have

$$\big\| u[\psi_h^*, \pi, g] \big\|_{\Gamma_p[\phi_{h-1}^*, \rho_{h-1}^*, \lambda, n]}^2 = \lambda \big\| u[\psi_h^*, \pi, g] \big\|_{\mathcal{H}}^2 + n\mathbb{E}_{(s,a)\sim\rho_{h-1}^*}\Big[ u^2[\psi_h^*, \pi, g]\big(\phi_{h-1}^*(s,a)\big) \Big]$$
$$\leq \lambda B^2 v^2 + n\mathbb{E}_{(s,a)\sim\rho_{h-1}^*}\Big[ u^2[\psi_h^*, \pi, g]\big(\phi_{h-1}^*(s,a)\big) \Big]. \qquad (J.23)$$

Similar with (J.14), we have

$$u[\psi_h^*, \pi, g]\big(\phi_{h-1}^*(s,a)\big) = c[\phi_{h-1}^*, \psi_h^*](s,a)\mathbb{E}_{s'\sim\mathcal{P}_{h-1}^*(\cdot|s,a),\pi_h}\big[ g(s', a') \big]. \qquad (J.24)$$

We plug (J.24) into (J.23) and have

$$\big\| u[\psi_h^*, \pi, g] \big\|_{\Gamma_p[\phi_{h-1}^*, \rho_{h-1}^n, \lambda, n]}^2 \leq \lambda B^2 v^2 + nc_{\max}^2 \mathbb{E}_{(s,a)\sim\rho_{h-1}^n}\Big[ \mathbb{E}_{s'\sim\mathcal{P}_h^*(\cdot|s,a),\pi_h}\big[ g(s', a') \big]^2 \Big]$$
$$\leq \lambda B^2 v^2 + nc_{\max}^2 \mathbb{E}_{\rho_{h-1}^n, \mathcal{P}_h^*(\cdot|s,a), \pi_h}\big[ g^2(s', a') \big], \qquad (J.25)$$

where the last inequality follows from Cauchy-Schwarz inequality. Since $g^2$ is non-negative, we have

$$\mathbb{E}_{\rho_{h-1}^n, \mathcal{P}_h^*(\cdot|s,a), \pi_h}\big[ g^2(s_h, a_h) \big] \leq |\mathcal{A}| \, \mathbb{E}_{\rho_{h-1}^n, \mathcal{P}_h^*(\cdot|s,a), \mathcal{U}(\mathcal{A})}\big[ g^2(s_h, a_h) \big]. \qquad (J.26)$$

Here $\mathbb{E}_{\rho_{h-1}^n, \mathcal{P}_h^*, \pi_h}[g(s_h, a_h)] = \mathbb{E}_{(s_{h-1}, a_{h-1})\sim\rho_{h-1}^n, s_h\sim\mathcal{P}_h^*(\cdot|s_{h-1}, a_{h-1}), a_h\sim\pi_h(\cdot|s_h)}[g(s_h, a_h)]$. Combining (J.22), (J.25) and (J.26), we have

$$\Big| \mathbb{E}_{\pi, \mathcal{P}_h^*}\big[ g(s_h, a_h) \big] \Big| \leq \mathbb{E}_{\pi, \mathcal{P}_h^*}\Big[ \beta_t \big\| k(\phi_{h-1}^*(s_{h-1}, a_{h-1})) \big\|_{\Gamma_p^{-1}[\phi_{h-1}^*, \rho_{h-1}^n, \lambda, n]} \Big],$$

where $\beta_t$ is defined in Lemma J.3. Thus, we conclude the proof of Lemma J.3.

$\square$

### J.4    PROOF OF LEMMA C.7

*Proof.* The lower bound of the $\epsilon$-Eluder dimension has been proven by Theorem 5.2 of Dong et al. (2021). Therefore, we only need to bound the $\epsilon$-log covering number of $\mathcal{R}$ from the above. Let $\mathcal{C}_1$ be an $\epsilon/2$-covering set of the unit sphere in $\mathbb{R}^m$, and $\mathcal{C}_2$ be an $\epsilon/2$-covering set of $[0, 1]$. We have $|\mathcal{C}_1| \leq 2^{2d}\epsilon^{-d}$ and $|\mathcal{C}_1| \leq 2/\epsilon$. We define the set $\mathcal{C}$ as

$$\mathcal{C} = \{r_{\theta,b} : \mathcal{A} \to \mathbb{R} \mid r_{\theta,b}(a) = \sigma(\theta^\top a + b), \ \theta \in \mathcal{C}_1, \ b \in \mathcal{C}_2\}.$$

For any $r_{\theta,b} \in \mathcal{R}$, there exists $\theta_0 \in \mathcal{C}_1$ and $b_0 \in \mathcal{C}_2$, such that $\|\theta - \theta_0\|_2 \leq \epsilon$ and $|b - b_0| \leq \epsilon/2$. We have $r_{\theta_0, b_0} \in \mathcal{C}$, and

$$|r_{\theta,b}(a) - r_{\theta_0, b_0}(a)| = \big| \max\{\theta^\top a + b, 0\} - \max\{\theta_0^\top a + b_0, 0\} + \big| \qquad (J.27)$$
$$\leq \|\theta - \theta_0\|_2 \|a\|_2 + |b - b_0| \leq \epsilon$$

for any $a \in \mathcal{A}$. Therefore, $\mathcal{C}$ is an $\epsilon$-covering set of $\mathcal{R}$, and

$$\log |\mathcal{C}| \leq \log |\mathcal{C}_1| + \log |\mathcal{C}_2| \leq (d + 1) \log(4/\epsilon),$$

which concludes the proof of Lemma C.7.

$\square$

### J.5    PROOF OF LEMMA D.2

*Proof.* By the reproducing property of $\mathcal{H}$ and Cauchy-Schwarz inequality, we have

$$\int_{\mathcal{X}} g^2(x) \mathrm{d}\rho(x) = \int_{\mathcal{X}} \langle g, k(x) \rangle_{\mathcal{H}}^2 \mathrm{d}\rho(x) \leq \int_{\mathcal{X}} \langle g, g \rangle_{\mathcal{H}} K(x, x) \mathrm{d}\rho(x) = \langle g, g \rangle_{\mathcal{H}} \qquad (J.28)$$

when $g \in \mathcal{H}$. Therefore, we have $\mathcal{H} \subset L_\rho^2(\mathcal{X})$ and $\|g\|_{L_\rho^2(\mathcal{X})}^2 \leq \|g\|_{\mathcal{H}}^2$. We also have

$$\int_{\mathcal{X}} \big[ \Gamma g(x) \big]^2 \mathrm{d}\rho(x) = \int_{\mathcal{X}} \left[ \int_{\mathcal{X}} K(x', x) g(x') \mathrm{d}\rho(x') \right]^2 \mathrm{d}\rho(x) \qquad (J.29)$$
$$\leq \int_{\mathcal{X}} \left[ \int_{\mathcal{X}} K^2(x', x) \mathrm{d}\rho(x') \right] \left[ \int_{\mathcal{X}} g^2(x'') \mathrm{d}\rho(x'') \right] \mathrm{d}\rho(x)$$
$$= \left[ \int_{\mathcal{X}} g^2(x'') \mathrm{d}\rho(x'') \right] \int_{\mathcal{X}} \int_{\mathcal{X}} K^2(x', x) \mathrm{d}\rho(x') \mathrm{d}\rho(x) = \int_{\mathcal{X}} g^2(x'') \mathrm{d}\rho(x'').$$

Where the last equation holds since $K(x, x') = \exp(-\|x - x'\|_2^2/2)$. Therefore, we have $\Gamma g \in L_\rho^2(\mathcal{X})$ when $g \in L_\rho^2(\mathcal{X})$, which implies that $\Gamma$ is a linear operator on $L_\rho^2(\mathcal{X})$. When we define the functional $\gamma_g : \mathcal{H} \to \mathbb{R}$ as $\gamma_g(\bar{g}) = \int_\mathcal{X} g(x)\bar{g}(x)\mathrm{d}\rho(x)$ for $g \in L_\rho^2(\mathcal{X})$, we have

$$\gamma_g(\bar{g}) \leq \sqrt{\int_\mathcal{X} g^2(x)\mathrm{d}\rho(x)}\sqrt{\int_\mathcal{X} \bar{g}^2(x)\mathrm{d}\rho(x)} \leq \sqrt{\langle \bar{g}, \bar{g}\rangle_\mathcal{H}}\sqrt{\int_\mathcal{X} g^2(x)\mathrm{d}\rho(x)}$$

by (J.28) and (J.29). Therefore, by Riesz's theorem, we know that there exists $u \in \mathcal{H}$, such that $\langle u, \bar{g}\rangle_\mathcal{H} = \gamma_g(\bar{g})$, which implies. $\Gamma g(x) \in \mathcal{H}$. The operator $\Gamma : \mathcal{H} \to \mathcal{H}$ is also non-negative definite since we have $\langle g, \Gamma g\rangle_\mathcal{H} = \int_\mathcal{X} g^2(x)\mathrm{d}\rho(x) \geq 0$ when $g \in \mathcal{H}$. Thus, we conclude the proof of Lemma D.2. $\square$

### J.6  PROOF OF LEMMA D.3

*Proof.* We first define the operator $\bar{\Gamma}[\mathcal{X}_n]$ as

$$\bar{\Gamma}[\mathcal{X}_n]g(x) = \Big(g(x) - g[\mathcal{X}_n]^\top\big(\lambda I + K[\mathcal{X}_n]\big)^{-1}k[\mathcal{X}_n](x)\Big)\Big/\lambda, \tag{J.30}$$

for any $g \in \mathcal{H}$, $x \in \mathcal{X}$. Here $g[\mathcal{X}_n]$ is defined in §1.2. We show that $\bar{\Gamma}[\mathcal{X}_n]$ is the inverse of $\Gamma[\mathcal{X}_n]$. By the definition of $\Gamma[\mathcal{X}_n]$ in Lemma D.3, we have $(\Gamma[\mathcal{X}_n]g)[\mathcal{X}_n] = (\lambda I + K[\mathcal{X}_n])g[\mathcal{X}_n]$, and

$$\Gamma[\mathcal{X}_n]g(x) = \lambda g(x) + g[\mathcal{X}_n]^\top k[\mathcal{X}_n](x), \tag{J.31}$$

where $k[\mathcal{X}_n](x)$ is defined in Lemma D.3. Combining Equaitions (J.30) with (J.31), we have

$$\bar{\Gamma}[\mathcal{X}_n]\Gamma[\mathcal{X}_n]g(x) = \Big(\Gamma[\mathcal{X}_n]g(x) - (\Gamma[\mathcal{X}_n]g)[\mathcal{X}_n]^\top\big(\lambda I + K[\mathcal{X}_n]\big)^{-1}k[\mathcal{X}_n](x)\Big)\Big/\lambda$$

$$= \big(\lambda g(x) + g[\mathcal{X}_n]^\top k[\mathcal{X}_n](x) - g[\mathcal{X}_n]^\top k[\mathcal{X}_n](x)\big)\big/\lambda = g(x).$$

By the definition in (J.30) and the definition of $(\bar{\Gamma}[\mathcal{X}_n]g)[\mathcal{X}_n]$ in §1.2, we have

$$\big(\bar{\Gamma}[\mathcal{X}_n]g\big)[\mathcal{X}_n] = \Big(g[\mathcal{X}_n] - K[\mathcal{X}_n]\big(\lambda I + K[\mathcal{X}_n]\big)^{-1}g[\mathcal{X}_n]\Big)\Big/\lambda = \big(\lambda I + K[\mathcal{X}_n]\big)^{-1}g[\mathcal{X}_n]. \tag{J.32}$$

Combining (J.32) with (J.30), we have

$$\Gamma[\mathcal{X}_n]\bar{\Gamma}[\mathcal{X}_n]g(x) = \lambda\bar{\Gamma}[\mathcal{X}_n]g(x) + \big(\bar{\Gamma}[\mathcal{X}_n]g\big)[\mathcal{X}_n]^\top k[\mathcal{X}_n](x)$$

$$= g(x) - g[\mathcal{X}_n]^\top\big(\lambda I + K[\mathcal{X}_n]\big)^{-1}k[\mathcal{X}_n](x) + g[\mathcal{X}_n]^\top\big(\lambda I + K[\mathcal{X}_n]\big)^{-1}k[\mathcal{X}_n](x) = g(x).$$

Therefore, the operator $\bar{\Gamma}[\mathcal{X}_n]$ is the inverse operator of $\Gamma[\mathcal{X}_n]$, and

$$\big\|k(x)\big\|_{\Gamma^{-1}[\mathcal{X}_n]}^2 = \langle k(x), \Gamma^{-1}[\mathcal{X}_n]k(x)\rangle_\mathcal{H} = \Big(K(x, x) - k[\mathcal{X}_n](x)^\top\big(\lambda I + K[\mathcal{X}_n]\big)^{-1}k[\mathcal{X}_n](x)\Big)\Big/\lambda.$$

Thus, we conclude the proof of Lemma D.3. $\square$

### J.7  PROOF OF LEMMA D.4

*Proof.* By the definition of the operator $\Gamma_e[\phi, \mathcal{D}, \lambda]$, we have

$$\langle g, \Gamma_e[\phi, \mathcal{D}, \lambda]g\rangle_\mathcal{H} = \lambda\langle g, g\rangle_\mathcal{H} + \sum_{(s,a,r,s')\in\mathcal{D}} g\big(\phi(s, a)\big)^2. \tag{J.33}$$

Therefore, we have $\lambda\langle g, g\rangle_\mathcal{H} \leq \langle g, \Gamma_e[\phi, \mathcal{D}, \lambda]g\rangle_\mathcal{H}$. By the definition of the reproducing kernel, we have $\sum_{(s,a,r,s')\in\mathcal{D}} g(\phi(s, a))^2 = \sum_{(s,a,r,s')\in\mathcal{D}} \langle g, k(\phi(s, a))\rangle_\mathcal{H}^2$. Therefore, we have

$$\sum_{(s,a,r,s')\in\mathcal{D}} g\big(\phi(s, a)\big)^2 \leq \langle g, g\rangle_\mathcal{H} \sum_{(s,a,r,s')\in\mathcal{D}} K\big(\phi(s, a), \phi(s, a)\big) \tag{J.34}$$

by Cauchy-Schwarz inequality. Combining (J.33) with (J.34), we have

$$\langle g, \Gamma_e[\phi, \mathcal{D}, \lambda]g\rangle_\mathcal{H} \leq \Big(\lambda + \sum_{(s,a,r,s')\in\mathcal{D}} K\big(\phi(s, a), \phi(s, a)\big)\Big)\langle g, g\rangle_\mathcal{H}.$$

Similarly, by the definition of the operator $\Gamma_p[\phi, \rho, \lambda, \tau]$, we have

$$\langle g, \Gamma_p[\phi, \rho, \lambda, \tau]g\rangle_\mathcal{H} = \lambda\langle g, g\rangle_\mathcal{H} + \tau\mathbb{E}_{(s,a)\sim\rho}\Big[g^2\big(\phi(s, a)\big)\Big]. \tag{J.35}$$

Therefore, we have $\lambda\langle g, g\rangle_\mathcal{H} \leq \langle g, \Gamma_p[\phi, \rho, \lambda, \tau]g\rangle_\mathcal{H}$. Similar with (J.34), we have

$$\mathbb{E}_{(s,a)\sim\rho}\Big[g^2\big(\phi(s, a)\big)\Big] \leq \langle g, g\rangle_\mathcal{H}\mathbb{E}_{(s,a)\sim\rho}\Big[K\big(\phi(s, a), \phi(s, a)\big)\Big]. \tag{J.36}$$

We conclude the proof of Lemma D.4 by combining (J.35) with (J.36). $\square$

## J.8 Proof of Lemma F.2

*Proof.* Since the operators $A$, $B$ are self-adjoint and positive-definite, the operator $B^{-1/2}AB^{-1/2}$ is also self-adjoint and positive-definite. Therefore, the operator $C = (B^{-1/2}AB^{-1/2})^{1/2}$ is also self-adjoint and positive-definite. Since $B^{-1/2}C^{-1}B^{-1/2}x \in \mathcal{D}$, we have

$$\langle B^{-1/2}C^{-1}B^{-1/2}x, BB^{-1/2}C^{-1}B^{-1/2}x\rangle_{\mathcal{H}} \geq \langle B^{-1/2}C^{-1}B^{-1/2}x, AB^{-1/2}C^{-1}B^{-1/2}x\rangle_{\mathcal{H}}$$

by the definition of $\mathcal{D}$ in Lemma F.2. Since the operators $B$ and $C$ are self-adjoint, we have

$$\langle B^{-1/2}C^{-1}B^{-1/2}x, BB^{-1/2}C^{-1}B^{-1/2}x\rangle_{\mathcal{H}} \tag{J.37}$$
$$= \langle x, B^{-1/2}C^{-1}B^{-1/2}BB^{-1/2}C^{-1}B^{-1/2}x\rangle_{\mathcal{H}} = \langle x, B^{-1/2}C^{-2}B^{-1/2}x\rangle_{\mathcal{H}}.$$

We also have $C^{-2} = B^{1/2}A^{-1}B^{1/2}$ when $C = (B^{-1/2}AB^{-1/2})^{1/2}$. Therefore, we have

$$\langle B^{-1/2}C^{-1}B^{-1/2}x, BB^{-1/2}C^{-1}B^{-1/2}x\rangle_{\mathcal{H}} = \langle x, B^{-1/2}B^{1/2}A^{-1}B^{1/2}B^{-1/2}x\rangle_{\mathcal{H}} \tag{J.38}$$
$$= \langle x, A^{-1}x\rangle_{\mathcal{H}}$$

by (J.37). Similarly, we have

$$\langle B^{-1/2}C^{-1}B^{-1/2}x, AB^{-1/2}C^{-1}B^{-1/2}x\rangle_{\mathcal{H}} = \langle x, B^{-1/2}C^{-1}B^{-1/2}AB^{-1/2}C^{-1}B^{-1/2}x\rangle_{\mathcal{H}}$$
$$= \langle x, B^{-1/2}C^{-1}C^2C^{-1}B^{-1/2}x\rangle_{\mathcal{H}} = \langle x, B^{-1}x\rangle_{\mathcal{H}}. \tag{J.39}$$

We conclude the proof of Lemma F.2 by combining (J.38) with (J.39). □

## J.9 Proof of Lemma F.3

*Proof.* Without loss of generality, we assume that $\langle g, g\rangle_{\mathcal{H}} = 1$. We then have

$$g(x) = \langle g, k(x)\rangle_{\mathcal{H}} \leq \sqrt{\langle g, g\rangle_{\mathcal{H}}\langle k(x), k(x)\rangle_{\mathcal{H}}} = 1$$

for any $x \in \mathcal{X}$. We also introduce the following version of Bernstein inequality.

**Lemma J.4.** *Let $\{\mathcal{F}_\tau\}_{\tau=0}^n$ be a filtration, and $\{X_\tau\}_{\tau=1}^n$ be a $\mathbb{R}$-valued stochastic process adapted to this filtration. Suppose $\mathbb{E}[X_\tau \mid \mathcal{F}_{\tau-1}] = 0$ and $|X_\tau| \leq c$ almost surely, we have*

$$P\Big(\sum_{\tau=1}^n X_\tau \leq (e-2)\lambda\mathbb{E}[X_\tau^2 \mid \mathcal{F}_{\tau-1}] + \log(1/\delta)/\lambda\Big) \geq 1 - \delta$$

*for any fix $0 < \lambda \leq 1/c$.*

*Proof.* See Appendix §J.12 for a detailed proof. □

Since $\mathbb{E}_{(s,a)\sim\varrho^\tau}[g^2(\phi(s,a))] = \mathbb{E}[g^2(\phi(s^\tau, a^\tau)) \mid \mathcal{F}_{\tau-1}]$, using Lemma J.4 with $\lambda = 1/2$, we have

$$\sum_{\tau=1}^n g^2\big(\phi(s^\tau, a^\tau)\big) - \sum_{\tau=1}^n \mathbb{E}_{(s,a)\sim\varrho^\tau}\Big[g^2\big(\phi(s,a)\big)\Big] \tag{J.40}$$
$$\leq (e-2)\sum_{\tau=1}^n \mathrm{Var}\Big(g^2\big(\phi(s^\tau, a^\tau)\big) \mid \mathcal{F}_{\tau-1}\Big)/2 + 2\log(1/\delta)$$

holds with probability at least $1 - \delta$. By the property of the variance, we have

$$\mathrm{Var}\Big(g^2\big(\phi(s^\tau, a^\tau)\big) \mid \mathcal{F}_{\tau-1}\Big) \leq \mathbb{E}\Big[g^4\big(\phi(s^\tau, a^\tau)\big) \mid \mathcal{F}_{\tau-1}\Big] \leq \mathbb{E}\Big[g^2\big(\phi(s^\tau, a^\tau)\big) \mid \mathcal{F}_{\tau-1}\Big]. \tag{J.41}$$

Combining (J.40) with (J.41), we have

$$\sum_{\tau=1}^n g^2\big(\phi(s^\tau, a^\tau)\big) \leq e\sum_{\tau=1}^n \mathbb{E}\Big[g^2\big(\phi(s^\tau, a^\tau)\big) \mid \mathcal{F}_{\tau-1}\Big]\Big/2 + 2\log(1/\delta)\langle g, g\rangle_{\mathcal{H}}$$

holds with probability at least $1 - \delta$. Similarly, we have

$$\sum_{\tau=1}^{n} \mathbb{E}_{(s,a)\sim\varrho^\tau}\Big[g^2\big(\phi(s,a)\big)\Big] - \sum_{\tau=1}^{n} g^2\big(\phi(s^\tau, a^\tau)\big)$$

$$\leq (e-2)\sum_{\tau=1}^{n} \mathbb{E}\Big[g^2\big(\phi(s^\tau, a^\tau)\big) \mid \mathcal{F}_{\tau-1}\Big]\Big/2 + 2\log(1/\delta),$$

holds with probability at least $1 - \delta$. By simple calculation, we have

$$(4-e)\sum_{\tau=1}^{n} \mathbb{E}\Big[g^2\big(\phi(s^\tau, a^\tau)\big) \mid \mathcal{F}_{\tau-1}\Big]\Big/2 \leq \sum_{\tau=1}^{n} g^2\big(\phi(s^\tau, a^\tau)\big) + 2\log(1/\delta)\langle g, g\rangle_\mathcal{H}$$

holds with probability at least $1 - \delta$. Thus, we conclude the proof of Lemma F.3. $\qquad\square$

### J.10    PROOF OF LEMMA F.4

*Proof.* By the definition of $\Gamma$ in (F.1), we have $\Gamma g = \Gamma_1^{-1/2}(\Gamma_1^{-1/2}\Gamma_2\Gamma_1^{-1/2})^{-1/2}\Gamma_1^{-1/2}g$. Since the operator $\Gamma_1^{-1/2}$ is self-adjoint, we have

$$\langle\Gamma g, \Gamma g\rangle_\mathcal{H} = \Big\langle (\Gamma_1^{-1/2}\Gamma_2\Gamma_1^{-1/2})^{-1/2}\Gamma_1^{-1/2}g, \Gamma_1^{-1}(\Gamma_1^{-1/2}\Gamma_2\Gamma_1^{-1/2})^{-1/2}\Gamma_1^{-1/2}g \Big\rangle_\mathcal{H}. \qquad (\text{J.42})$$

By Lemma D.4, we have $\langle g_1, \Gamma_1^{-1}g_1\rangle_\mathcal{H} \leq \langle g_1, g_1\rangle_\mathcal{H}/\lambda$ for any $g_1 \in \mathcal{H}$. Therefore, we have

$$\langle\Gamma g, \Gamma g\rangle_\mathcal{H} \leq \Big\langle (\Gamma_1^{-1/2}\Gamma_2\Gamma_1^{-1/2})^{-1/2}\Gamma_1^{-1/2}g, (\Gamma_1^{-1/2}\Gamma_2\Gamma_1^{-1/2})^{-1/2}\Gamma_1^{-1/2}g \Big\rangle_\mathcal{H}/\lambda \qquad (\text{J.43})$$

$$= \Big\langle \Gamma_1^{-1/2}g, (\Gamma_1^{-1/2}\Gamma_2\Gamma_1^{-1/2})^{-1}\Gamma_1^{-1/2}g \Big\rangle_\mathcal{H}/\lambda.$$

The following lemma characterize the operator $\Gamma_1^{-1/2}\Gamma_2\Gamma_1^{-1/2}$.

**Lemma J.5.** *When we define $\Gamma_1 = \Gamma_p[\phi, \rho_{h,i}^n, \lambda, n]$ and $\Gamma_2 = \Gamma_e[\phi, \mathcal{D}_{h,i}^n, \lambda]$, we have*

$$\lambda\langle g, g\rangle_\mathcal{H}/(\lambda+n) \leq \big\langle g, \Gamma_1^{-1/2}\Gamma_2\Gamma_1^{-1/2}g\big\rangle_\mathcal{H} \leq (\lambda+n)\langle g, g\rangle_\mathcal{H}/\lambda$$

*for any $g \in \mathcal{H}$. Similarly we have*

$$\lambda\langle g, g\rangle_\mathcal{H}/(\lambda+n) \leq \Big\langle g, (\Gamma_1^{-1/2}\Gamma_2\Gamma_1^{-1/2})^{-1}g \Big\rangle_\mathcal{H} \leq (\lambda+n)\langle g, g\rangle_\mathcal{H}/\lambda.$$

*Here $\Gamma_p[\phi, \rho_{h,i}^n, \lambda, n]$ and $\Gamma_e[\phi, \mathcal{D}_{h,i}^n, \lambda]$ are defined in (E.2) and (E.4).*

*Proof.* See Appendix §J.13 for a detailed proof. $\qquad\square$

Combining Lemma J.5 with (J.43), we have

$$\langle\Gamma g, \Gamma g\rangle_\mathcal{H} \leq (\lambda+n)\langle\Gamma_1^{-1/2}g, \Gamma_1^{-1/2}g\rangle_\mathcal{H}/\lambda^2 = (\lambda+n)\langle g, \Gamma_1^{-1}g\rangle_\mathcal{H}/\lambda^2.$$

By Lemma D.4, we have $\langle g_1, \Gamma_1^{-1}g_1\rangle_\mathcal{H} \leq \langle g_1, g_1\rangle_\mathcal{H}/\lambda$ for any $g_1 \in \mathcal{H}$. Therefore, we have

$$\langle\Gamma g, \Gamma g\rangle_\mathcal{H} \leq (\lambda+n)\langle g, g\rangle_\mathcal{H}/\lambda^3.$$

Similarly, by Lemma D.4, we have $\langle g_1, \Gamma_1^{-1}g_1\rangle_\mathcal{H} \geq \langle g_1, g_1\rangle_\mathcal{H}/(\lambda+n)$ for any $g_1 \in \mathcal{H}$. Therefore, by (J.42), we have

$$\langle\Gamma g, \Gamma g\rangle_\mathcal{H} \geq \Big\langle (\Gamma_1^{-1/2}\Gamma_2\Gamma_1^{-1/2})^{-1/2}\Gamma_1^{-1/2}g, (\Gamma_1^{-1/2}\Gamma_2\Gamma_1^{-1/2})^{-1/2}\Gamma_1^{-1/2}g \Big\rangle_\mathcal{H}/(\lambda+n) \qquad (\text{J.44})$$

$$= \Big\langle \Gamma_1^{-1/2}g, (\Gamma_1^{-1/2}\Gamma_2\Gamma_1^{-1/2})^{-1}\Gamma_1^{-1/2}g \Big\rangle_\mathcal{H}/(\lambda+n).$$

Combining Lemma J.5 with (J.44), we have

$$\langle\Gamma g, \Gamma g\rangle_\mathcal{H} \geq \lambda\langle\Gamma_1^{-1/2}g, \Gamma_1^{-1/2}g\rangle_\mathcal{H}/(\lambda+n)^2 = \lambda\langle g, \Gamma_1^{-1}g\rangle_\mathcal{H}/(\lambda+n)^2.$$

By Proposition D.4, we have $\langle g_1, \Gamma_1^{-1}g_1\rangle_\mathcal{H} \geq \langle g_1, g_1\rangle_\mathcal{H}/(\lambda+n)$ for any $g_1 \in \mathcal{H}$. Therefore, we have

$$\langle\Gamma g, \Gamma g\rangle_\mathcal{H} \geq \lambda\langle g, \Gamma_1^{-1}g\rangle_\mathcal{H}/(\lambda+n)^3.$$

Thus, we conclude the proof of Lemma F.4. $\qquad\square$

### J.11   PROOF OF LEMMA F.5

*Proof.* We denote by $\mathcal{C}$ the $\bar{\epsilon}$-covering of the unit ball in $m$ dimension, and define $\mathcal{C}'$ by $\mathcal{C}' = \{\Gamma K(x, \cdot) \mid x \in \mathcal{C}\}$. In the following part of the proof, we show that $\mathcal{C}'$ is a $\sqrt{(\lambda + n)\bar{\epsilon}^2/\lambda^3}$-covering of $\mathcal{H}_1$ with regard to the infinity norm.

For any $g \in \mathcal{H}_1$, we have $g(\cdot) = \Gamma K(x, \cdot)$ for some $x$ in the unit ball. By the definition of the set $\mathcal{C}$, there exist $x' \in \mathcal{C}$, such that $\|x - x'\| \leq \bar{\epsilon}$. By the reproducing property of the kernel, we have

$$\left|\Gamma K(x, y) - \Gamma K(x', y)\right| = \left|\left\langle \Gamma\big(k(x) - k(x')\big), k(y)\right\rangle_{\mathcal{H}}\right| \tag{J.45}$$

for any $y \in \mathbb{R}^m$. Combining (J.45) with Cauchy-Schwarz inequality, we have

$$\left|\Gamma K(x, y) - \Gamma K(x', y)\right| \leq \sqrt{\left\langle \Gamma\big(k(x) - k(x')\big), \Gamma\big(k(x) - k(x')\big)\right\rangle_{\mathcal{H}} \left\langle k(y), k(y)\right\rangle_{\mathcal{H}}} \tag{J.46}$$

$$= \sqrt{\left\langle \Gamma\big(k(x) - k(x')\big), \Gamma\big(k(x) - k(x')\big)\right\rangle_{\mathcal{H}}}$$

for any $y \in \mathbb{R}^m$, where the second equation follows from the fact that $\langle k(y), k(y)\rangle_{\mathcal{H}} = 1$. By Lemma F.4, we have $\langle \Gamma g_1, \Gamma g_1\rangle_{\mathcal{H}} \leq (\lambda + n)\langle g_1, g_1\rangle/\lambda^3$ for any $g_1 \in \mathcal{H}$. Therefore, by (J.46), we have

$$\left|\Gamma K(x, y) - \Gamma K(x', y)\right| \leq \sqrt{(\lambda + n)\langle k(x) - k(x'), k(x) - k(x')\rangle_{\mathcal{H}}/\lambda^3} \tag{J.47}$$

Similar with (F.7), we have $\langle k(x) - k(x'), k(x) - k(x')\rangle_{\mathcal{H}} \leq \bar{\epsilon}^2$ when $\|x - x'\|_2 \leq \bar{\epsilon}$. By (J.47), we have $|\Gamma K(x, y) - \Gamma K(x', y)| \leq \sqrt{(\lambda + n)\bar{\epsilon}^2/\lambda^3}$. Therefore, $\mathcal{C}'$ is a $\sqrt{(\lambda + n)\bar{\epsilon}^2/\lambda^3}$-covering of $\mathcal{H}_1$ with regard to the infinity norm. We also have $|\mathcal{C}'| = |\mathcal{C}| \leq (R/\bar{\epsilon})^m$. We conclude the proof of Lemma F.5 by setting $\bar{\epsilon} = \epsilon\lambda\sqrt{\lambda/(\lambda + n)}$. $\qquad\square$

### J.12   PROOF OF LEMMA J.4

*Proof.* Since $e^x \leq 1 + x + (e - 2)x^2$ holds for any $x \leq 1$, we have

$$\mathbb{E}\big[\exp(\lambda X_\tau) \mid \mathcal{F}_{\tau-1}\big] \leq \mathbb{E}\big[1 + \lambda X_\tau + (e - 2)\lambda^2 X_\tau^2 \mid \mathcal{F}_{\tau-1}\big] = \mathbb{E}\big[1 + (e - 2)\lambda^2 X_\tau^2 \mid \mathcal{F}_{\tau-1}\big]$$

$$\leq \exp\Big\{\mathbb{E}\big[(e - 2)\lambda^2 X_\tau^2 \mid \mathcal{F}_{\tau-1}\big]\Big\}$$

when $\lambda \leq 1/c$. Therefore, when we define $Y_t = \exp\{\lambda \sum_{\tau=1}^t X_\tau - (e - 2)\lambda^2 \sum_{\tau=1}^t \mathbb{E}[X_\tau^2 \mid \mathcal{F}_{\tau-1}]\}$, we have $\mathbb{E}[Y_t \mid \mathcal{F}_{t-1}] \leq \mathbb{E}[Y_{t-1} \mid \mathcal{F}_{t-1}]$. Using induction, we have $\mathbb{E}[Y_n] \leq 1$. Therefore, we have

$$P\Big(\sum_{\tau=1}^n X_\tau \leq (e - 2)\lambda \sum_{\tau=1}^n \mathbb{E}[X_\tau^2 \mid \mathcal{F}_{\tau-1}] + \log(1/\delta)/\lambda\Big) \geq 1 - \delta$$

by Markov inequality. Thus, we conclude the proof of Lemma J.4. $\qquad\square$

### J.13   PROOF OF LEMMA J.5

*Proof.* By the property of the inverse operator, it remains to prove that

$$\lambda\langle g, g\rangle_{\mathcal{H}}/(\lambda + n) \leq \langle g, \Gamma_1^{-1/2}\Gamma_2\Gamma_1^{-1/2}g\rangle_{\mathcal{H}} \leq (\lambda + n)\langle g, g\rangle_{\mathcal{H}}/\lambda$$

for any $g \in \mathcal{H}$. By Lemma D.4, we have $\langle g_1, \Gamma_2 g_1\rangle_{\mathcal{H}} \leq (\lambda + n)\langle g_1, g_1\rangle_{\mathcal{H}}$. Therefore, we have

$$\langle g, \Gamma_1^{-1/2}\Gamma_2\Gamma_1^{-1/2}g\rangle_{\mathcal{H}} = \langle \Gamma_1^{-1/2}g, \Gamma_2\Gamma_1^{-1/2}g\rangle_{\mathcal{H}} \leq (\lambda + n)\langle \Gamma_1^{-1/2}g, \Gamma_1^{-1/2}g\rangle_{\mathcal{H}}.$$

By Lemma D.4, we have $\langle g_1, \Gamma_1^{-1}g_1\rangle_{\mathcal{H}} \leq \langle g_1, g_1\rangle_{\mathcal{H}}/\lambda$ for any $g_1 \in \mathcal{H}$. Therefore, we have

$$\langle g, \Gamma_1^{-1/2}\Gamma_2\Gamma_1^{-1/2}g\rangle_{\mathcal{H}} \leq (\lambda + n)\langle g, \Gamma_1^{-1}g\rangle_{\mathcal{H}} \leq (\lambda + n)\langle g, g\rangle_{\mathcal{H}}/\lambda.$$

Similarly, by Lemma D.4, we have $\langle g_1, \Gamma_2 g_1\rangle_{\mathcal{H}} \geq \lambda\langle g_1, g_1\rangle_{\mathcal{H}}$. Therefore, we have

$$\langle g, \Gamma_1^{-1/2}\Gamma_2\Gamma_1^{-1/2}g\rangle_{\mathcal{H}} = \langle \Gamma_1^{-1/2}g, \Gamma_2\Gamma_1^{-1/2}g\rangle_{\mathcal{H}} \geq \lambda\langle \Gamma_1^{-1/2}g, \Gamma_1^{-1/2}g\rangle_{\mathcal{H}}$$

By Lemma D.4, we have $\langle g_1, \Gamma_1^{-1}g_1\rangle_{\mathcal{H}} \geq \langle g_1, g_1\rangle_{\mathcal{H}}/(\lambda + n)$ for any $g_1 \in \mathcal{H}$. Therefore, we have

$$\langle g, \Gamma_1^{-1/2}\Gamma_2\Gamma_1^{-1/2}g\rangle_{\mathcal{H}} \geq \lambda\langle g, \Gamma_1^{-1}g\rangle_{\mathcal{H}} \geq \lambda\langle g, g\rangle_{\mathcal{H}}/(\lambda + n).$$

Thus, we conclude the proof of Lemma J.5. $\qquad\square$

