# OpenReview forum: "Optimistic Exploration with Learned Features Provably Solves Markov Decision Processes with Neural Dynamics"
_ICLR.cc/2023/Conference — ICLR 2023 poster_

### Official Review · Reviewer_DNrX · 2022-10-16

**Confidence:** 4
**Correctness:** 1
**Technical Novelty And Significance:** 2
**Empirical Novelty And Significance:** 2
**Recommendation:** 3

**Clarity, Quality, Novelty And Reproducibility:**

Many elements in the paper do not seem well supported/nor correct or they are at least not written in a way that makes the paper understandable. Here is a subset of these elements:

- Beginning of page 5, it is written that "Thus, (3.4) can be equivalently written as (...)". It is however very unclear how the authors manage that. Where is the exponential for instance?
- The following paragraph is strange

"To simplify the presentation of the algorithm in our work, we introduce an extended MDP, where we assign meanings to steps h = −1, 0, H + 1, and H + 2. In particular, the interaction of an agent with the extended MDP starts with a dummy initial state s−1. During the interaction, all the dummy state and action sequences {s−1, a−1, s0, a0} lead to the same initial state sinit. Moreover, the agent is allowed to interact with the environment for two steps after observing the final state sH+1 of an episode. Nevertheless, the agent only collects the reward rh(sh, ah) at steps h ∈ [H], which leads to thesamelearningobjectiveastheoriginalMDP.Inaddition,wedenoteby[H]+ =[−1,0,...,H+2] the set of steps in the extended MDP. In the sequel, we do not distinguish between an MDP and an extended MDP for the simplicity of presentation."

It basically introduce an extended MDP that is not actually different in a meaningful way than the original MDP. All that to say at the end of the paragraph that both will be considered identical. What is the the point?
- Almost the whole page 6 is a very long explanation for the basic operation of collecting a tuple $(s,a,r,s')$ and adding it to the dataset. Why is such a long explanation needed here? What does the $\tau$ superscript mean? The same symbols are used with different subscripts/superscripts which makes the paper not clear because there is no definition of what adding these subscripts/superscripts mean.
- Equations 4.3 to 4.5 are introduced without any explanation on how it relates to the earlier introduced theory (Equations 3.2 to 3.4)

Minor comments:
- Below Equation 3.4, it is written "$(h,s,a) \in \Phi \times \mathcal S \times \mathcal A$". Why is $h \in \phi$ possible as $\phi$ seems to be defined three lines earlier as a function space?

**Strength And Weaknesses:**

Strength:
The aim of the paper seems ambitious and interesting

Weaknesses:
The paper is not clear

**Summary Of The Paper:**

The paper aims at building "novel algorithm that designs exploration incentives via learnable representations of the dynamics model by embedding the neural dynamics into a kernel space induced by the system noise."

**Summary Of The Review:**

The paper is not clear

---

> ### Author Response · Authors · 2022-11-11
> **Author response**
>
> Thank you very much for your valuable comments. We admit that our presentation might be intense since we want to present a thorough analysis rigorously, and we edit the notation table to make it clearer. We address the concerns as follows.
>
> >Beginning of page 5, it is written that "Thus, (3.4) can be equivalently written as (...)". It is however very unclear how the authors manage that. Where is the exponential for instance?
>
> a. (Explanation on Equation on Page Five.) The equation on page 5  holds since the inner product has the form of exponential when the corresponding reproducing kernel Hilbert space (RKHS) is induced by the Gaussian radius-based kernel (RBF). The lemma in Appendix D.1 shows that, there exists a reproducing kernel Hilbert space $\mathcal{H}$ with the inner product $\langle\cdot,\cdot\rangle_\mathcal{H}$ and the kernel $K$, such that $ \langle K(x_1,\cdot), K(x_1,\cdot)\rangle_\mathcal{H}=\exp(-\Vert x_1-x_2\Vert_2^2/2)$.  Due to the space limit, we defer the details on the RKHS to Appendix D. We will edit our paper to highlight the reference to the appendix.
>
> >The following paragraph is strange
>
> b. (Explanation on Extended MDPs.) We introduced the extended MDP to simplify the presentation of our algorithm. In our algorithm, we combine the greedy policy with the uniform policy to explore the environment eﬀiciently. Without the notion of the extended MDP, it will be complicated to present the boundary case of the sampling procedure, where $h=1$, $2$, $H-1$, and $H$, rigorously. Therefore, we introduced the extended MDP to make the presentation simpler while keeping it rigorous.
>
> >Almost the whole page 6 is a very long explanation for the basic operation of collecting a tuple  $(s,a,r,s^\prime)$ and adding it to the dataset. Why is such a long explanation needed here?
>
> c. (Explanation on Sampling Procedure.) We spend much effort on describing the sampling procedure since the sampling
> procedure is not standard, and we want to make it clear while keeping the description rigorous. Typically, the agent explores the environment with the greedy policy with regard to the learned model. However, since our feature is unknown NN, we need to combine the greedy policy with the uniform policy to control the feature estimation error and provably avoid ineﬀicient exploration. The algorithm section is devoted to making the presentation of this novel sampling procedure clear and rigorous.
>
> >What does the  superscript $\tau$ mean? The same symbols are used with different subscripts/superscripts which makes the paper not clear because there is no definition of what adding these subscripts/superscripts mean.
>
> d. (Explanation on $\tau$.) We use $n$ to iterate from $1$ to $N$, and use $\tau$ to iterate from $1$ to $n$ in the whole paper. We will revise the notation table in the appendix to explain this notation. Thank you for your comment.
>
> >Equations 4.3 to 4.5 are introduced without any explanation on how it relates to the earlier introduced theory (Equations 3.2 to 3.4)
>
> e. (Explanation on the relation between (4.3)-(4.5) to (3.2)-(3.4).) We do not make any extra explanation on (4.3) and (4.4) since (4.3)-(4.4), which is the least square estimator and the maximum likelihood estimator, are standard techniques for model estimation and can be used in any model. Nevertheless, we will add a discussion on the connection between (3.2)-(3.4) and (4.3)-(4.4).  We have commented below (4.5) that the form of the bonus in (4.5) aligns with the bonus in other previous works whose model is similar to our model in (3.4), which connects (4.5) with our model in (3.2) to (3.4).
>
> >Why is  $ h\in\Phi$ possible as $\Phi$  seems to be defined three lines earlier as a function space?
>
> f. (Explanation on $ h\in\Phi$.) We will change $h$ to another notation to avoid causing confusion. Thank you for pointing this out.

---

> > ### Comment · Reviewer_DNrX · 2022-11-16
> > **Not fully convinced by the reply from the authors**
> >
> > Thanks for the clarifications.
> >
> > I'm however still not convinced by the presentation of the paper. There are many elements that are introduced in a complicated way without a clear motivation. This takes in some cases large parts of the paper for seemingly simple things while other elements that are much more important for the core of the contribution are left in the appendix (i.e. hidden for the reader to easily understand).
> >
> > Let me focus on only two elements that I believe are clear examples of that and that were already pointed out in my initial review:
> > - The paragraph that starts with "To simplify the presentation of the algorithm in our work" introduces new notations and a new "extended MDP". I still do not understand the usefulness of this. It takes a lot of time to process the information for the reader and it is then written "In the sequel, we do not distinguish between an MDP and an extended MDP for the simplicity of presentation". I don't believe that this presentation can make sense for writing a clear and solid contribution.
> > - Almost the whole page 6 is a very long explanation for the basic operation of collecting a tuple $(s,a,r,s')$. The authors mention in the rebuttal that it is done to keep the "procedure clear and rigorous". I hardly believe that such a sampling procedure can't be described in a simple manner (potentially with a few non-important details in the appendix).
> >
> > Based on these two examples, I'm questioning many of the complexities that are present in the paper. I'm most likely going to keep my score of reject. However, if the authors still want to convince me, they can maybe precisely describe 2 or 3 key parts for each of these 2 examples that are (1) so important that justify why it is not in the appendix and (2) where they are key in the paper to justify the complexity of notations.

---

> > > ### Author Response · Authors · 2022-11-17
> > > **Author response**
> > >
> > > Thank you very much for your valuable comments. We have revised our submission to simplify the presentation of our algorithm and defer the formal and rigorous presentation of our algorithm to the appendix. Please let us know if this revision is appropriate. We also address some potential concerns as follows.
> > >
> > > a. (Necessity of Notations) We use $h$, $n$, $\bar{h}$, $\tau$, and $i$ when presenting our algorithm.
> > >
> > > - The notation $h$ is necessary since it is the index that iterate from $1$ to $H$.
> > >
> > > - The notation $n$ is necessary since it is the index that iterate from $1$ to $N$.
> > >
> > > - The notation $\tau$ is necessary since it is the index that iterate from $1$ to $n$, which naturally appears when describing the estimator and the bonus in each iteration $n$.
> > >
> > > - The notation $\bar{h}$ is necessary since it is the index that iterate from $1$ to $h$, which naturally appears when describing the sampling procedure of $\mathcal{D}^n_{h,i}$.
> > >
> > > - The notation $i$, which is the second subscript of the labeled data, is necessary since it indicates the number of the uniform policy we take for sampling the data. We remark that we treat the data with different $i$ differently when constructing the estimators and the bonuses. We only use the data with $i=1$ when constructing the exploration bonuses, and only use the data with $i=1$, $2$ when estimating the model. Since we treat the data with different $i$ differently in the following algorithm and analysis, it is important for us to label them differently when describing the sampling scheme. Therefore, the index $i$ is also necessary.
> > >
> > > b. (Description of the Revision) Thank you very much for your advice, we have deferred some of rigorous description, including the definition of the extended MDP, to the appendix. Please let us know if this revision is appropriate. Here we want to emphasize that this definition is important to simplify the rigorous presentation of the algorithm. Without this notion, we need to describe the algorithm for sampling $\mathcal{D}^n_{1,1}$, $\mathcal{D}^n_{1,2}$, $\mathcal{D}^n_{2,2}$, $\mathcal{D}^n_{H-1,0}$, $\mathcal{D}^n_{H,0}$, and $\mathcal{D}^n_{H,1}$ case by case, which makes the presentation tedious.
> > >
> > > c. (Necessity of Uniform Sampling) As we mentioned in the proof sketch of the revised version, the uniform policy in the sampling scheme is necessary since it enables us to bound the influence of the distribution shift and show that the bonuses defined by the learned feature are valid uncertainty quantification. Since the exploration incentive in our algorithm is constructed with the learned feature and might have errors, the exploration incentive might be invalid. Therefore, we combine the greedy policy with the uniform policy to provably avoid inefficient exploration.

---

> > > ### Author Response · Authors · 2022-11-17
> > > **Discussion**
> > >
> > > We are wondering if our responses and revision addressed your concerns. We will be happy to answer if there are additional issues/questions.

---

> > > ### Author Response · Authors · 2022-12-09
> > > **Discussion**
> > >
> > > We are wondering if our responses and revision addressed your concerns. We will be happy to answer if there are additional issues/questions.

---

### Official Review · Reviewer_rz87 · 2022-10-23

**Confidence:** 2
**Correctness:** 3
**Technical Novelty And Significance:** 3
**Empirical Novelty And Significance:** Not applicable
**Recommendation:** 6

**Clarity, Quality, Novelty And Reproducibility:**

Modulo my comments above, the works seem to be of a high quality, original, and a good contribution.


**Strength And Weaknesses:**

This is an **intensely** theoretical paper, in a domain that I am not overly familiar with.  While I have an understanding of the high-level findings, I did not and cannot faithfully review the derivations in the supplement (which is over 40 pages long).

This paper is really in two parts: there is the high-level algorithm and result (that an MDP with NN link functions can be handled efficiently), and a long sequence of incredibly low-level results proving out the theory that underpins this algorithm.  I think there is merit in this paper for both audiences at ICLR:  a high-level audience which will benefit from the formalism, discussion of MDPs and the final result, and a low-level audience that will enjoy the theoretical analysis.

## Strengths
1.  The paper does an excellent job of defining the core RL terms and formalism.
2.  The core idea is appealing, and provides a good generalisation of existing ideas.
3.  The authors appear to have done an incredibly thorough job of theoretically analysing the topic.
4.  Although a bit too theoretically and technically heavy for my liking, the material that is there is presented exceptionally well, with basically no typos, typographical errors or grammatical errors.

## Weaknesses

### Challenging presentation.
The intensity of theoretical analysis is also a weakness.  While the core claims seem reasonable, I don’t believe many, if any, readers (or reviewers!) will review the supplementary material.  While the authors are to be commended for the depth of their theoretical analysis, they must do a better job in breaking down the analysis and making it more accessible for the average reader.  Sketch proofs are the main way of doing this.

Some of the theoretical material and nuanced comparison could be dropped to the supplement and replaced with higher-level summaries to expand the target audience of the paper.

### Methodological Comments
I understand that assuming oracles is common in RL analysis (because the alternative is realistically intractable to analyse), but I would like to see some comment on what the impact of not being able to use an oracle in practice is.

Building on this, I would like the authors to confirm that my evaluation of their method is correct:  You essentially rollout under the current policy, and then take two steps from a uniform policy.  This uniform sampling increases the diversity, but degrades the sampling efficiency.  I am struggling to see the relevance of exactly two steps under a uniform policy, after rolling out for P steps.  Is there not a more principled way of doing this?  Eg. rolling out under a policy with an entropy increase (adding uniform at each timestep, increasing the variance)?  Why is exactly two additional steps the best option?

Then, the “method” is using an oracle to “solve” for the transition and reward functions from a very expressive variational family using these extra two steps.  Why do you use just the extra two steps?  Isn’t it more sample efficient to use the whole trajectory?  Does using just the final two steps bias the function estimator to be biased towards the regions of state space with a high occupancy at the *end* of the trajectory?

Then the remaining analysis is showing that the approximation error is bounded with a tractable complexity?  One thing that I don’t understand is the “exploration bonus”.  This exploration bonus modifies the policy that is currently held, and so does this bonus go to zero during optimization, so that the final policy is “purely exploit”?  The policy learning is also done exactly through using the planning algorithm in the appendix?  Does this algorithm impose any further constraints on the space?  Does inaccuracies in the planning algorithm lead to pathological failures in the algorithm?


### Empirical Validation
I acknowledge that ICLR has a strong purely theoretical remit.  However, for an algorithm as practically motivated and generally applicable as this, I would have liked to have seen at least a toy example.  This is partly to “validate” the theory, but also to help the reader understand the different facets of the algorithm.  While this is not a requirement for my response, I implore the authors to at least explore applying this to an LQR, an in-built MuJoCo example, even something like mountain cart.  This would help the reader a lot, and would further increase the reach and impact of the paper.


## Minor Weaknesses / Typographical Comments

(a)  Advanced concepts such as covering number and Eluder number are completely undefined (although maybe there is little in this paper for anyone that doesn’t already understand those terms).

(b)  The bold-facing of fonts is unnecessary.

(c)  Are the terms in the norm in (3.4) reversed compared to the preceding and next equation?  It would probably be clearer if they were the same way around in each expression.


**Summary Of The Paper:**

This paper analyses a method for learning transition and reward dynamics in MDPs.  They show that a more generally parameterised MDP can be solved efficiently.  No empirical results are presented, but extensive theoretical analyses are.


**Summary Of The Review:**

Since I cannot faithfully comment on the full theoretical extent of the work, but the content that I have analysed is good, I will rate this paper as a weak accept.

---

> ### Author Response · Authors · 2022-11-11
> **Author response**
>
> Thank you very much for your valuable comments. We address the concerns as follows.
>
> >The intensity of theoretical analysis is also a weakness. While the core claims seem reasonable, I don’t believe many, if any, readers (or reviewers!) will review the supplementary material. While the authors are to be commended for the depth of their theoretical analysis, they must do a better job in breaking down the analysis and making it more accessible for the average reader. Sketch proofs are the main way of doing this.
>
> a. (Improving Presentation.) Thank you very much for your valuable comments. We will add a proof sketch to present our analysis better.
>
> >Building on this, I would like the authors to confirm that my evaluation of their method is correct: You essentially rollout under the current policy, and then take two steps from a uniform policy. This uniform sampling increases the diversity, but degrades the sampling efficiency. I am struggling to see the relevance of exactly two steps under a uniform policy, after rolling out for P steps. Is there not a more principled way of doing this? Eg. rolling out under a policy with an entropy increase (adding uniform at each timestep, increasing the variance)? Why is exactly two additional steps the best option?
>
> b. (Role of Uniform Sampling.) We combine the uniform sampling with the current policy to handle the distribution shift. In order to bound the suboptimality from the above, we show that (1) the uncertainty of the learned model can be bounded from the above by the bonus we construct (Lemma E.5), and (2) the bonus we construct can be bounded by above by the bonus defined by the unseen true feature (Lemma I.2), which can be telescoped. We need to handle the distribution shift when proving the above inequalities. Two steps of uniform sampling improve the coverage of the sampling and enable us to prove the inequalities in the presence of the distribution shift that occurs when proving Lemma E.5 and Lemma I.2.
>
> A potential improvement can be combining the greedy policy with the uniform policy
> in each step instead of executing them separately. We think that the current algorithm is more straightforward for the reader to connect the algorithm with the proof and leave the combination for future research.
>
> >Then, the “method” is using an oracle to “solve” for the transition and reward functions from a very expressive variational family using these extra two steps. Why do you use just the extra two steps? Isn’t it more sample efficient to use the whole trajectory? Does using just the final two steps bias the function estimator to be biased towards the regions of state space with a high occupancy at the end of the trajectory?
>
> c. (Property of the estimator.) We only use the final two steps for model estimation since the sample distribution has better coverage over the state-action space, and we can obtain a desirable upper bound of the estimation error and explore the environment efficiently. In our algorithm, we construct the bonus based on the learned feature. However, the learned feature has an error, and the visitation measure of the corresponding greedy policy is biased due to the error. To deal with this problem, we combine the greedy policy with the uniform policy. Thanks to the uniform policy, the distribution of the samples that we collect has better coverage over the state-action space, and we can obtain a desirable upper bound of the estimation error and explore the environment efficiently
>
> >Then the remaining analysis is showing that the approximation error is bounded with a tractable complexity?
>
> d. (Remaining Analysis.) Bounding the approximation error with the complexity measure is a part of our analysis, but it is not the only part. We also need to connect the regret with the approximation error and show that the summation of the approximation error is bounded even with the presence of the distribution shift.
>
> >One thing that I don’t understand is the “exploration bonus”. This exploration bonus modifies the policy that is currently held, and so does this bonus go to zero during optimization, so that the final policy is “purely exploit”?
>
> e. (Exploration bonus.) When the environment is fully explored, the bonus goes to zero and the policy will be almost purely exploitive. This is desirable since we don’t need to explore the environment when the environment is fully explored.
>
> >The policy learning is also done exactly through using the planning algorithm in the appendix? Does this algorithm impose any further constraints on the space?
>
> f. (Planning Algorithm.) The greedy policy in each iteration can be obtained using Algorithm 3 in our paper. Algorithm 3 does not impose any further constraints on the space, and the final output policy of our algorithm is the combination of the greedy policy in each iteration. We can use other planning algorithms in each iteration, and the suboptimality can still be bounded if the error of the planning algorithm is bounded.

---

> ### Author Response · Authors · 2022-11-11
> **Author response**
>
> Thank you very much for your valuable comments. We address the concerns as follows.
>
> >Does this algorithm impose any further constraints on the space? Does inaccuracies in the planning algorithm lead to pathological failures in the algorithm?
>
> g. (Influence of Inaccurate Planning Oracle.) This will not happen. When the error of the planning oracle is bounded by $\eta$, the suboptimality of our algorithm can still be bounded from the above. The planning oracle allows us to bound Term (b) in (E.9) (Page 21) from the above. If we have a planning oracle whose error can be bounded by $\eta$, the upper bound of Term (b) in (E.9) will be changed from 0 to $\eta$.
>
> >I acknowledge that ICLR has a strong purely theoretical remit. However, for an algorithm as practically motivated and generally applicable as this, I would have liked to have seen at least a toy example. This is partly to “validate” the theory, but also to help the reader understand the different facets of the algorithm. While this is not a requirement for my response, I implore the authors to at least explore applying this to an LQR, an in-built MuJoCo example, even something like mountain cart. This would help the reader a lot, and would further increase the reach and impact of the paper.
>
> h. (Empirical Validation.) Thank you very much for your valuable comments. We will try to conduct empirical validation and will post the result if we can successfully validate our algorithm.
>
> >Minor Weaknesses / Typographical Comments
>
> i. Thank you very much for pointing these out, we will revise accordingly.

---

> ### Author Response · Authors · 2022-11-17
> **Discussion**
>
> We are wondering if our responses and revision addressed your concerns. We will be happy to answer if there are additional issues/questions.

---

> > ### Comment · Reviewer_rz87 · 2022-11-18
> > **Staying Put.**
> >
> > To the authors,
> >
> > Thank you for your response, and sorry for the delay in my response.  I am glad the authors took some of the feedback on board and modified the paper.  Including proof sketches and more summaries has made it somewhat more accessible and self-contained.  Unfortunately, I am not swayed enough to vote strongly for acceptance.  I still think there is work to be done thinning the presentation and improving accessibility to a less hyper-specialist audience.  I would also strongly encourage the authors to provide some -- any -- empirical exploration.
> >
> > I would also encourage the authors to consider seeking out a more specialist venue or journal, where you can let the length and intensity of analysis breathe, unlike in a condensed conference format.  It may well yield a better paper and help the paper reach a more suitable audience.
> >
> > Good work, and good luck going forwards.
> > rz87

---

> > > ### Author Response · Authors · 2022-11-18
> > > **Author response**
> > >
> > > We very much appreciate your detailed feedback. In the latest version (updated yesterday), we incorporated many great suggestions from you and the other reviewer and simplified the current presentation. For example, we simplify the presentation of the sampling scheme in our algorithm significantly. We also provide a proof sketch to improve the accessibility to a less hyper-specialist audience. We also welcome further advice on further improving the presentation.
> > >
> > >
> > > Our current work is not aimed at achieving state-of-the-art performances. Our stylized algorithm is mirrored by many existing algorithms that are known to achieve outstanding empirical performances (For example, the model state $s_{t+1}$ in [1] can be interpreted as the feature of $s_{h+1}$ in our model.). Instead, we aim to explain why and when it is possible to learn a nonlinear dynamic system with provable statistical and computational guarantees, which we believe is of interest to the ICLR audience.
> > >
> > > [1]. Hafner, Danijar, Timothy Lillicrap, Jimmy Ba, and Mohammad Norouzi. "Dream to control: Learning behaviors by latent imagination." arXiv preprint arXiv:1912.01603 (2019).

---

### Official Review · Reviewer_UMwu · 2022-10-24

**Confidence:** 4
**Correctness:** 4
**Technical Novelty And Significance:** 3
**Empirical Novelty And Significance:** Not applicable
**Recommendation:** 8

**Clarity, Quality, Novelty And Reproducibility:**

The statements in the paper is very clear, and the organization is pretty good. See above for the discussion on the novelties.

**Strength And Weaknesses:**

I found this paper very well motivated with inspiring results. The major observation made in the paper is that the kernelized nonlinear regulator, linear MDPs, and neural-nets parameterized MDPs can be summarized by the so-called MDPs with neural dynamics models. The neural dynamics models assume the transition dynamics is proportional to the Gaussian RBF kernel in the low-rank embedding space of the states and actions. To find the optimal policies, the proposed algorithm ELNF adapts estimation and planning techniques based on MLE estimation scheme in low-rank MDPs and kernelized LSVI algorithm proposed by Yang et al. (2020). However, it requires much effort to adapt these techniques in the neural dynamics model, which I believe is another key novelty.

Nevertheless, I do have some concerns about the results of the paper.

1. The first question is about the number of actions appearing in the sample complexity. For general MDPs with neural dynamics, the number of actions should also be large. Examples include continuous control tasks, combinatorial optimization tasks, etc. The famous AlphaTensor algorithm has exponential number of actions. I think it is better to discuss how to perform "discretization" or establish some Lipschitz assumptions on the action space to reduce the cost of uniform sampling.
2. Intuitively the log-covering number of neural nets should have polynomial dependency on the number of parameters. It seems that the sample efficient neural dynamics model does not include the well-known over parameterized neural nets whose number of parameters are very large.
3. The authors claim that the ELNF algorithm is computational-efficient if the optimization oracle (4.3) and (4.4) can be implemented efficiently. How does this oracle implemented when the dynamics is parameterized by neural nets?
4. I think there should be more discussions on the decay rate $\gamma$ of the covering number. How large is this decay rate for common neural nets? Could you provide some intuitions on why $\gamma$ affects the sample complexity in the exponents?
5. Why do you need step -1 and step 0, and how do you assure $s_1$ is exactly the initial state?

A minor question:
Page 9 (comparison with Done et al.(2021)): $\epsilon=0$ should be $\gamma=0$

**Summary Of The Paper:**

This paper studies the exploration problem for MDPs with transition kernel captured by general function classes with bounded covering number. The transition dynamics is assumed to be a low-rank Gaussian RBF kernel with unknown features of states and actions, which captures both linear MDPs and a number of neural nets as special cases. It proposes an exploration algorithm named ELNF that has provable sublinear sample complexity to find a near-optimal policy. The sample complexity is guaranteed to depend only on the covering number instead of the Eluder dimension, which is shown to be exponentially large even for two-layer neural nets.


**Summary Of The Review:**

This is a good paper studying the neural dynamics models in RL, though I do have some problems about the results. I recommend for accepting the paper, and may raise my score if my concerns are addressed properly.

---

> ### Author Response · Authors · 2022-11-11
> **Author response**
>
> Thank you very much for your valuable comments. We address the concerns as follows.
> > The first question is about the number of actions appearing in the sample complexity. For general MDPs with neural dynamics, the number of actions should also be large. Examples include continuous control tasks, combinatorial optimization tasks, etc. The famous AlphaTensor algorithm has exponential number of actions. I think it is better to discuss how to perform "discretization" or establish some Lipschitz assumptions on the action space to reduce the cost of uniform sampling.
>
> a. (Discussion on Number of Actions.) Thank you very much for pointing this out. We will add a discussion on the number of actions. Our sample complexity depends on the number of actions, such a dependency is essential due to the following reason.  Recall that the feature mapping involves an unknown neural network. To bound the regret, we need to control the error incurred in estimating the unknown feature. By taking uniform actions, we are able to handle such an error across different actions. Similar sampling schemes also appear in [3] and [4].
>
> Furthermore, we do not consider the case of the continuous action space since efficient exploration with the presence of the continuous action space and NN is essentially hard. Theorem 5.1 of [1] provides an exponential lower bound of the sample complexity of finding the global optimal policy for a smooth reward function. These examples show the hardness of sample-eﬀicient exploration in the presence of neural networks and large action space. In spite of the dependency on the number of actions, our result is still significant since it demonstrates that we can explore the environment eﬀiciently when the dynamics is described by a neural network. More importantly, we prove that the complexity of our algorithm does not depend on the Eluder dimension of NN class, but only depends on its log-covering number, and the effective dimension of the kernel induced by noise. We would like to highlight that, unlike previous results, the eluder dimension of the NN class (which is prohibitively large) does appear in the upper bound. Instead, the effective dimension plays the role of characterizing the tradeoff between exploration and exploitation. Therefore, we can efficiently explore the environment in the neural dynamics when $\vert \mathcal{A}\vert$ is small, which is new and significant to the community.
>
> >Intuitively the log-covering number of neural nets should have polynomial dependency on the number of parameters. It seems that the sample efficient neural dynamics model does not include the well-known over parameterized neural nets whose number of parameters are very large.
>
>
> b. (Discussion on Overparameterized Neural Network.)  [5] shows that when training an overparameterized NN, ​​the dynamics of the training process can be captured by the framework of the neural tangent kernel (NTK), and the covering number of a bounded subset in the Hilbert space induced by the neural tangent kernel satisfies Assumption 5.3 with $\gamma=0$.  Let $f(\cdot, W)$ denote the overparameterized NN with parameter $W$, let $m$ denote the width of the NN, let $\phi(\cdot, W)$ denote the gradient of $f(\cdot, W)$ with respect to $W$, and let $W^{(0)}$ denote the initial value of $W$. Conditioning on the realization of $W^{(0)}$, we define the kernel $K_m$ by
> \begin{align*}
> K_m(z,z^\prime)=\Bigl\langle \phi\bigl(z, W^{(0)}\bigr), \phi\bigl(z^\prime, W^{(0)}\bigr)\Bigr \rangle.
> \end{align*}
>
>  As shown in [6], we have
>  \begin{align*}
>  f(\cdot, W) \approx \hat{f}(\cdot, W)=f(\cdot, W^{(0)}) + \Bigl\langle   W-W^{(0)}, \phi\bigl(\cdot , W^{(0)}\bigr) \Bigr\rangle.
>  \end{align*}
>  When $m$ is sufficiently large, the linearized function $\hat{f}(\cdot, W)$ belongs to an RKHS with kernel $K_m$. Moreover, as $m$ goes to infinity, due to the random initialization, $K_m$ converges to a kernel $K_{\mathrm{NTK}}$. Appendix B.2 of [6] provides examples of overparameterized NNs where the eigenvalues of the integral operator induced by the neural tangent kernel decay quickly, and the covering number satisfies Assumption 5.3 with $\gamma=0$.
>
> >The authors claim that the ELNF algorithm is computational-efficient if the optimization oracle (4.3) and (4.4) can be implemented efficiently. How does this oracle implemented when the dynamics is parameterized by neural nets?
>
> c. (Discussion on Optimization Oracle.) Equation (4.3) is the least square estimation, which is frequently used in machine learning. When we assume that the normalization coeﬀicient is a constant, Equation (4.4) is equivalent to
>
> \begin{align*}
> \min_{\phi,\psi}\sum_{i=1}^2\sum_{(s_h,a_h,r_h,s_{h+1})\in \mathcal{D}^n_h} \Vert\phi_h(s_h,a_h)-\psi_{h+1}(s_{h+1})\Vert^2,
> \end{align*}
>
> which is the least square estimator and can be easily applied.

---

> ### Author Response · Authors · 2022-11-11
> **Author response**
>
> Thank you very much for your valuable comments. We address the concerns as follows.
>
> >I think there should be more discussions on the decay rate $\gamma$ of the covering number. How large is this decay rate for common neural nets? Could you provide some intuitions on why $\gamma$ affects the sample complexity in the exponents?
>
> d. (Discussion on $\gamma$.) The decay parameter $\gamma$ characterizes the complexity of the neural
> network class. Bigger $\gamma$ implies that the class has higher complexity, and we need more samples to obtain a valid estimator of the model. Theorem 3.3 in [2] provides an upper bound of the covering number of neural networks. We provide two examples in Appendix C, where the NN class satisfies Assumption 5.3 with $\gamma=0$. Overparameterized NN is another example. As we mention in the Comment b ​,​the dynamics of the training process can be captured by the framework of the neural tangent kernel (NTK) when training an overparameterized NN. When the activation and the distribution of the initialization are chosen properly, the eigenvalues of the integral operator of the corresponding NTK have an exponential decay, and the covering number of a bounded subset in the Hilbert space induced by the neural tangent kernel satisfies Assumption 5.3 with $\gamma=0$.
>
> >Why do you need step -1 and step 0, and how do you assure  is exactly the initial state?
>
> e. (Explanation on Extended MDP.) The states $s_{-1}$ and $s_0$ are the dummy states that we define to simplify the rigorous presentation of the algorithm. In other words, the states $s_{-1}$ and $s_0$ do not actually exist, and we make them up just to simplify the presentation of our algorithm in the boundary case. In our algorithm, we combine the greedy policy with the uniform policy to explore the environment eﬀiciently. Without the notion of the extended MDP, it will be complicated to present the boundary case of the sampling procedure, where $h=1$, $2$, $H-1$, and $H$, rigorously. Therefore, we introduced the extended MDP to make the presentation simpler while keeping it rigorous.
>
> >A minor question: Page 9 (comparison with Done et al.(2021)):  $\epsilon=0$ should be $\gamma=0$.
>
> f. Thank you very much for pointing this out, we will revise accordingly.
>
> [1] Dong, Kefan, Jiaqi Yang, and Tengyu Ma. (2021). ”Provable model-based nonlinear bandit and reinforcement learning: Shelve optimism, embrace virtual curvature.”
>
> [2] Bartlett, Peter L., Dylan J. Foster, and Matus J. Telgarsky. ”Spectrally-normalized margin bounds for neural networks.” Advances in neural information processing systems 30 (2017).
>
> [3] Agarwal, Alekh, and Tong Zhang. "Model-based rl with optimistic posterior sampling: Structural conditions and sample complexity." arXiv preprint arXiv:2206.07659 (2022).
>
> [4] Du, Simon, et al. "Bilinear classes: A structural framework for provable generalization in rl." International Conference on Machine Learning. PMLR, 2021.
>
> [5] Jacot, Arthur, Franck Gabriel, and Clément Hongler. "Neural tangent kernel: Convergence and generalization in neural networks." Advances in neural information processing systems 31 (2018).
>
> [6] Yang, Zhuoran, et al. "Provably efficient reinforcement learning with kernel and neural function approximations." Advances in Neural Information Processing Systems 33 (2020): 13903-13916.

---

> > ### Comment · Reviewer_UMwu · 2022-11-18
> > **Good Paper**
> >
> > Thanks for your responses to my questions.
> >
> > Your respones have mostly addressed my concerns. I agree that the continuous action space setting is very difficult for the NN dynamics. As for the over-parameterized NN classes, the covering number of NTK classes does not depend on the number of parameters. One thing still not very clear is the quantitative role of $\gamma$ in the sub-optimality bound. In all the examples given in the paper (and the rebuttal), it holds that $\gamma = 0$. Therefore, it confuses me that why you define the covering number of NN classes with $\gamma$, instead of just saying $\gamma = 0$. Moreover, the upper bound provided by Theorem 3.3 in [1] holds for $\gamma = 2$, which is not covered in this paper because Theorem 5.3 requires $\gamma < 1/2$.
> >
> > In general, I still believe this is a good paper with enough contributions and have raised my score. However, I also agree the presentations (especially for less hyper-specialist readers) should be improved as mentioned by other reviewers. For example, you can provide some intuitions when constructing the exploration bonuses in (4.5), such as how these bonuses are derived from the kernerlized LSVI.
> >
> > [1] Bartlett, Peter L., Dylan J. Foster, and Matus J. Telgarsky. ”Spectrally-normalized margin bounds for neural networks.” Advances in neural information processing systems 30 (2017).

---

> > > ### Author Response · Authors · 2022-11-19
> > > **Author response**
> > >
> > > Thank you for your time reviewing our response. We are glad to hear that your concerns have been addressed.
> > >
> > > > One thing still not very clear is the quantitative role of $\gamma$ in the sub-optimality bound.
> > >
> > > Regarding the Role of $\gamma$
> > >
> > > We introduce the term $\gamma$ in the analysis for two reasons. First, we want to clearly quantify the influence of the decay rate. When $\gamma>0$, we need to select a bigger $\beta$ to balance the bias and the variance when constructing the bonus. The generalization error of the MLE also depends on $\gamma$. In our analysis, we clearly characterize the influence of bigger $\gamma$, and we believe that our analysis is valuable for the community. Second, allowing non-zero $\gamma$ makes our result more general. For example, Theorem D of [1] provides an example where $0<\gamma<1/2$.
> > >
> > > We also remark that although we do not cover the case where $\gamma=2$, our result is still novel and significant to the community since the suboptimality bound in our paper only depends on the covering number instead of the Eluder dimension of NNs and is general enough to include a large class of NN, which is shown in Appendix C. We also emphasize that even for the NN analyzed in [2], we might still obtain a mild decay rate of the covering number by using the number of parameters for the analysis instead of the norm-based analysis, which is still better than using the Eluder dimension since the Eluder dimension of the NN class can be exponentially large.
> > >
> > > >For example, you can provide some intuitions when constructing the exploration bonuses in (4.5), such as how these bonuses are derived from the kernerlized LSVI.
> > >
> > > Regarding Equation (4.5)
> > >
> > > Thank you very much for pointing this out. We will edit our paper to explain it. The bonus we defined in (4.5) is almost the same as the bonus defined in kernelized LSVI [2] except for two differences. The first difference is that we use the learned feature to define the bonus since we do not know the true feature. The second difference is that we only use the data whose second subscript is $1$, instead of all the data when defining the bonus.
> > >
> > >
> > > [1]. Cucker, Felipe, and Steve Smale. "On the mathematical foundations of learning." Bulletin of the American mathematical society 39.1 (2002): 1-49.
> > >
> > > [2] Bartlett, Peter L., Dylan J. Foster, and Matus J. Telgarsky. "Spectrally-normalized margin bounds for neural networks. " Advances in neural information processing systems 30 (2017).
> > >
> > > [3] Yang, Zhuoran, et al. "Provably efficient reinforcement learning with kernel and neural function approximations." Advances in Neural Information Processing Systems 33 (2020): 13903-13916.

---

> ### Author Response · Authors · 2022-11-17
> **Discussion**
>
> We are wondering if our responses and revision addressed your concerns. We will be happy to answer if there are additional issues/questions.

---

### Author Response · Authors · 2022-11-16
**Updated submission**

Hello all!

We have uploaded an updated version of the paper. Specifically, we have:

- Added a proof sketch of the main theorem in the appendix.
- Added the definition of the covering number in Appendix C.
- Added a comment on the explanation on the extended MDP.
- Modified the notation table.
- Added a comment on the optimization objective (4.4).
- Fixed the typographical problem that is pointed out by the reviewers.


We believe we have addressed most of the actionable feedback that we have received from the reviewers at this point, and are happy to discuss any remaining questions, feedback, or confusions that may exist. The deadline for us to make further modifications to the paper is Nov 18th.

---

### Author Response · Authors · 2022-11-17
**Updated Submission (2)**

Hello all!

We have uploaded an updated version of the paper. Specifically, we have:

- Simplify the presentation of the algorithm.

- Explain the role of the uniform policy in the proof sketch

- Move the proof sketch to the main text.

We believe we have addressed most of the actionable feedback that we have received from the reviewers at this point, and are happy to discuss any remaining questions, feedback, or confusions that may exist. The deadline for us to make further modifications to the paper is Nov 18th.

---

### Decision · Program_Chairs · 2023-01-20

**Decision:**

Accept: poster

**Justification For Why Not Higher Score:**

See above.

**Justification For Why Not Lower Score:**

See above.

**Metareview: Summary, Strengths And Weaknesses:**

This paper contains several novel, albeit highly theoretical results. It formalises a class of MDPs with nonlinear transitions that can be learned by neural networks; proposes an oracle-efficient algorithm to find the optimal policy, and shows that the convergence rate of this method does not depend on the (potentially exponentially large) eluder dimension, but only on the log-covering number.

The reviewers broadly agree that this paper makes a technically sound, substantial contribution. The also all worry that the presentation could be improved, and that the impact of this paper may suffer from this dense, highly technical presentation.

The revision uploaded by the authors seems to have improved the presentation a bit, though, so that this paper can now be accepted.

**Note From Pc:**

if the above contains the word "oral" or "spotlight" please see: "oral" presentation means -> notable-top-5% and "spotlight" means -> notable-top-25%. As stated in our emails, we are disassociating presentation type from AC recommendations

**Summary Of Ac-Reviewer Meeting:**

There was an extended email thread. The results are outlined above.